# Circadian rhythms of macrophages are altered by the acidic tumor microenvironment

Amelia M Knudsen-Clark [1], Daniel Mwangi [2], Juliana Cazarin [2], Kristina Morris[2], Cameron Baker [3], Lauren M Hablitz [4], Matthew N McCall[2,5,6], Minsoo Kim[1,6] & Brian J Altman [2,6 ✉]

## Abstract

Tumor-associated macrophages (TAMs) are prime therapeutic targets due to their pro-tumorigenic functions, but varying efficacy of macrophage-targeting therapies highlights our incomplete understanding of how macrophages are regulated within the tumor microenvironment (TME). The circadian clock is a key regulator of macrophage function, but how circadian rhythms of macrophages are influenced by the TME remains unknown. Here, we show that conditions associated with the TME such as polarizing stimuli, acidic pH, and lactate can alter circadian rhythms in macrophages. While cyclic AMP (cAMP) has been reported to play a role in macrophage response to acidic pH, our results indicate pH-driven changes in circadian rhythms are not mediated solely by cAMP signaling. Remarkably, circadian disorder of TAMs was revealed by clock correlation distance analysis. Our data suggest that hetero-geneity in circadian rhythms within the TAM population level may underlie this circadian disorder. Finally, we report that circadian regulation of macrophages suppresses tumor growth in a murine model of pancreatic cancer. Our work demonstrates a novel mechanism by which the TME influences macrophage biology through modulation of circadian rhythms.

**Keywords** Tumor Microenvironment; Macrophage; Circadian Rhythms; Immunology; Immuno-oncology
**Subject Categories** Cancer; Chromatin, Transcription & Genomics; Immunology

## Introduction

Tumor-associated macrophages (TAMs) are one of the most abundant leukocytes found in solid tumors, with high intra-tumoral TAM density generally associated with poor clinical outcome (Cassetta and Pollard, 2018; Gentles et al, 2015; Zhang et al, 2012). This is consistent with the largely pro-tumorigenic role

of macrophages within tumors (DeNardo and Ruffell, 2019). Macrophages are highly plastic professional phagocytes whose ability to sense and respond to the environment makes them uniquely equipped to protect tissue integrity under normal homeostatic conditions (Mosser and Edwards, 2008; Murray and Wynn, 2011). However, within the chronically inflamed tumor microenvironment (TME), failure to resolve the inflammation can lead to uncontrolled secretion of tissue repair factors by TAMs, promoting tumor growth and metastatic capacity (Coussens and Werb, 2002; Murray, 2018; Schoppmann et al, 2002). At the same time, conditions in the TME can drive TAMs to suppress potentially anti-tumorigenic inflammatory activity through various mechanisms including secretion of anti-inflammatory cytokines and expression of checkpoint inhibitors such as PD-L1, promoting immune suppression (Cook et al, 2013; DeNardo and Ruffell, 2019; Fadok et al, 1998; Graham et al, 2014; Huber et al, 2016b; Kuang et al, 2009; Roberts et al, 2017; Strassmann et al, 1994).

TAMs are known to suppress the response to many standard of care treatments through their pro-tumorigenic and immunosup-pressive functions, making them a prime therapeutic target (Ruffell and Coussens, 2015). However, we still have an incomplete understanding of how the TME influences macrophages, limiting our ability to target them; this is highlighted by the varying efficacy of therapeutic approaches used to target macrophages (Mantovani et al, 2022). This is thought to be due in part to the significant phenotypic heterogeneity of TAMs within tumors (Chevrier et al, 2017; Cuccarese et al, 2017; Mantovani et al, 2022). Evidence suggests this heterogeneity is due to the ability of macrophages to sense and adapt to the local microenvironment, which varies within tumors depending on several factors including the distance from blood vessels and neighboring cells (Huang et al, 2019; Laviron et al, 2022; Nalio Ramos et al, 2022; Yano et al, 2014). Indeed, various conditions in the TME have been shown to influence macrophage phenotype and function (DeNardo and Ruffell, 2019).

In particular, poor vascularization of solid tumors leads to inefficient delivery of oxygen, creating regions of hypoxia (Lyssiotis and Kimmelman, 2017). The hypoxic response promotes enhanced glycolytic activity of cells within the region, which, coupled with poor tissue drainage as a result of leaky vasculature, results in elevated levels of protons and lactate, acidifying the

[1]Department of Microbiology and Immunology, University of Rochester Medical Center, Rochester, NY, USA. [2]Department of Biomedical Genetics, University of Rochester Medical Center, Rochester, NY, USA. [3]Genomics Research Center, University of Rochester Medical Center, Rochester, NY, USA. [4]Center for Translational Neuromedicine, University of Rochester Medical Center, Rochester, NY, USA. [5]Department of Biostatistics and Computational Biology, University of Rochester Medical Center, Rochester, NY, USA. [6]Wilmot Cancer Institute, University of Rochester Medical Center, Rochester, NY, USA. ✉E-mail: Brian_Altman@URMC.rochester.edu

microenvironment (Gillies et al, 2019; Raghunand et al, 2003). Of conditions in the TME, it has been well appreciated that acidic (low) pH can promote a pro-resolution macrophage phenotype, thereby contributing to the pro-tumorigenic and immunosuppressive functions of macrophages within tumors (Bohn et al, 2018; El-Kenawi et al, 2019; Jiang et al, 2021b).

The myriad ways in which the TME can impact regulation of macrophages remain to be fully elucidated. Circadian rhythms are a key regulatory system present in almost all cells of the body, and are an understudied facet of macrophage biology (Partch et al, 2014). Acidic pH is a condition commonly associated with the TME that has been shown to alter circadian rhythms in cell lines (Walton et al, 2018); however, whether pH influences circadian rhythms in macrophages remain unknown.

Circadian rhythms are 24-h rhythms that impart oscillations in the levels of circadian-regulated gene transcripts and proteins in a tissue- and cell-specific manner, resulting in time-of-day-dependent variation in many cellular processes (Mure et al, 2018; Zhang et al, 2014). The molecular clock, which we will refer to as the circadian clock, drives these rhythms in cells through a cell-autonomous transcription/translation feedback loop, which is controlled in part by the transcription factor BMAL1 (Partch et al, 2014). Circadian clocks are synchronized by signals sent out from the central circadian clock housed within the suprachiasmatic nucleus of the hypothalamus, which entrains circadian clocks to the time of day (Partch et al, 2014). This allows for the temporal coordination of cells in spatially distinct tissues, although how the local microenvironment influences circadian rhythms remains to be elucidated.

All leukocytes tested to date have functional circadian clocks (Adrover et al, 2019; Arjona and Sarkar, 2005; Baumann et al, 2013; Druzd et al, 2017; Haspel et al, 2020; Keller et al, 2009; Nguyen et al, 2013; Silver et al, 2012a). As such, nearly every arm of the immune response (both innate and adaptive) is subject to circadian regulation (Haspel et al, 2020; Scheiermann et al, 2018; Silver et al, 2012a). Time-of-day-dependent regulation of immune responses is achieved through temporal gating of response to stimuli, effector function, and cell trafficking (Druzd et al, 2017; Fortier et al, 2011; Gibbs et al, 2012; He et al, 2018; Kitchen et al, 2020; Nobis et al, 2019), all of which promote coordination between the multiple phases of the immune response (Adrover et al, 2019; Beam et al, 2020; Druzd et al, 2017; Long et al, 2016; Mazzoccoli et al, 2011; Phillips et al, 2008; Silver et al, 2012b).

Key aspects of macrophage function are subject to circadian regulation, including cytokine secretion and phagocytosis (Curtis et al, 2015; Gibbs et al, 2012; Keller et al, 2009; Kitchen et al, 2020; Silver et al, 2012b). This results in a time-of-day-dependent macrophage response to stimuli, which modulates the magnitude of the resulting adaptive immune response and determines disease progression (Deng et al, 2018; Kitchen et al, 2020; Silver et al, 2012b). Circadian regulation of macrophages is of particular interest given recent evidence of 24-h circadian variation in the frequency of TAMs expressing surface markers associated with pro- or anti-tumorigenic phenotypes (Aiello et al, 2020; Strauss et al, 2020; Tsuruta et al, 2022). A promising application of such circadian variation was to leverage observations of circadian frequency in TAMs expressing immune checkpoint blockade (ICB) target PD-1 to subsequently increase efficacy of PD-1/PD-L1 ICB therapy by timing treatment to the time of day when PD-

1+ TAMs were most frequent (Qian et al, 2021; Tsuruta et al, 2022). This suggests that leveraging time-of-day variations in therapeutic targets could be a promising avenue to increase efficacy, highlighting the importance of understanding how circadian rhythms of macrophages may be influenced by conditions in the TME, which remains unclear.

In this work, we present evidence that circadian rhythms of macrophages are altered in the TME. We uncover a novel way in which two conditions within the TME, acidic pH and lactate, can influence macrophage biology through modulation of circadian rhythms. We also find that macrophages of different phenotypes have distinct circadian rhythms. Remarkably, we found evidence of circadian disorder in tumor-associated macrophages, indicating that circadian rhythms are altered in macrophages within the TME. Furthermore, our data suggest that heterogeneity in circadian rhythms at the population level may underlie the observed circadian disorder. This work elucidates a novel way in which the TME can alter macrophage biology, and represents the first steps to understanding how the tumor microenvironment can alter circadian rhythms of immune cells such as macrophages.

## Results

### Macrophages of different phenotypes exhibit different circadian rhythms

As macrophages are a phenotypically heterogeneous population in the TME, we first sought to understand whether diversity in macrophage phenotype could translate to diversity in circadian rhythms of macrophages. To this end, we used two well-established in vitro polarization models to study distinct macrophage phenotypes (Biswas and Mantovani, 2010; Mosser and Edwards, 2008; Munder et al, 1998; Murray, 2017; Viola et al, 2019). For a model of pro-inflammatory macrophages, we stimulated macrophages with interferon gamma (IFNγ) and lipopolysaccharide (LPS) to elicit a pro-inflammatory phenotype (Munder et al, 1998; Nathan et al, 1983). These macrophages are often referred to as 'M1' and are broadly viewed as anti-tumorigenic, and we will refer to them throughout this paper as 'pro-inflammatory' macrophages (Hörhold et al, 2020; Mills et al, 2000). For a model at the opposite end of the phenotypic spectrum, we stimulated macrophages with IL-4 and IL-13 (Munder et al, 1998; Stein et al, 1992). While these type 2 stimuli play a role in response to parasites and allergy, they are also major drivers of wound healing; in line with this, IL-4 and IL-13-stimulated macrophages have been well-characterized to adopt gene expression profiles associated with wound-healing and anti-inflammatory macrophage phenotypes (Allen, 2023; Mantovani et al, 2013; McWhorter et al, 2013; Pesce et al, 2009). As such, these macrophages are often used as a model to study pro-tumorigenic macrophages in vitro and are often referred to as 'M2' macrophages; throughout this paper, we will refer to IL-4 and IL-13-stimulated macrophages as 'pro-resolution' macrophages (Biswas et al, 2006; Ding et al, 2019; Mills et al, 2000). Consistent with previous studies, we found that genes associated with anti-inflammatory and pro-resolution programming characteristic of IL-4 and IL-13-stimulated macrophages such as *Arg1, Retnla, Chil3* (Ym1), *Clec10a* (MGL1), and *Mrc1* (CD206) were induced in IL-4 and IL-13-stimulated macrophages, but not IFNγ and

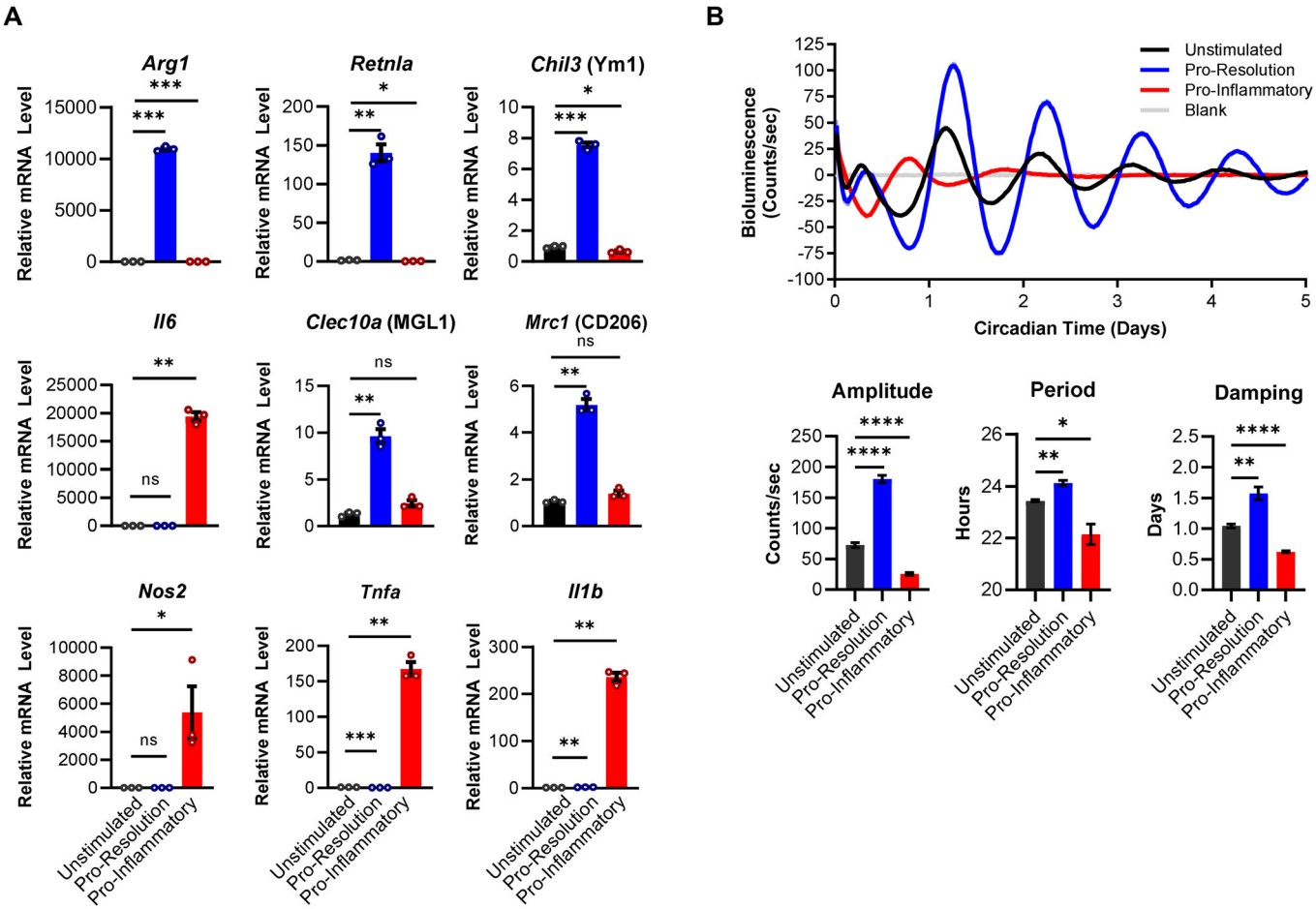

**Figure 1. Macrophages of different phenotypes have distinct circadian rhythms.**

Bone marrow-derived macrophages (BMDMs) were obtained from C57BL/6 mice expressing PER2-Luc. The circadian clocks of BMDMs were synchronized by a 24-h period of serum starvation in media with 0% serum, followed by a 2-h period of serum shock in media with 50% serum. BMDMs were then stimulated with either 10 ng/mL IL-4 and 10 ng/mL IL-13 (pro-resolution), or 50 ng/mL IFNγ and 100 ng/mL LPS (pro-inflammatory); or left unstimulated. (A) RNA was collected at 6 h post-synchronization, and qt-PCR was performed to assess expression of phenotype-associated genes. Shown are mean and standard error of the mean (SEM), $n = 3$ biological replicates. (B) Luciferase activity of BMDMs was monitored in real time by LumiCycle. Data was baseline-subtracted using the running average. Oscillation parameters of BMDMs were measured by LumiCycle Analysis. Shown are mean and standard error of the mean (SEM), $n = 5$ biological replicates. Data information: Statistical significance was determined by unpaired two-tailed t-test with Welch's correction of unstimulated vs pro-inflammatory or unstimulated vs pro-resolution; *$p < 0.05$; **$p < 0.005$; ***$p < 0.0005$; ****$p < 0.0001$; ns: not significant. Experiment was replicated twice. Exact p values: (A) *Arg1* 0.0002 (unstimulated vs pro-resolution), 0.0005 (unstimulated vs pro-inflammatory); *Retnla* 0.0061 (unstimulated vs pro-resolution), 0.0387 (unstimulated vs pro-inflammatory); *Chil3* 0.0003 (unstimulated vs pro-resolution), 0.0278 (unstimulated vs pro-inflammatory); *Il6* 0.0016 (unstimulated vs pro-inflammatory); *Clec10a* 0.0062 (unstimulated vs pro-resolution); *Mrc1* 0.0026 (unstimulated vs pro-resolution); *Nos2* 0.0462 (unstimulated vs pro-inflammatory); *Tnfa* 0.0006 (unstimulated vs pro-resolution), 0.0034 (unstimulated vs pro-inflammatory); *Il1b* 0.0029 (unstimulated vs pro-resolution), 0.0016 (unstimulated vs pro-inflammatory). (B) Amplitude <0.0001 (unstimulated vs pro-resolution), <0.0001 (unstimulated vs pro-inflammatory); Period 0.0018 (unstimulated vs pro-resolution), 0.0299 (unstimulated vs pro-inflammatory); Damping 0.0059 (unstimulated vs pro-resolution), <0.0001 (unstimulated vs pro-inflammatory).

LPS-stimulated macrophages. In contrast, genes associated with pro-inflammatory activity characteristic of IFNγ and LPS-stimulated macrophages such as *Nos2* (iNOS), *Tnfa, Il1b,* and *Il6* were induced in IFNγ and LPS-stimulated macrophages, but not IL-4 and IL-13-stimulated macrophages (Fig. 1A) (Binger et al, 2015; El-Kenawi et al, 2019; Hörhold et al, 2020; Jiang et al, 2021b; Martinez and Gordon, 2014; McWhorter et al, 2013). This indicates that macrophages stimulated with IL-4 and IL-13 were polarized toward a pro-resolution phenotype, while macrophages stimulated with IFNγ and LPS were polarized toward a pro-inflammatory phenotype.

Circadian rhythms of macrophages were measured by monitoring PER2, a key component of the circadian clock, via the rhythmic activity of the PER2-Luciferase (PER2-Luc) fusion protein in a live cell LumiCycle luminometer (Fig. EV1A) (Yoo et al, 2004). Bone marrow-derived macrophages (BMDMs) were generated from bone marrow of mice expressing PER2-Luc. Following differentiation, the circadian clocks of BMDMs were synchronized by a 24-h period of serum starvation followed by 2 h of serum shock (Collins et al, 2021), and rhythms were observed for up to 4 days (Fig. EV1B,C).

To determine whether phenotype can influence circadian rhythms in macrophages, BMDMs were cultured in the presence

or absence of polarizing stimuli, and rhythms were observed by LumiCycle (Fig. 1B). The amplitude of rhythms is the magnitude of change between the peak and the trough, and is indicative of the strength of rhythms (Rodheim et al, 2021; Wu et al, 2021). Amplitude of rhythms was suppressed in pro-inflammatory macrophages compared to unstimulated macrophages. In contrast, amplitude of rhythms in pro-resolution macrophages was enhanced. This suggests that rhythms are suppressed in pro-inflammatory macrophages but enhanced in pro-resolution macrophages, which agrees with previous observations (Chen et al, 2020; Lellupitiyage Don et al, 2022). Period is the amount of time it takes to complete one full oscillation (Pilorz et al, 2014). Compared to unstimulated macrophages, period was lengthened in pro-resolution macrophages but shortened in pro-inflammatory macrophages. This is in line with others' observations and suggests that the clock runs with a longer period (slower) in pro-resolution macrophages but runs with a shorter period (faster) in pro-inflammatory macrophages (Chen et al, 2020; Lellupitiyage Don et al, 2022).

Interestingly, we observed differences in damping of rhythms in polarized macrophages. Damping is measured as the number of days required for the amplitude of rhythms to decrease by 30% of the first cycle (Abe et al, 2002). Damping of rhythms in most free-running cell populations (defined as cells cultured in the absence of external synchronizing stimuli) occurs naturally as the circadian clocks of individual cells in the population become desynchronized from each other; thus, damping can be indicative of desynchrony within a population (Gaspar et al, 2019). The damping rate increases as the time it takes for rhythms to dissipate decreases; conversely, as damping rate decreases as the time it takes for rhythms to dissipate increases. We observed increased rate of damping in pro-inflammatory macrophages compared to unstimulated macrophages (Fig. 1B), indicating that population-level rhythms were maintained for a shorter length of time in pro-inflammatory macrophages. In contrast, damping rate was decreased in pro-resolution macrophages, indicating that population-level rhythms were maintained for longer in pro-resolution macrophages. These data suggest that pro-inflammatory macrophages may have an impaired ability to maintain synchrony, while pro-resolution macrophages may have an enhanced ability to maintain synchrony.

Collectively, these data suggest that pro-inflammatory macrophages have weaker rhythms and impaired ability to maintain synchrony, while pro-resolution macrophages have enhanced rhythms and an increased ability to maintain synchrony. This is evidence that macrophages of different phenotypes have distinct circadian rhythms, suggesting that diversity of macrophage phenotype may lead to diversity in macrophage circadian rhythms.

## Acidic pH alters circadian rhythms of macrophages

The TME has previously been shown to be acidic, with a pH ranging from 6.8 to 6.3; much more acidic than the typical pH in blood and healthy tissue of 7.3–7.4 (Boedtkjer and Pedersen, 2020; Estrella et al, 2013; Gillies et al, 1994). It was previously reported that acidic pH can alter circadian rhythms in cell lines, but whether this applies to macrophages remains unknown (Walton et al, 2018). Thus, we cultured BMDMs under conditions of varying pH within a range that mimics that found within the TME (pH 6.5–7.4). As

macrophages are a heterogeneous population in the TME, we assessed the influence of acidic pH on rhythms of unstimulated, pro-resolution, and pro-inflammatory macrophages. In line with previous observations, macrophages cultured at pH 6.5 were polarized toward a pro-resolution phenotype, characterized by increased expression of *Arg1* and *Vegf* compared to macrophages cultured at pH 7.4 (Fig. EV2A). Pro-inflammatory macrophages cultured at pH 6.5 had decreased expression of *Nos2* compared to those cultured at pH 7.4, suggesting that an acidic pH of 6.5 both promotes a pro-resolution phenotype and suppresses a pro-inflammatory phenotype.

It has been observed that inducible cyclic AMP early repressor (*Icer*), an isoform of cyclic AMP (cAMP)-response modulator (*Crem*), is upregulated downstream of acid-sensing in macrophages, and has been used as a "biomarker" for macrophages exposed to acidic conditions in tumors. We observed induction of *Icer* in unstimulated and pro-resolution macrophages cultured at pH 6.5 compared to pH 7.4, indicating that these macrophages were sensing acidic conditions (Fig. EV2B). In line with previous observations that *Icer* is induced downstream of LPS-driven TLR4 signaling, *Icer* was also upregulated in pro-inflammatory macrophages compared to unstimulated macrophages even at neutral pH 7.4 (Lv et al, 2017). Although *Icer* was not further upregulated in pro-inflammatory macrophages at pH 6.5 compared to pH 7.4, *Nos2* was suppressed at pH 6.5 compared to pH 7.4, suggesting that pro-inflammatory macrophages responded to acidic pH. In all, these data confirm that macrophages of various phenotypes can sense and respond to acidic conditions within the range of pH found in the TME.

To determine whether an acidic microenvironment influences circadian rhythms in macrophages, we assessed rhythms of unstimulated, pro-resolution, and pro-inflammatory macrophages under normal and acidic conditions. To this end, BMDMs were polarized toward a pro-resolution or a pro-inflammatory phenotype, or left unstimulated, and cultured in media at a normal pH of 7.4 or at acidic pH of 6.8 or 6.5; PER2-Luc rhythms were then observed by LumiCycle. In unstimulated and pro-resolution BMDMs, lower pH led to enhanced amplitude, a shortening in period, and increased damping rate of rhythms at pH 6.8 and pH 6.5 relative to neutral pH 7.4 (Fig. 2A,B; Appendix Fig. S1A). This suggests that in unstimulated and pro-resolution macrophages, acidic pH can strengthen rhythms by enhancing amplitude and speeding up the circadian clock, but may impair ability to maintain synchrony. Notably, changes in amplitude and period occurred in a dose-dependent fashion as pH decreased, indicating that rhythms are altered in a pH-dependent manner. In contrast, pro-inflammatory macrophages cultured at pH 6.5 exhibited suppressed amplitude, elongated period, and decreased damping rate of rhythms compared to those cultured at pH 7.4 (Fig. 2C; Appendix Fig. S1A). This suggests that in pro-inflammatory macrophages, acidic pH can weaken rhythms by decreasing amplitude and slowing down the speed of the clock, but may promote the ability to maintain synchrony. Low pH was also observed to alter the expression of the circadian clock genes *Per2*, *Cry1*, and *Nr1d1* (REV-ERBα) over time across different macrophage phenotypes, confirming that multiple components of the circadian clock are altered by acidic pH (Fig. 2D–F; Appendix Fig. S1B). Notably, the patterns in expression of circadian genes did not always match the patterns of PER2-Luc levels observed by LumiCycle. This is perhaps

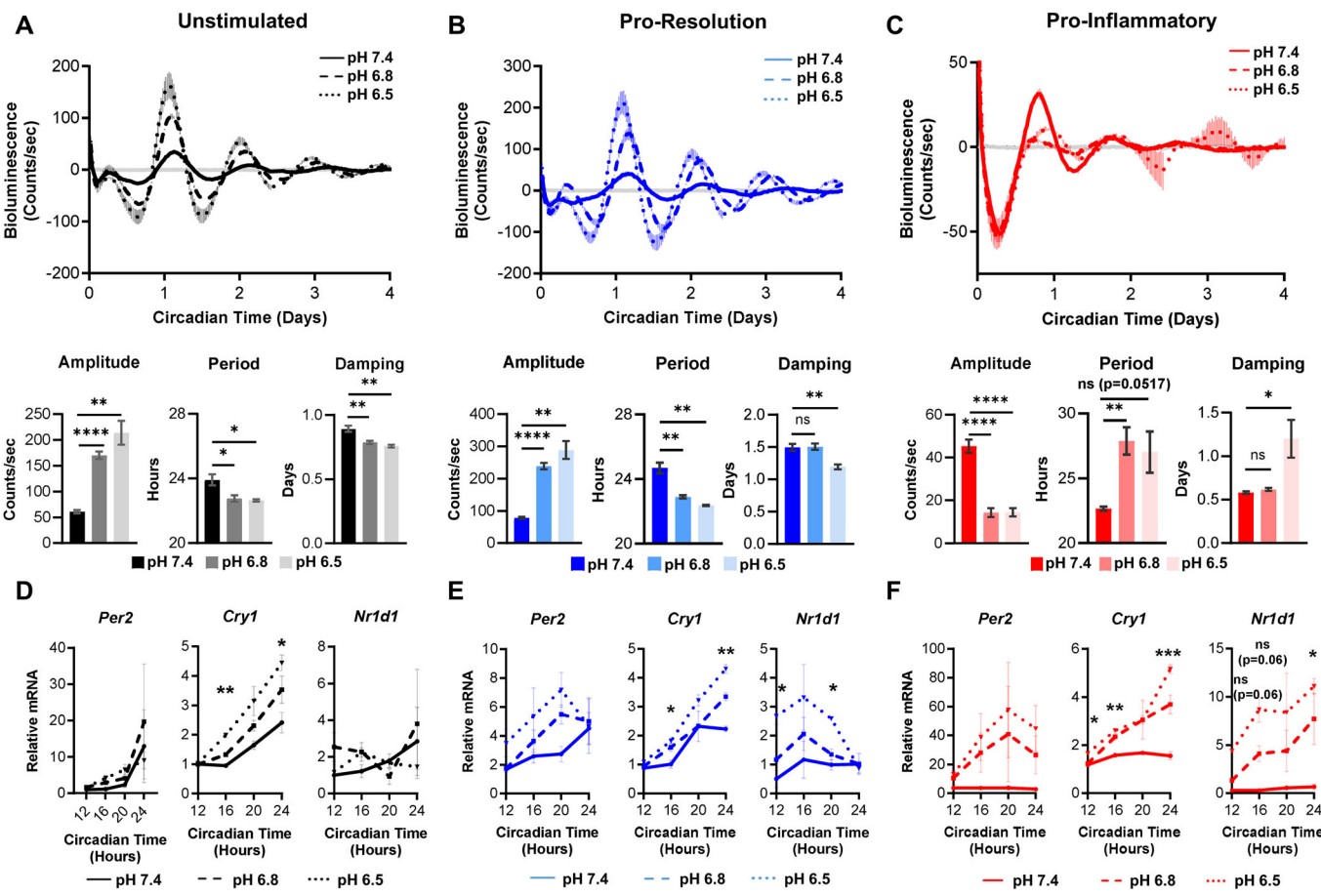

**Figure 2. Acidic pH alters circadian rhythms of bone marrow-derived macrophages in vitro.**

(A–F) Bone marrow-derived macrophages (BMDMs) were obtained from C57BL/6 mice expressing PER2-Luc. The circadian clocks of BMDMs were synchronized by a 24-h period of serum starvation in media with 0% serum, followed by a 2-h period of serum shock in media with 50% serum. BMDMs were then cultured in media with neutral pH 7.4 or acidic media with pH 6.8 or 6.5, and stimulated with either (B, E) 10 ng/mL IL-4 and 10 ng/mL IL-13 (pro-resolution), or (C, F) 50 ng/mL IFNγ and 100 ng/mL LPS (pro-inflammatory); or (A, D) left unstimulated. (A–C). Luciferase activity was monitored in real time by LumiCycle, n = 2 biological replicates. Data from both experiments was baseline-subtracted using the running average, and oscillation parameters were measured by LumiCycle Analysis, n = 5 biological replicates. (D–F). In parallel, RNA was collected at 12, 16, 20, and 24 h post-synchronization, and qt-PCR was performed to assess oscillation of transcripts encoding core clock proteins in macrophages under acidic conditions. n = 3 biological replicates. Data information: For (A–C), top panel (LumiCycle traces), shown are mean and SEM, experiment was replicated twice. For bottom panel (LumiCycle parameter analysis), shown are mean and SEM. For (D–F), shown are mean and SEM. Experiment was replicated twice. For all panels, statistical significance was determined by unpaired two-tailed t-test with Welch's correction of pH 7.4 vs pH 6.8 or 6.5 (pH 7.4 vs 6.8 was not tested in D–F). The Holm-Šídák correction for multiple t-tests was applied; *p < 0.05; **p < 0.005; ***p < 0.0005; ****p < 0.0001; ns: not significant. Exact p values: (A). Amplitude <0.0001 (pH 7.4 vs 6.8), 0.0025 (pH 7.4 vs 6.5); Period 0.0243 (pH 7.4 vs 6.8), 0.0184 (pH 7.4 vs 6.5); Damping 0.0098 (pH 7.4 vs 6.8), 0.0035 (pH 7.4 vs 6.5). (B) Amplitude <0.0001 (pH 7.4 vs 6.8), 0.0014 (pH 7.4 vs 6.5); Period 0.0039 (pH 7.4 vs 6.8), 0.002 (pH 7.4 vs 6.5); Damping 0.0022 (pH 7.4 vs 6.5). (C) Amplitude <0.0001 (pH 7.4 vs 6.8), <0.0001 (pH 7.4 vs 6.5); Period 0.0073 (pH 7.4 vs 6.8), 0.0517 (pH 7.4 vs 6.5); Damping 0.046 (pH 7.4 vs 6.5). (D) Cry1 0.005321 (pH 7.4 vs pH 6.5 CT 16), 0.03417 (pH 7.4 vs pH 6.5 CT 24). (E) Cry1 0035141 (pH 7.4 vs pH 6.5 CT 16), 0.002073 (pH 7.4 vs pH 6.5 CT 24); Nr1d1 0.025736 (pH 7.4 vs pH 6.5 CT 12), 0.036436 (pH 7.4 vs pH 6.5 CT 20). (F) Cry1 0.014339 (pH 7.4 vs pH 6.5 CT 12), 0.005812 (pH 7.4 vs pH 6.5 CT 16), 0.000525 (pH 7.4 vs pH 6.5 CT 24); Nr1d1 0.062514 (pH 7.4 vs pH 6.5 CT 12), 0.062514 (pH 7.4 vs pH 6.5 CT 16), 0.010905 (pH 7.4 vs pH 6.5 CT 24). Some 'ns' have been omitted from (D–F) for visual clarity.

unsurprising, as circadian rhythms are regulated at multiple levels (transcriptional, post-transcriptional, translational, post-translational); as a result, circadian patterns observed in circadian proteins such as PER2-Luc do not always match those of their gene transcripts (Collins et al, 2021). Together, these data indicate that exposure to acidic pH can induce changes in circadian rhythms of macrophages. Interestingly, our data indicate that while rhythms of unstimulated and pro-resolution macrophages are enhanced under acidic pH despite increased damping rate, rhythms of pro-inflammatory macrophages are suppressed under acidic conditions but have improved damping rate. This suggests that acidic pH

modulates rhythms differently in macrophage of different phenotypes.

The observation that acidic pH can enhance rhythms is particularly interesting, given that acidic pH is a stressful condition that can compromise macrophage survival (Appendix Fig. S2A) (Jiang et al, 2021b). In line with their documented enhanced glycolytic capacity, pro-inflammatory macrophages acidified the media over time (Appendix Fig. S2B). Notably, while pH of the media the pro-inflammatory macrophages were cultured in decreased over time pH, the pH differential between the pH 7.4, pH 6.8, and pH 6.5 samples groups of pro-inflammatory

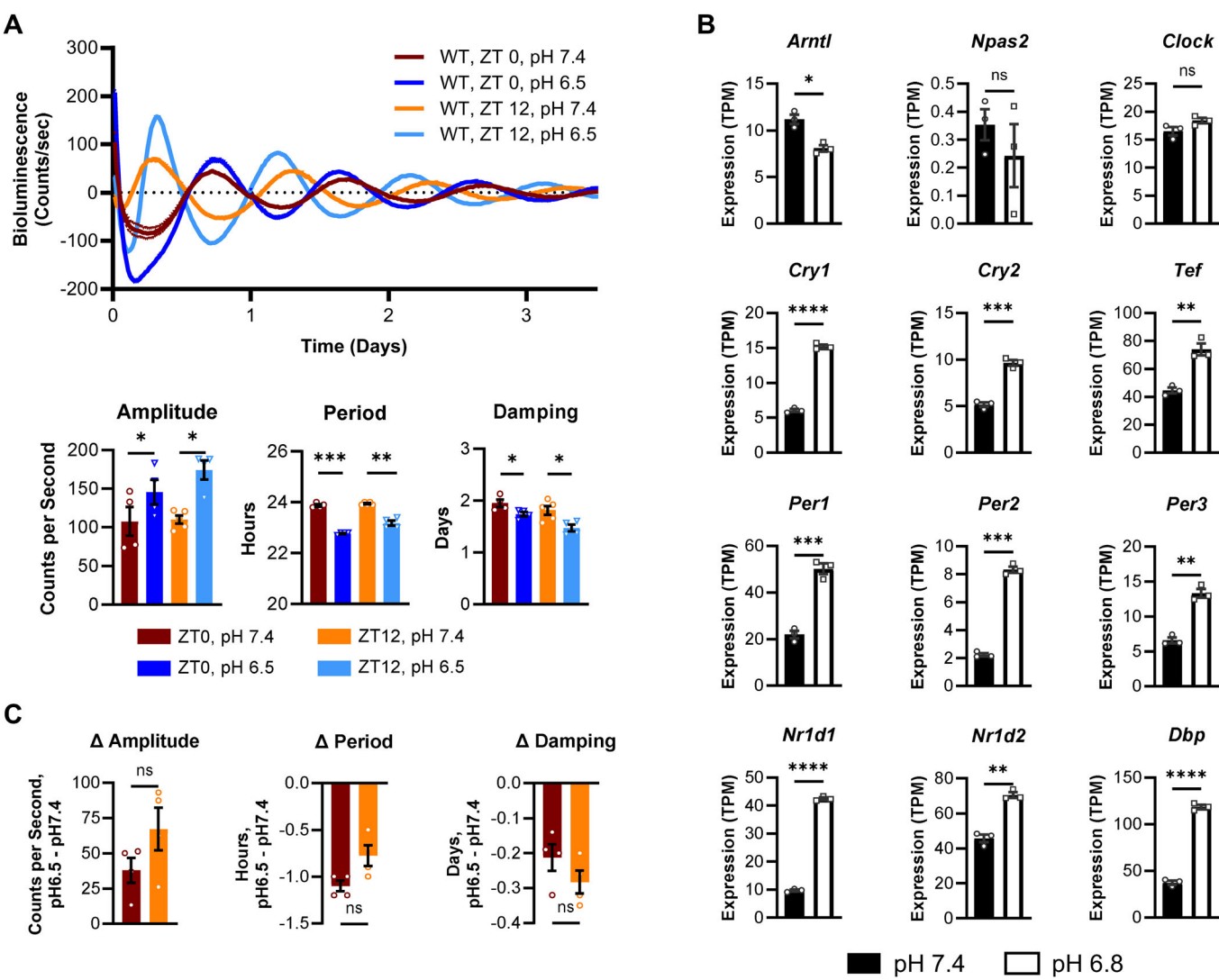

**Figure 3. Acidic pH alters circadian rhythms of peritoneal macrophages ex vivo at temporally distinct times of day.**

(A) Peritoneal macrophages were obtained at ZT0 or ZT12 from C57BL/6 mice expressing Per2-Luc and cultured in media with neutral pH 7.4 or acidic pH 6.5. (A) Luciferase activity was monitored in real time by LumiCycle. Data was baseline-subtracted using the running average; $n = 2$ biological replicates, each replicate and connecting line shown. Oscillation parameters were measured by LumiCycle Analysis; shown is the mean and SEM, $n = 4$ biological replicates, data pooled from 2 independent experiments. (B) Clock gene expression, in transcripts per million (TPM), of peritoneal macrophages cultured in media at pH 7.4 or pH 6.8 for 24 h, sourced from publicly available data (Data ref: GSE164697), $n = 3$ biological replicates [30]. (C) The magnitude of change in circadian oscillation parameters from (A) between macrophages at pH 7.4 and pH 6.5 was compared between peritoneal macrophages taken at ZT0 or ZT12. $n = 4$ biological replicates; data pooled from 2 independent experiments. Data information: For (A), shown are individual points and mean, experiment was replicated twice. For (B), shown is mean and SEM, For (C), shown is the mean and SEM experiment replicated twice. For all panels, statistical significance was determined by paired two-tailed t-test with Welch's correction; *$p < 0.05$; **$p < 0.005$; ***$p < 0.0005$; ns: not significant. Exact p values: (A) Amplitude 0.0231 (ZT0 pH 7.4 vs pH 6.5), 0.0204 (ZT12 pH 7.4 vs pH 6.5); Period 0.0003 (ZT0 pH 7.4 vs pH 6.5), 0.0043 (ZT12 pH 7.4 vs pH 6.5); Damping 0.0104 (ZT0 pH 7.4 vs pH 6.5), 0.0317 (ZT12 pH 7.4 vs pH 6.5). (B) *Arntl* 0.0103; *Cry1* < 0.0001; *Cry2* 0.0004; *Tef* 0.0093; *Per1* 0.0009; *Per2* 0.0001; *Per3* 0.0017; *Nr1d1* < 0.0001; *Nr1d2* 0.0012; *Dbp* < 0.0001.

macrophages was maintained out to 2 days, consistent with the changes in rhythms that we observe and measure between these groups.

While BMDMs are a widely used model for studying macrophages in vitro, there are biological differences between BMDMs generated in culture and tissue-resident macrophages. Thus, we sought to determine whether our observations of pH-induced changes in rhythms were relevant to tissue-resident macrophages

differentiated in vivo. To this end, we harvested peritoneal macrophages from mice expressing PER2-Luc either in the morning at ZT0 (6 AM) or in the evening at ZT12 (6 PM). Peritoneal macrophages were cultured in media at neutral pH of 7.4 or acidic pH of 6.5 and observed by LumiCycle. Recapitulating our results in BMDMs, peritoneal macrophages exhibited increased amplitude, decreased period, and increase rate of damping at pH 6.5 compared to pH 7.4 (Fig. 3A). To test whether pH-driven

changes in circadian rhythms of peritoneal macrophages were reflected at the mRNA level, we compared expression of circadian clock genes in peritoneal macrophages cultured at neutral pH 7.4 or acidic pH 6.8 for 24 h using publicly available RNA-sequencing data (Jiang et al, 2021b; Jiang et al, 2021a). In line with altered circadian rhythms observed by Lumicycle, peritoneal macrophages cultured at pH 6.8 expressed different levels of circadian clock genes than peritoneal macrophages culture at pH 7.4 (Fig. 3B). The trends in changes of gene expression in peritoneal macrophages cultured at pH 6.8 matched what we observed in BMDMs, where low pH generally led to higher levels of circadian clock gene expression (Fig. 2D–F). These data support our observations by LumiCycle and indicate that acidic pH drives transcriptional changes in multiple components of the circadian clock. In all, these data are evidence that pH-dependent changes in circadian rhythms are relevant to in vivo-differentiated macrophages.

Circadian rhythms confer time-of-day variability in response to stimuli. As we have observed that acidic pH can influence circadian rhythms of peritoneal macrophages, we sought to understand if peritoneal macrophages would be more or less susceptible to pH-induced changes in rhythms depending on time of day of exposure. To this end, we compared the magnitude of change in amplitude, period, and damping in peritoneal macrophages when exposed to acidic pH 6.5 compared to neutral pH 7.4 at different times of day (Fig. 3C). We observed no significant difference in the pH-driven change in amplitude, period, or damping in rhythms of peritoneal macrophages taken in the morning at ZT0 compared to those taken in the evening at ZT12. This indicates that the influence of pH on rhythms of macrophages was similar when exposed to acidic pH in the morning or in the evening, which suggests that macrophages are similarly susceptible to pH-induced changes in rhythms regardless of time of day of exposure.

## Lactate alters circadian rhythms of macrophages in a manner distinct from acidic pH

Elevated lactate concentrations often co-localize to regions of high acidity, due to the export of both protons and lactate by glycolytic cells (Kraut and Madias, 2014; Pastorekova and Gillies, 2019; Swietach, 2019). In tumors, lactate has been observed in concentrations of up to 30 mM, which is elevated over typical lactate levels in blood and healthy tissue of 1.5–3 mM (de la Cruz-López et al, 2019). There are previous reports that lactic acid can promote polarization of macrophages toward a pro-resolution phenotype (Colegio et al, 2014). Thus, we sought to understand if lactate may be a feature of the TME capable of influencing circadian rhythms of macrophages, in addition to acidic pH. To this end, we cultured BMDMs in the presence or absence of 25 mM sodium-L-lactate. In line with previous observations, BMDMs exposed to lactate had elevated levels of *Vegf*; however, we did not observe significant elevation of *Arg1* (Fig. 4A) (Colegio et al, 2014).

We next cultured BMDMs at normal pH 7.4 or acidic pH 6.5, in the presence or absence of 25 mM sodium-L-lactate, and monitored circadian rhythms (Fig. 4B,C; Appendix Fig. S3A). Rhythms of BMDMs at pH 7.4 exposed to lactate had an elongated period and decreased damping time compared to BMDMs cultured at pH 7.4 without lactate (Fig. 4B). This suggests that lactate can slow the circadian clock and may impair the ability of macrophages to

maintain synchrony. Interestingly, these changes in rhythms are different from those observed in acidic conditions, indicating that lactate can modulate circadian rhythms in macrophages in a manner distinct from acidic pH.

As previously observed, macrophages exposed to acidic pH 6.5 exhibited increased amplitude, shortened period, and increased damping rate of circadian rhythms. When BMDMs were exposed to both acidic pH and elevated lactate, the increased amplitude observed at pH 6.5 was maintained; however, the shortened period observed at pH 6.5 was lost, with period lengthened in BMDMs cultured in 25 mM sodium-l-lactate (Fig. 4B). The increased damping rate of rhythms in BMDMs cultured at pH 6.5 compared to pH 7.4 was maintained, and was further dampened by exposure to lactate (Fig. 4B). These data indicate that changes in rhythms associated with acidic conditions persisted when co-exposed to elevated lactate. Lactate was also observed to alter expression of the circadian clock genes *Per2*, *Cry1*, and *Nr1d1* over time in BMDMs cultured at pH 6.5, while having more subtle effects at pH 7.4 (Fig. 4C). Notably, lactate blunted the effect of pH 6.5 on *Cry1* expression, while enhancing the effect of low pH on *Nr1d1* expression. In all, these data indicate that concentration of lactate similar to that present in the TME can influence circadian rhythms and circadian clock gene expression of macrophages. Lactate altered rhythms differently than acidic pH, and when macrophages were exposed to acidic pH and lactate together, rhythms were further altered. This suggests that when macrophages are exposed to multiple conditions capable of altering circadian rhythms, each condition may contribute to a combined effect on rhythms that differs from its individual impact.

## Cancer cell supernatant alters circadian rhythms in macrophages in a manner partially reversed by neutralization of pH

We have observed that polarizing stimuli, acidic pH, and lactate can alter circadian rhythms. However, the tumor microenvironment is complex. Cancer cells secrete a variety of factors and deplete nutrients in the environment. To model this, we cultured BMDMs in RPMI or supernatant collected from KCKO cells, which are a murine model of pancreatic ductal adenocarcinoma (PDAC) (Besmer et al, 2011; Kidiyoor et al, 2014), at pH 6.5 or neutralized to pH 7.4 (Fig. EV3; Appendix Fig. S3B). Circadian rhythms of BMDMs cultured in cancer cell supernatant at pH 7.4 or pH 6.5 exhibited increased amplitude and lengthened period compared to RPMI control at pH 7.4 or 6.5, respectively, indicating that cancer cell supernatant contains factors that can alter circadian rhythms of BMDMs. Notably, BMDMs cultured in cancer cell supernatant at pH 6.5 had increased amplitude and shortened period compared to BMDMs cultured in cancer cell-conditioned media at pH 7.4, indicating that pH-driven changes in rhythms were maintained in BMDMs cultured in cancer cell supernatant. When the pH of cancer cell supernatant was neutralized to pH 7.4, the increased amplitude was decreased, and the shortened period was lengthened, indicating that neutralizing acidic pH partially reverses the changes in rhythms observed in macrophages cultured in cancer cell supernatant at pH 6.5. These data further support our observations that acidic pH can alter circadian rhythms of macrophages both alone and in combination with various factors in the TME.

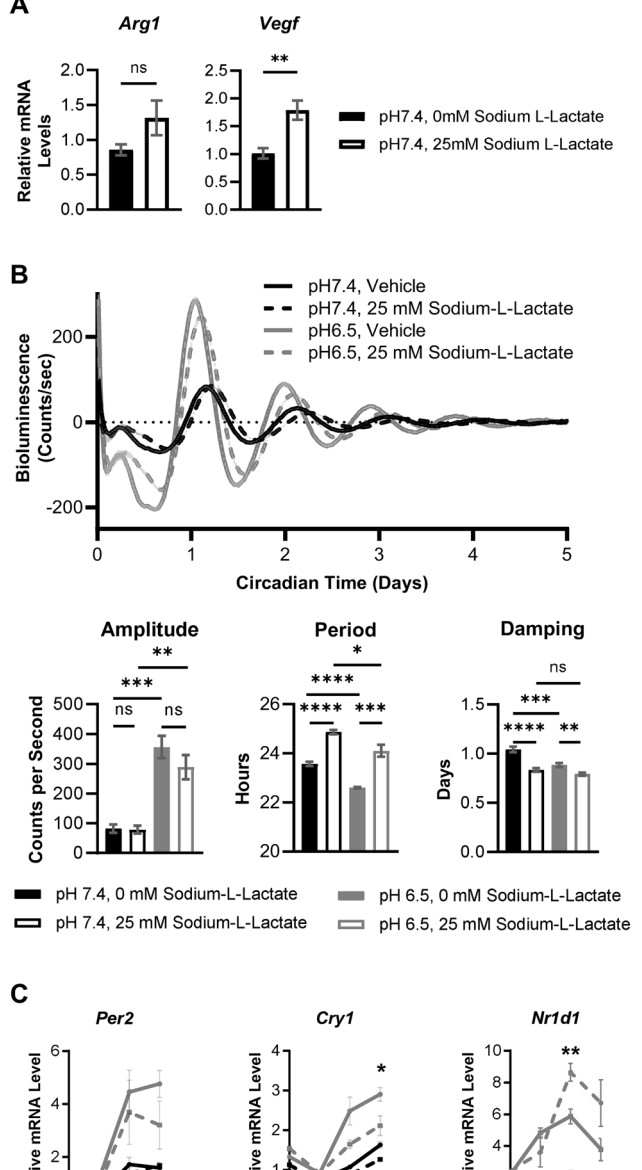

**A**

Arg1 / Vegf — Relative mRNA Levels
■ pH7.4, 0mM Sodium L-Lactate
□ pH7.4, 25mM Sodium L-Lactate

**B**

Bioluminescence (Counts/sec) vs Circadian Time (Days)
— pH7.4, Vehicle
---- pH7.4, 25 mM Sodium-L-Lactate
— pH6.5, Vehicle
---- pH6.5, 25 mM Sodium-L-Lactate

Amplitude (Counts per Second) / Period (Hours) / Damping (Days)
■ pH 7.4, 0 mM Sodium-L-Lactate
□ pH 7.4, 25 mM Sodium-L-Lactate
■ pH 6.5, 0 mM Sodium-L-Lactate
□ pH 6.5, 25 mM Sodium-L-Lactate

**C**

Per2 / Cry1 / Nr1d1 — Relative mRNA Level vs Circadian Time (Hours)
— pH 7.4, 0 mM Sodium-L-Lactate
---- pH 7.4, 25 mM Sodium-L-Lactate
— pH 6.5, 0 mM Sodium-L-Lactate
---- pH 6.5, 25 mM Sodium-L-Lactate

**Figure 4. Lactate alters circadian rhythms in macrophages, both alone and in conjunction with acidic pH.**

(A–C) Bone marrow-derived macrophages (BMDMs) were obtained from C57BL/6 mice expressing Per2-Luc. The circadian clocks of BMDMs were synchronized by a 24-h period of serum starvation in media with 0% serum, followed by a 2-h period of serum shock in media with 50% serum. BMDMs were then cultured in media with neutral pH 7.4 or acidic pH 6.5, supplemented with 0 mM or 25 mM sodium-L-lactate. (A) RNA was collected at 6 h post-treatment, and expression of pro-resolution phenotype markers Arg1 or Vegf was quantified by qtPCR. n = 6 biological replicates. (B) Luciferase activity was monitored in real time by LumiCycle. Data was baseline-subtracted using the running average, n = 4 biological replicates. Oscillation parameters were measured by LumiCycle Analysis. Data was pooled from 2 replicated experiments (pH 7.4, 0 mM sodium-L-lactate: n = 10. pH 7.4, 25 mM sodium-L-lactate, n = 10. pH 6.5, 0 mM sodium-L-lactate: n = 7. pH 6.5, 25 mM sodium-L-lactate, n = 7, all replicates were biological replicates). (C) RNA was collected at 12, 16, 20, and 24 h post-synchronization, and qt-PCR was performed to assess oscillation of transcripts encoding core clock proteins in macrophages exposed to lactate. Data was pooled from 2 independent replicated experiments. (pH 7.4, 0 mM sodium-L-lactate: n = 6. pH 7.4, 25 mM sodium-L-lactate: n = 6. pH 6.5, 0 mM sodium-L-lactate: n = 5. pH 6.5, 25 mM sodium-L-lactate: n = 6, all replicates were biological replicates). Data information: For (A), shown is the mean and SEM, experiment replicated twice. For (B), top panel (LumiCycle traces), shown is the mean and SEM, experiment was replicated twice. For bottom panel (LumiCycle parameter analysis), shown is the mean and SEM. For (C), shown is the mean and SEM. For all panels, statistical significance was determined by unpaired two-tailed t-test with Welch's correction for pH 7.4 0 mM sodium-L-lactate vs 25 mM mM sodium-L-lactate and pH 6.5 0 mM sodium-L-lactate vs 25 mM sodium-L-lactate (other comparisons were not performed on these data). The Holm-Šídák correction for multiple t-tests was applied; $*p < 0.05$; $**p < 0.005$; $***p < 0.0005$; $****p < 0.0001$; ns: not significant. Exact p values: (A) Vegf 0.0045. (B) Amplitude 0.0002 (pH 7.4 0 mM sodium-L-lactate vs pH 6.5 0 mM sodium-L-lactate), 0.0015 (pH 7.4 25 mM sodium-L-lactate vs pH 6.5 25 mM sodium-L-lactate); Period <0.0001 (pH 7.4 0 mM sodium-L-lactate vs pH 6.5 0 mM sodium-L-lactate), <0.0001 (pH 7.4 0 mM sodium-L-lactate vs pH 7.4 25 mM sodium-L-lactate), 0.0209 (pH 7.4 25 mM sodium-L-lactate vs pH 6.5 25 mM sodium-L-lactate), 0.0008 (pH 6.5 0 mM sodium-L-lactate vs pH 6.5 25 mM sodium-L-lactate); Damping 0.0003 (pH 7.4 0 mM sodium-L-lactate vs pH 6.5 0 mM sodium-L-lactate), <0.0001 (pH 7.4 0 mM sodium-L-lactate vs pH 7.4 25 mM sodium-L-lactate), 0.0036 (pH 6.5 0 mM sodium-L-lactate vs pH 6.5 25 mM sodium-L-lactate). (C) Cry1 0.04153 (pH 7.4 0 mM sodium-L-lactate vs pH 7.4 25 mM sodium-L-lactate CT 20), 0.00594 (pH 7.4 0 mM sodium-L-lactate vs pH 7.4 25 mM sodium-L-lactate CT 24), 0.028433 (pH 6.5 0 mM sodium-L-lactate vs pH 6.5 25 mM sodium-L-lactate CT 24); Nr1d1 0.015989 (pH 7.4 0 mM sodium-L-lactate vs pH 7.4 25 mM sodium-L-lactate CT 20), 0.003984 (pH 6.5 0 mM sodium-L-lactate vs pH 6.5 25 mM sodium-L-lactate CT 20). Some 'ns' have been omitted from (C) for visual clarity.

## Induction of cAMP signaling alone is not sufficient to fully drive changes in circadian rhythms associated with acidic pH

Evidence in the literature suggests that acidic pH is primarily sensed by macrophages via certain G protein-coupled receptors (GPCRs), inducing an increase in intracellular cAMP that drives downstream signaling through the cAMP pathway (Bohn et al, 2018). Transcriptional changes downstream of cAMP signaling subsequently promote a pro-resolution phenotype (Bohn et al, 2018; Polumuri et al, 2021; Tavares et al, 2020). Transcription of the Crem isoform Icer is also induced downstream of cAMP

signaling, and has been used as a "biomarker" for macrophages exposed to acidic conditions in tumors (Bohn et al, 2018). In line with previous reports, we observed induction of Icer in macrophages under acidic pH (Fig. EV2B), suggesting that cAMP signaling is being induced under acidic conditions (Bohn et al, 2018). This occurs as early as 2 h, concurrent with changes in rhythms, which are observed by 6 h following exposure to acidic conditions. It has been shown that induction of cAMP signaling alone is sufficient to drive a pro-resolution phenotype in macrophages similar to that observed under acidic conditions (Bohn et al, 2018; Polumuri et al, 2021). In addition, cAMP signaling has been previously observed to modulate circadian rhythms in SCN and rat fibroblasts (O'Neill et al, 2008; Yagita and Okamura, 2000). Thus, we sought to understand if the cAMP signaling pathway may be mediating the pH-induced changes in circadian rhythms in macrophages.

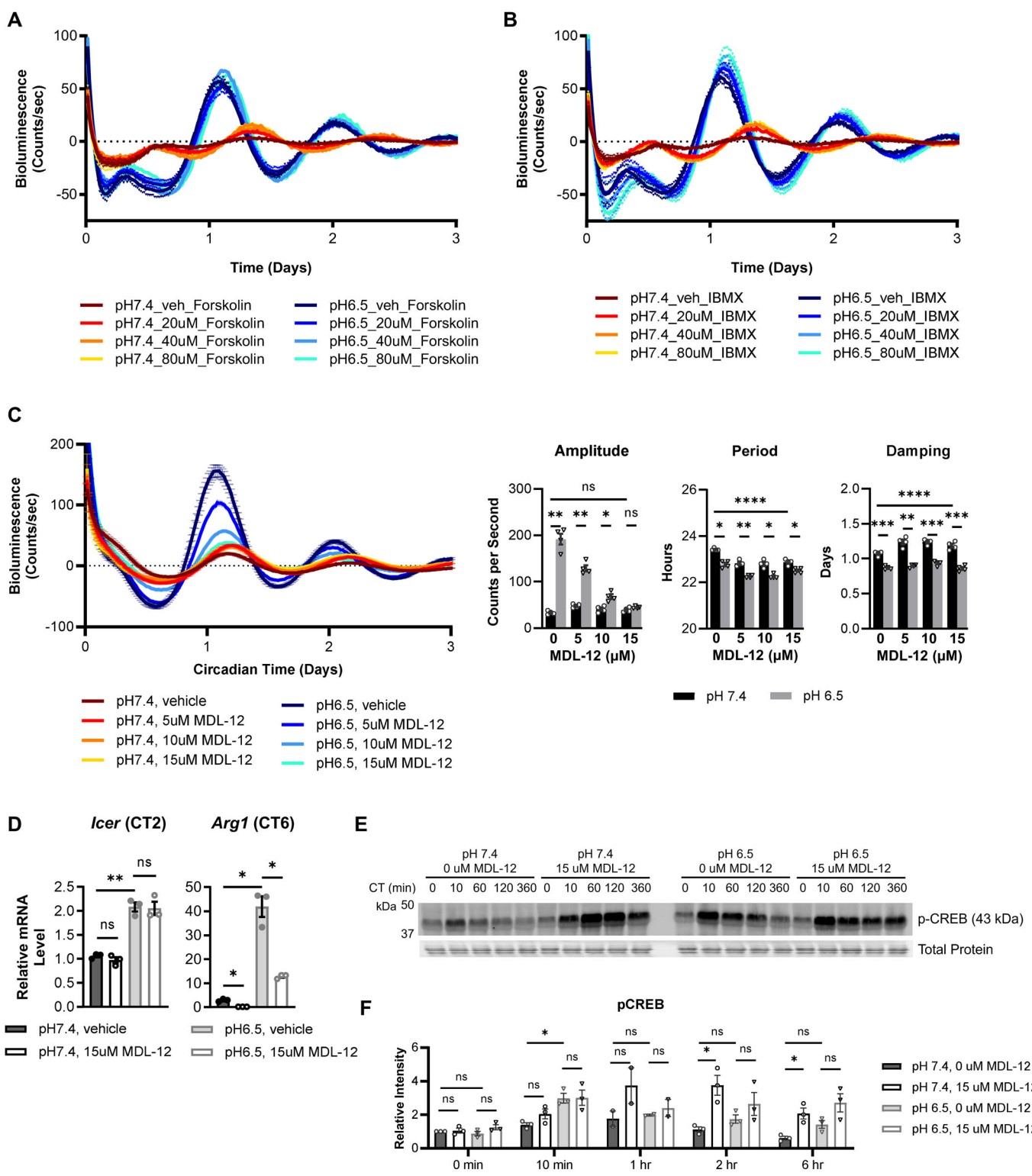

The synchronization protocol we use to study circadian rhythms in BMDMs involves a 24-h period of serum starvation followed by 2 h of serum shock. It has previously been shown that serum shock can induce signaling through the cAMP pathway in rat fibroblasts (Yagita and Okamura, 2000). To determine whether the synchronization protocol impacts cAMP signaling in macrophages, we harvested macrophages before and after serum shock. We then assessed *Icer* expression and phosphorylation of cyclic

Figure 5. pH-induced changes in circadian rhythms are not driven by cAMP signaling alone.

(A, B) Bone marrow-derived macrophages (BMDMs) were obtained from C57BL/6 mice expressing Per2-Luc. BMDMs were cultured in media with neutral pH 7.4 or acidic pH 6.5, and (A) treated with vehicle or 20, 40, or 80 μM Forsokolin, or (B) treated with vehicle or 20, 40, 80 μM IBMX. Luciferase activity was monitored in real time by LumiCycle. Data was baseline-subtracted using the running average, $n = 2$ biological replicates. (C) The circadian clocks of BMDMs were synchronized by a 24-h period of serum starvation in media with 0% serum, followed by a 2-h period of serum shock in media with 50% serum. BMDMs were then cultured in media with neutral pH 7.4 or acidic pH 6.5, and treated with vehicle or 5, 10, or 15 μM MDL-12. Luciferase activity was monitored in real time by LumiCycle. Data was baseline-subtracted using the running average, and oscillation parameters were measured by LumiCycle Analysis, $n = 4$ biological replicates. (D) Expression of the acid sensing gene (Icer) was measured at 2 h post treatment (CT2), and gene associated with a pro-resolution phenotype (Arg1) was measured at 6 h post-treatment (CT6), $n = 3$ biological replicates. (E, F). (E) Phosphorylation of CREB was assessed by immunoblot at the indicated timepoints after synchronization, MDL-12 treatment, and exposure to low pH. (F) Relative pCREB levels from (E) were quantified; band intensity was normalized to total protein, $n = 3$ biological replicates. Data information: For (A, B), shown are individual points and mean. For (C, D, F), mean and SEM are shown. Each experiment was replicated at least two times. For (C, right panels), statistical significance was determined with two-way ANOVA to compare all pH 7.4 vs pH 6.5 data points, and for each individual MDL-12 comparison, unpaired two-tailed t-tests with Welch's correction and the Holm–Šídák correction for multiple t-tests were applied. For (D, F), statistical significance was determined by unpaired two-tailed t-test with Welch's correction (pH 7.4 vehicle vs pH 6.5 15 μM MDL-12 was not tested); *$p < 0.05$; **$p < 0.005$; ***$p < 0.0005$; ****$p < 0.0001$; ns: not significant. Exact P values: Amplitude 0.002346 (pH 7.4 vs pH 6.5 vehicle), 0.002346 (pH 7.4 vs pH 6.5 5 μM MDL-12), 0.01276 (pH 7.4 vs pH 6.5 10 μM MDL-12); Period <0.0001 (pH 7.4 vs pH 6.5 by two-way ANOVA), 0.005197 (pH 7.4 vs pH 6.5 vehicle), 0.003662 (pH 7.4 vs pH 6.5 5 μM MDL-12), 0.005197 (pH 7.4 vs pH 6.5 10 μM MDL-12), 0.005197 (pH 7.4 vs pH 6.5 15 μM MDL-12); Damping <0.0001 (pH 7.4 vs pH 6.5 by two-way ANOVA), 0.000307 (pH 7.4 vs pH 6.5 vehicle), 0.00119 (pH 7.4 vs pH 6.5 5 μM MDL-12), 0.000105 (pH 7.4 vs pH 6.5 10 μM MDL-12), 0.000307 (pH 7.4 vs pH 6.5 15 μM MDL-12). (D) Icer 0.0053 (pH 7.4 vehicle vs pH 6.5 vehicle); Arg1 0.0148 (pH 7.4 vehicle vs pH 7.4 15 μM MDL-12), 0.0117 (pH 7.4 vehicle vs pH 6.5 vehicle), 0.0202 (pH 6.5 vehicle vs pH 6.5 15 μM MDL-12). (F) 10 min 0.01823 (pH 7.4 vs pH 6.5 0 μM MDL-12); 2 hr 0.03913 (pH 7.4 0 μM MDL-12 vs pH 7.4 15 μM MDL-12); 6 hr 0.036106 (pH 7.4 0 μM MDL-12 vs pH 7.4 15 μM MDL-12).

AMP-response element binding protein (CREB), which occur downstream of cAMP and have been used as readouts to assess induction of cAMP signaling in macrophages (Bohn et al, 2018; Misra and Pizzo, 2005; Polumuri et al, 2021). Serum shock of macrophages following serum starvation led to rapid phosphorylation of CREB and Icer expression that quickly returned to baseline (Fig. EV1D,E). This indicates that serum starvation followed by serum shock in the synchronization protocol we use to study circadian rhythms in BMDMs induces transient signaling through the cAMP signaling pathway.

As acidic pH induces signaling through the cAMP pathway, we sought to determine whether acidic pH independently contributed to the pH-driven changes in circadian rhythms we observe in BMDMs. To test this, we omitted the synchronization step and observed BMDM rhythms by LumiCycle when cultured in neutral pH 7.4 or acidic pH 6.8 or pH 6.5 (Fig. EV4). Circadian rhythms of BMDMs cultured at pH 6.5 exhibited similar changes as previously observed, with enhanced amplitude and shortened period relative to BMDMs at pH 7.4. This indicates pH-driven changes observed in circadian rhythms of BMDMs occur in the absence of prior serum starvation and serum shock.

To determine if elevation in intracellular cAMP alone was sufficient to drive changes in rhythms observed in macrophages under acidic conditions, we treated macrophages with forskolin, an adenylyl cyclase activator that stimulates production of cAMP, or 3-isobutyl-1-methylxanthine (IBMX), which drives accumulation of cAMP through inhibition of phosphodiesterases (PDEs). We used a range of doses similar to those previously shown to induce cAMP signaling in macrophages in the literature (Bohn et al, 2018; Misra and Pizzo, 2005; Yagita and Okamura, 2000). Treatment with either forskolin or IBMX increased amplitude of rhythms in macrophages, but not to the same magnitude as acidic pH, and did not result in a changed period (Fig. 5A,B). Moreover, amplitude of rhythms was not altered in forskolin- or IBMX-treated macrophages at pH 6.5, indicating that neither forskolin nor IBMX had any additional effect on rhythms under acidic conditions. These data indicate that in macrophages, cAMP signaling alone induces enhanced amplitude of rhythms similar to low pH, but the magnitude of this change is far less; additionally, period, which is altered under acidic conditions, remains unchanged. This suggests that cAMP signaling may contribute to but is not sufficient to fully recapitulate the changes in rhythms observed under acidic conditions.

## Adenylyl cyclase inhibitor MDL-12330A suppresses pH-mediated changes in amplitude of circadian rhythms and pro-resolution phenotype without suppressing cAMP signaling

To further test whether pH-induced changes in rhythms are mediated by cAMP signaling, we treated BMDMs with MDL-12330A (henceforth referred to as MDL-12), an adenylyl cyclase inhibitor which has previously been shown to suppress cAMP signaling in macrophages under acidic conditions (Bohn et al, 2018). When BMDMs cultured at pH 6.5 were treated with MDL-12, the elevated amplitude of rhythms observed at pH 6.5 was suppressed (Fig. 5C; Appendix Fig. S3C). Notably, this occurred in a dose-dependent manner, suggesting that this is a drug-dependent effect. Importantly, rhythms of MDL-12-treated macrophages at pH 7.4 had similar amplitude to vehicle-treated macrophages at pH 7.4. This suggests that the inhibitory effect of MDL-12 on pH-induced enhancement of amplitude in macrophage rhythms was specific to acidic conditions. However, MDL-12 treatment of macrophages at pH 7.4 resulted in shortened period and decreased damping rate compared to vehicle-treated macrophages. Together, MDL-12-mediated suppression of pH-driven changes in amplitude, but not period or damping, suggests that the pH-driven changes in these different parameters of rhythms may occur through different pathways. Interestingly, although the adenylyl cyclase inhibition by MDL-12 is reported to be irreversible, we found that pre-treatment up to 2 h was not sufficient to suppress pH-induced changes in amplitude (Appendix Fig. S4). Only when macrophages continued to be cultured with MDL-12 while exposed to acidic conditions was amplitude suppressed. Meanwhile, co-treating cells with acidic pH and MDL12 without any pre-treatment was sufficient to suppress elevation of amplitude under acidic conditions (Fig. 5C).

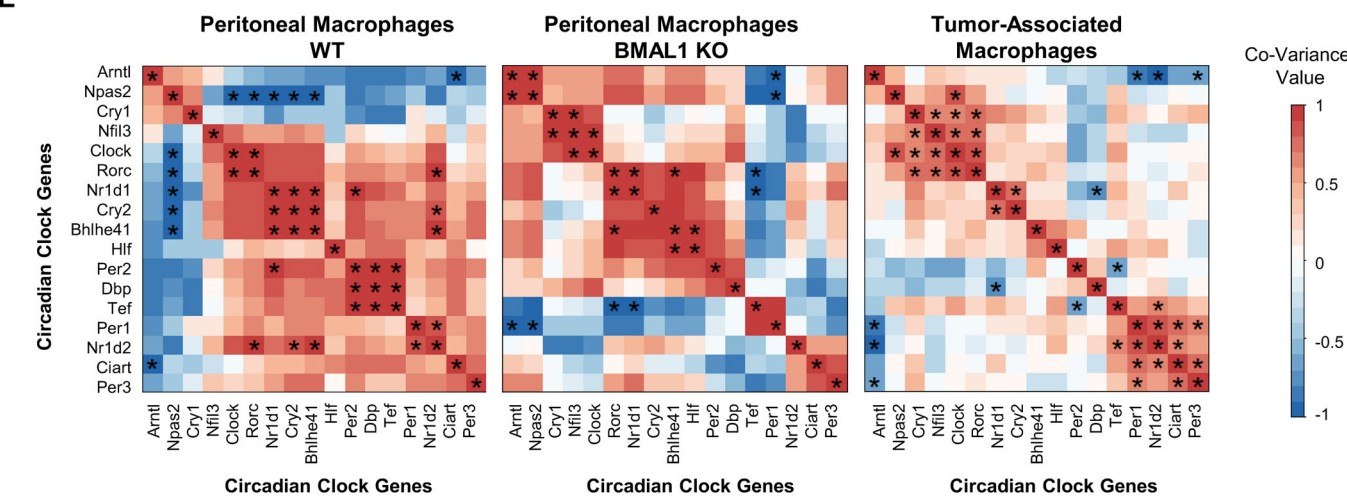

Figure 6. Clock correlation distance (CCD) and weighted gene co-expression network analysis (WGCNA) provide evidence of circadian disorder in murine tumor-associated macrophages.

(A) RNAseq datasets (see Methods) of WT peritoneal macrophages (pMac), BMAL1 KO peritoneal macrophages, and tumor-associated macrophages (TAMs) were analyzed for expression of *Arg1* and *Crem* (WT pMac: $n = 5$. BMAL1KO pMac: $n = 4$. TAMs: $n = 10$, all replicates are biological replicates). (B) A pan-tissue murine reference control was used for clock correlation distance (CCD). (C, D). (C) Clock correlation distance (CCD) analysis was performed and (D) statistical analysis to compare CCD scores was performed by calculating delta CCD. For BMDMs, $n = 72$ biological replicates, for all other conditions, see (A). (E) Weighted gene co-expression network analysis (WGCNA) was performed on the indicated circadian clock genes using data from (A–D). Data information: For (A), mean and SEM are shown. Statistical significance was determined with Ordinary one-way ANOVA with Tukey's multiple comparisons test, ****$p < 0.0001$ for each indicated comparison. For (D), *$p < 0.05$ by delta CCD analysis compared to control group only (pMacs WT). For (E), *$p < 0.05$ by WGCNA for significant covariance by Pearson correlation, and blank squares are not significant. Exact $p$ values: (D) 0.00802 (pMacs WT vs pMacs BMAL1KO), 0.007826 (pMacs WT vs TAMs).

Evidence suggests that acidic pH signals through the cAMP pathway to promote a pro-resolution phenotype in macrophages, with induction of *Icer* occurring directly downstream of cAMP signaling (Bohn et al, 2018). Despite preventing changes in amplitude under acidic pH, MDL12 treatment at the dose and treatment schedule used does not suppress induction of *Icer* in macrophages under acidic conditions (Fig. 5D). However, induction of *Arg1* expression in macrophages under acidic conditions was suppressed by MDL-12. This suggests that at the dose and treatment strategy used, MDL-12 partially suppresses the response of macrophages to acidic pH by suppressing the pH-driven polarization toward a pro-resolution phenotype and changes in amplitude.

To further investigate how MDL-12 was influencing cAMP signaling at the dose and treatment strategy used, we evaluated phosphorylation of cyclic AMP-response element binding protein (CREB). Phosphorylation of CREB occurs downstream of cAMP and has commonly been used as a readout to assess induction of cAMP production in macrophages (Misra and Pizzo, 2005; Polumuri et al, 2021). In line with evidence in the literature that exposure to acidic pH drives an increase in intracellular cAMP in macrophages (Bohn et al, 2018), we observed that downstream phosphorylation of CREB was elevated in macrophages exposed to acidic pH compared to those in non-acidic conditions (Fig. 5E,F). Unexpectedly, pCREB levels remained elevated in BMDMs at pH 6.5 despite treatment with MDL-12, indicating that pH-driven phosphorylation of CREB was not suppressed by MDL12 treatment. In fact, pCREB was elevated in MDL-12-treated BMDMS at pH 7.4, suggesting that MDL-12 treatment alone induced phosphorylation of CREB. This is particularly surprising considering that amplitude was not altered in MDL-12-treated macrophages at neutral pH 7.4 despite elevated pCREB. This suggests that some elements of the cAMP signaling pathway, such as pCREB, may be divorced from the pH-induced changes in rhythms. Collectively, our data indicate that while the cAMP signaling pathway is induced under acidic conditions, pH-induced changes in rhythms may not be attributable to cAMP signaling alone, as MDL-12 treatment suppressed pH-induced changes in amplitude of rhythms, but not period or damping, without suppressing signaling through the cAMP pathway.

## There is evidence of circadian disorder in the tumor-associated macrophage population

As we have observed that acidic pH at levels commonly observed in the TME can alter circadian rhythms in macrophages in vitro and ex vivo, we next sought to investigate whether circadian rhythms can be altered in the TME in vivo. Using publicly available data, we analyzed gene expression of tumor-associated macrophages (TAMs) isolated from LLC (Lewis Lung carcinoma) tumors in mice (Geeraerts et al, 2021a, b). As a positive control for circadian clock disruption, we used data from BMAL1 KO peritoneal macrophages (Kitchen et al, 2019; Kitchen et al, 2020). BMAL1 KO macrophages have a genetic disruption of the circadian clock due to the loss of Bmal1, the central clock gene. As a result, circadian rhythms of BMAL1 KO macrophages are disrupted, lacking rhythmicity and downstream circadian regulation of macrophage function (see Fig. 10A,B) (Curtis et al, 2015; Gibbs et al, 2012). In line with previous observations, TAMs had elevated expression of *Arg1* relative to WT and BMAL1 KO peritoneal macrophages (Fig. 6A). Expression of *Crem*, which encodes *Icer*, was also elevated in TAMs, indicating that these TAMs were exposed to acidic conditions within the TME (Fig. 6A) (Bohn et al, 2018).

To understand the status of the circadian clock in TAMs, we performed clock correlation distance (CCD) analysis. This analysis has previously been used to assess functionality of the circadian clock in whole tumor and in normal tissue (Shilts et al, 2018). As the circadian clock is comprised of a series of transcription/translation feedback loops, gene expression is highly organized in a functional, intact clock, with core clock genes existing in predictable and ordered levels relative to each other irrespective of the time of day. In a synchronized population of cells, this ordered relationship is maintained at the population level, which can be visualized in a heatmap. CCD is designed to compare circadian clock gene co-expression patterns between different tissues and cell types. To accomplish this, CCD was built using datasets from multiple different healthy tissues from mouse and human to be a universal tool to compare circadian rhythms. Each sample is compared to a reference control built from these multiple tissues (Fig. 6B) (Shilts et al, 2018). To validate the use of this analysis for assessing circadian disorder in macrophages, we performed CCD analysis using publicly available RNA-sequencing data from bone marrow-derived macrophages (Collins et al, 2020; Collins et al, 2021) and wild-type peritoneal macrophages (Kitchen et al, 2019, 2020) as healthy controls for functional rhythms in synchronized cell populations, and BMAL1 KO peritoneal macrophages as a positive control for circadian disorder (Kitchen et al, 2019, 2020). We found that gene co-expression of clock genes was ordered in wild-type macrophages with functional clocks and intact circadian rhythms (Fig. 6C). In contrast, clock gene co-expression was disordered in BMAL1 KO macrophages with a genetic disruption of the circadian clock, leading to disruption of circadian rhythms (Fig. 6C,D). This indicates that CCD analysis can be used to measure circadian disorder in a macrophage population. To assess the status of the circadian clock in tumor-associated macrophages, we next performed CCD analysis using RNA-sequencing data of TAMs derived from LLC tumors (Geeraerts et al, 2021a, b). Clock correlation distance analysis revealed

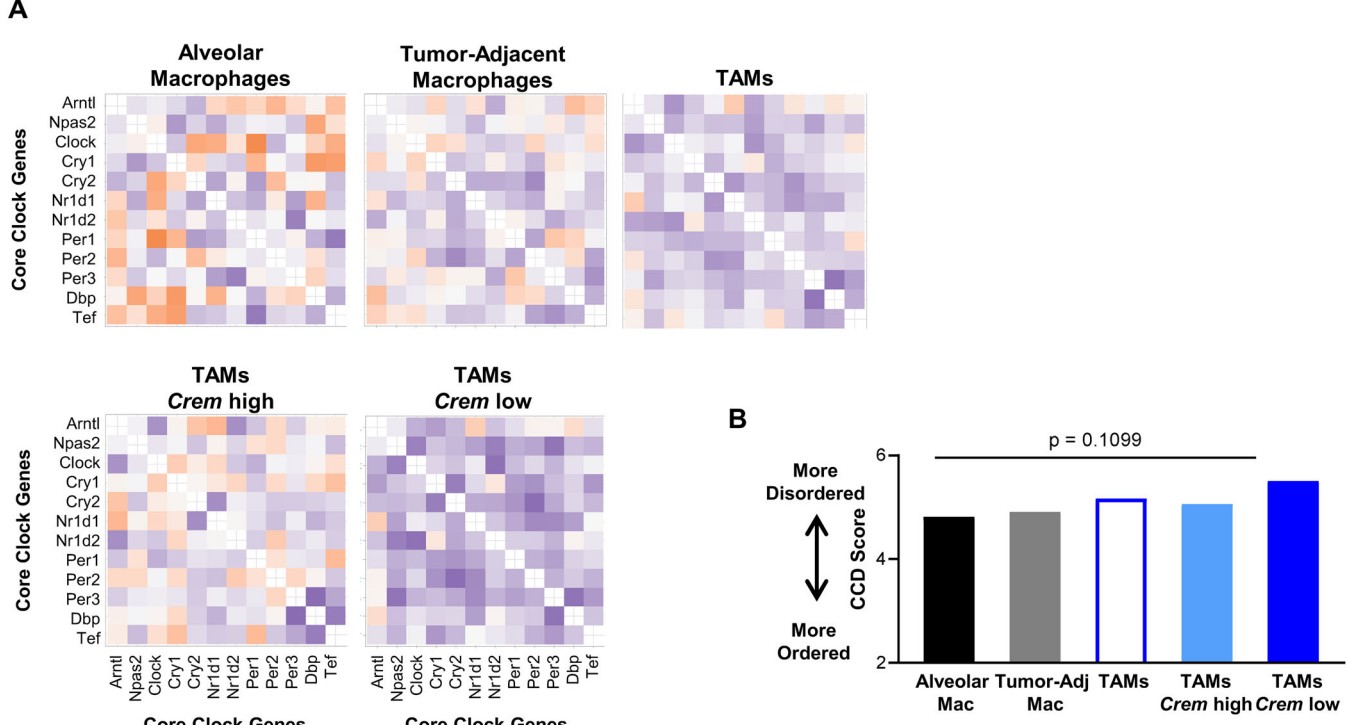

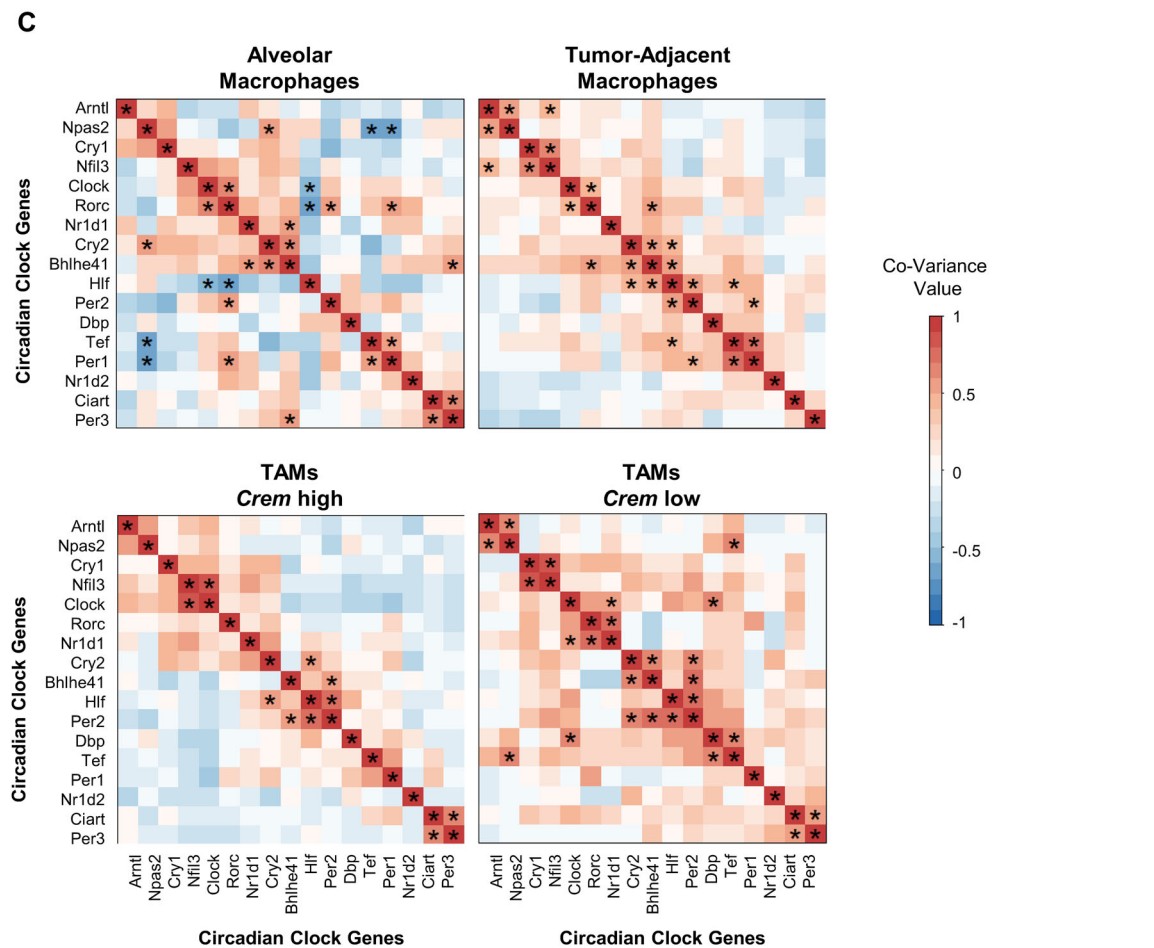

**Figure 7. Clock correlation distance (CCD) and weighted gene co-expression network analysis (WGCNA) provide evidence of circadian disorder in human tumor-associated macrophages.**

(A) Clock correlation distance (CCD) analysis was performed using RNAseq datasets of macrophages from tumor (TAMs, $n = 40$ patients), tumor-adjacent tissue from NSCLC patients ($n = 34$ patients), and alveolar macrophages from healthy donors ($n = 24$ donors) (see Methods for information on these samples). TAM samples were subset by median *Crem* expression into *Crem* high TAM samples (TAMs *Crem* high, $n = 20$ patients) and *Crem* low TAM samples (TAMs *Crem* low, $n = 20$ patients). (B) Statistical analysis to compare CCD scores was performed by calculating delta CCD, with $p < 0.05$ being deemed significantly different from the control group. (C) Weighted gene co-expression network analysis (WGCNA) was performed on the indicated circadian clock genes using data from (A, B). Data information: For (B), $p$ value shown is by delta CCD analysis compared to control group. For (C), $*p < 0.01$ by WGCNA for significant covariance by Pearson correlation, and blank squares are not significant.

that, similar to the BMAL1 KO peritoneal macrophages, the co-expression relationship between the core circadian clock genes in TAMs was significantly more disordered than that of WT peritoneal macrophages (Fig. 6C,D). Weighted gene co-expression network analysis (WGCNA) has been used as an alternate approach to measure the covariance between clock genes and thus assess bi-directional correlations among the core clock gene network in healthy tissue and tumor samples (Chun et al, 2022). In line with the circadian disorder observed by CCD, while many bi-directional correlations among the core clock gene network were significant and apparent in wild type peritoneal macrophages, these relationships were altered or abolished within BMAL1 KO peritoneal macrophages and TAM samples, and in some cases replaced by new relationships (Fig. 6E). This indicates that there is population-level disorder in the circadian rhythms of tumor-associated macrophages in murine lung cancer.

We next assessed the status of the circadian clock in human TAMs from NSCLC patients. We performed CCD with publicly available RNA-seq data of tumor-adjacent macrophages and tumor-associated macrophages from NSCLC patients, using alveolar macrophages from healthy donors as a control (Garrido-Martin et al, 2020a, b; Shaykhiev et al, 2009a, b). To assess the contribution of the acidic TME to circadian disorder, we subset TAM NSCLC patient samples into groups (*Crem* high TAMs and *Crem* low TAMs) based on median *Crem* expression. Notably, in macrophages from human NSCLC there was a trend toward disorder in *Crem* low but not *Crem* high TAM samples (Fig. 7A,B). In addition, the covariance among core clock genes observed in alveolar macrophages from healthy donors was absent within *Crem* low and *Crem* high TAM samples (Fig. 7C). In all, these data indicate that there is population-level disorder in the circadian rhythms of tumor-associated macrophages in humans and mice, suggesting that circadian rhythms are indeed altered in macrophages within the TME.

## Heterogeneity of circadian rhythms within a population can underlie circadian disorder as measured by CCD

Circadian disorder assessed by CCD has previously been used to infer disruption of circadian rhythms (Pariollaud et al, 2022). Indeed, we observed that genetic disruption of circadian rhythms by BMAL1 KO resulted in a disordered clock, as observed in peritoneal macrophages (Fig. 6B). However, since CCD is a population-level analysis, heterogeneity of rhythms, as observed in a desynchronous cell population, rather than disruption of rhythms, may also underlie the circadian disorder observed by CCD. Heterogeneity in macrophage phenotype, exposure to acidic pH, and lactate are all factors present in the TME and relevant to tumor-associated macrophages. We observed that each of these factors can alter circadian rhythms in macrophages, both alone and

in combination with each other. Furthermore, we observed a trend toward circadian disorder in *Crem* low TAM samples but not *Crem* high TAM samples. Thus, we sought to understand if heterogeneity in macrophage rhythms could be contributing to the disorder in clock gene co-expression and poor CCD score indicative of population-level disorder in TAM rhythms.

To address this, we examined if differences in rhythms of macrophages within a population might contribute to population-level disorder as measured by CCD. To this end, we used publicly available data of peritoneal macrophages taken at different times of day in 4-h intervals across 2 days (Keller et al, 2009; Keller et al, 2010). We then constructed four different sample groups in which samples were pooled according to time of day of harvest. As a control for a synchronized cell population with homogenous rhythms, samples taken at the same time of day were pooled. We then modeled a progressively desynchronized population with increased differences in phase of rhythms by pooling samples that were taken 4 h apart, 8 h apart, or 12 h apart (Fig. 8A). CCD was performed on these four populations (Fig. 8B). CCD score worsened as populations became increasingly desynchronized, with the 12 h desynchronized population having a significantly worse CCD score than synchronized, homogenous macrophage population (Fig. 8C). This indicates that as circadian rhythms of individual macrophages within a population become more different from each other, circadian disorder increases at the population-level. This is further supported by WGCNA, which revealed that the significant covariance of circadian clock genes in the synchronized population was progressively altered and lost as the population is increasing desynchronized to 12 h (Fig. EV5). The results of these analyses suggest that heterogeneity in rhythms, as observed with desynchrony, may underlie population-level disorder of the circadian clock as measured by CCD.

## Tumor-associated macrophages exhibit heterogeneity in circadian clock gene expression

Our findings suggested that heterogeneity of the circadian clock may lead to disorder in bulk macrophage populations, but did not reveal if specific gene expression changes exist in tumor-associated macrophages at the single-cell level. To determine whether heterogeneity exists within the expression of circadian clock genes of the tumor-associated macrophage population, we analyzed publicly available single-cell RNA-sequencing data of macrophages isolated from B16-F10 tumors (Wang et al, 2024a, b). To capture the heterogeneity of macrophage subsets within the TAM population, we performed unbiased clustering (Fig. 9A). We then performed differential gene expression to determine if circadian clock genes were differentially

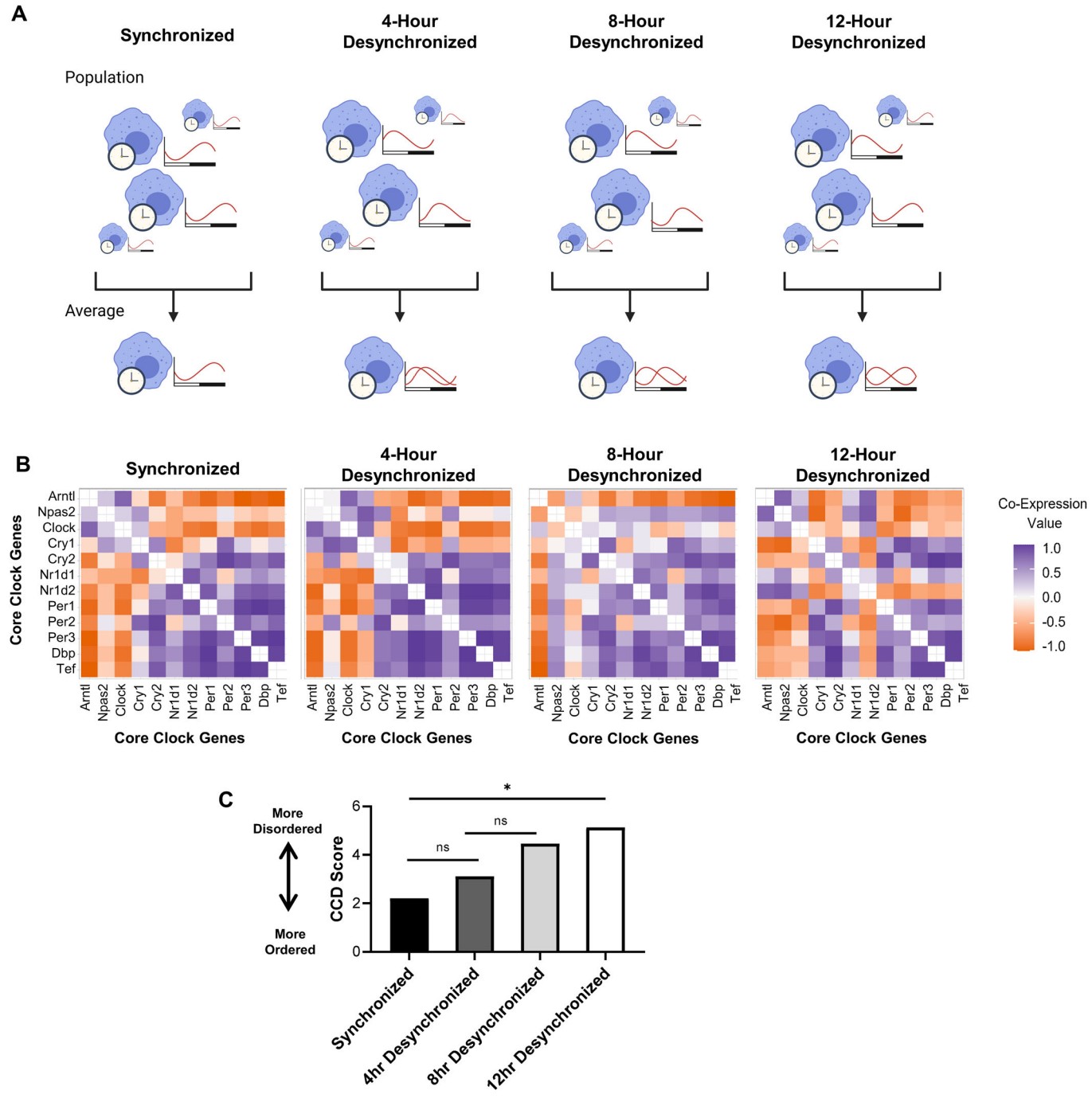

**Figure 8. Heterogeneity in circadian rhythms of cells within a population can lead to circadian disorder observed by CCD.**

(A–C) Increasingly desynchronized populations were modeled using an RNAseq data set of WT peritoneal macrophages ($n = 12$ biological replicates) taken at 4-h intervals across two days (see Methods). (A) A schematic of the populations used in experimental design. (B, C) (B) Clock correlation distance (CCD) analysis was performed and (C) statistical analysis to compare CCD scores was performed by calculating delta CCD. Data information: *$p < 0.05$ by delta CCD analysis compared to control group. Exact $p$ values: (C) 0.016442 (Synchronized vs 12 h Desynchronized).

expressed within the TAM subpopulations. The circadian clock genes *Bhlhe40* (DEC1), *Bhlhe41* (DEC2), *Nfil3* (E4BP4), *Rora* (RORα), *Dbp* (DBP), and *Nr1d2* (REV-ERBβ) were significantly (adj.p < 0.005) differentially expressed between TAM clusters (Fig. 9B; Appendix Table S1A). This indicates that there is heterogeneity in expression of circadian clock genes within the TAM population.

We next sought to determine whether differences in circadian clock gene expression between TAM subpopulations were associated with

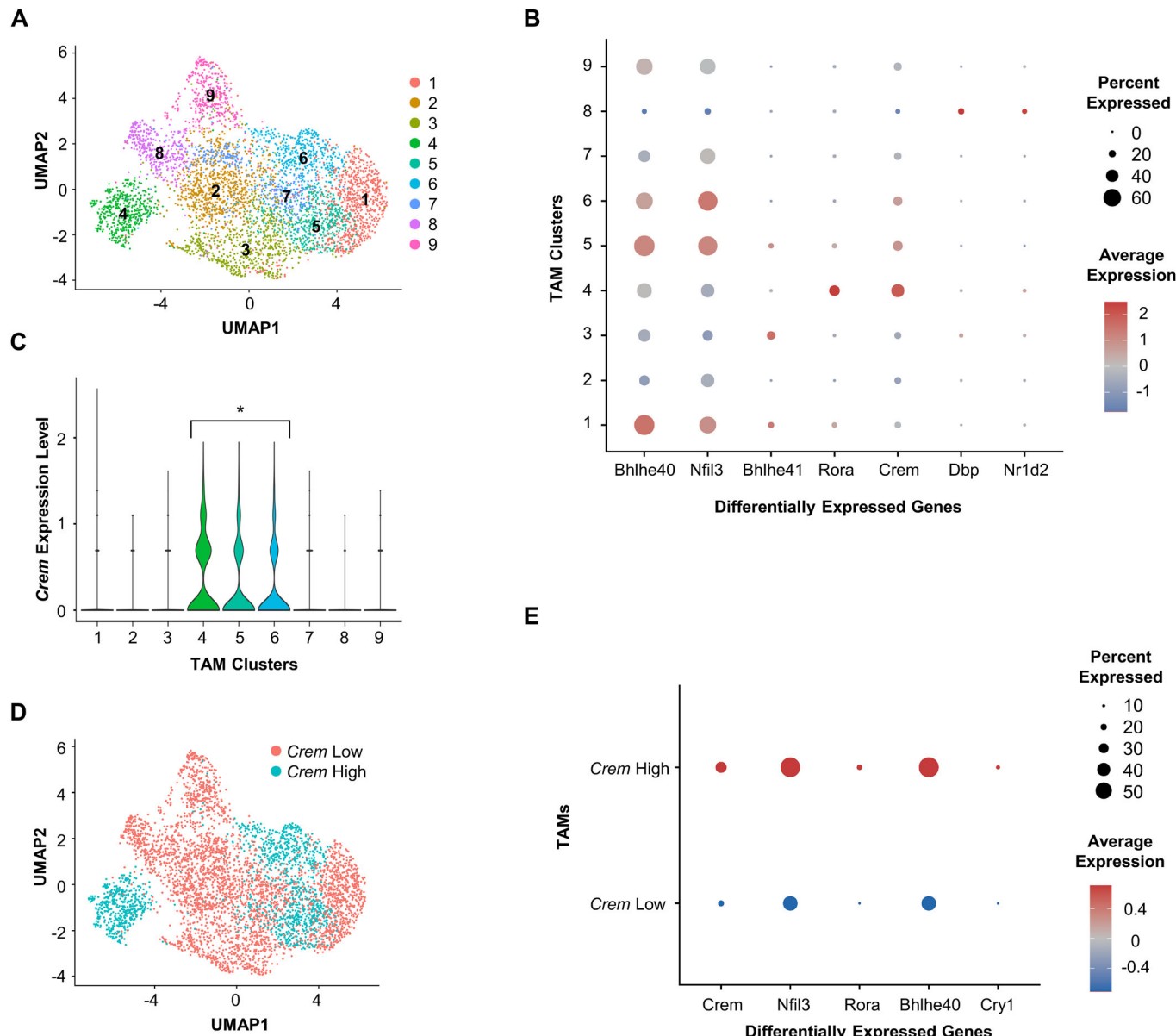

**Figure 9. There is heterogeneity in expression of circadian clock genes within the tumor-associated macrophage population.**

We analyzed a single-cell RNAseq dataset of tumor-associated macrophages (see Methods for information on these samples). (**A**) Unbiased clustering was performed to identify TAM subpopulations. (**B**) Differential gene expression analysis was performed on these TAM clusters, and expression of significantly different circadian genes was plotted along with *Crem*. (**C**) Crem expression of macrophages in TAM clusters was measured. (**D**) TAMs were subset by *Crem* expression into *Crem* high TAMs and *Crem* low TAMs, and these groups were overlaid on the UMAP plot shown in (**A**). (**E**) Differential gene expression analysis was performed on *Crem* high vs *Crem* low, and expression of significantly different circadian genes was plotted. Data information: All differentially expressed genes or groups of clusters in (**B**), (**C**), and (**E**) are adj.p < 0.005 by limma implementation of the Wilcoxon Rank-Sum test. Exact *P* value: (**C**) 2.71 E-27 (*Crem*-low clusters vs *Crem*-high clusters). Exact *P* values for (**B**, **E**) are available in Appendix Fig. S4.

exposure to acidic pH in the TME. To this end, we first assessed *Crem* expression in the TAM subpopulations that were identified by unbiased clustering. *Crem* expression was significantly higher in TAM clusters 4, 5, and 6 compared to TAM clusters 1–3 and 7–9 (Fig. 9C). Clusters were subset based on *Crem* expression into *Crem* high (clusters 4–6) and *Crem* low (clusters 1–3, 7–9) (Fig. 9D), and differential gene expression analysis was performed. The circadian clock genes *Nfil3, Rora, Bhlhe40*, and *Cry1* (CRY1) were significantly (adj.p < 0.005) differentially

expressed between *Crem* high and *Crem* low TAMs (Fig. 9E; Appendix Table S1B). This suggests that acidity within the TME is associated with heterogeneity in expression of circadian clock genes within the TAM population. Interestingly, expression of circadian clock genes varied between clusters designated as *Crem* high or *Crem* low (Fig. 9B); for instance, *Nfil3* was more highly expressed in cluster 1 than cluster 3, both of which had low *Crem* expression. This indicates that there is diversity in circadian clock gene expression within the *Crem* high and

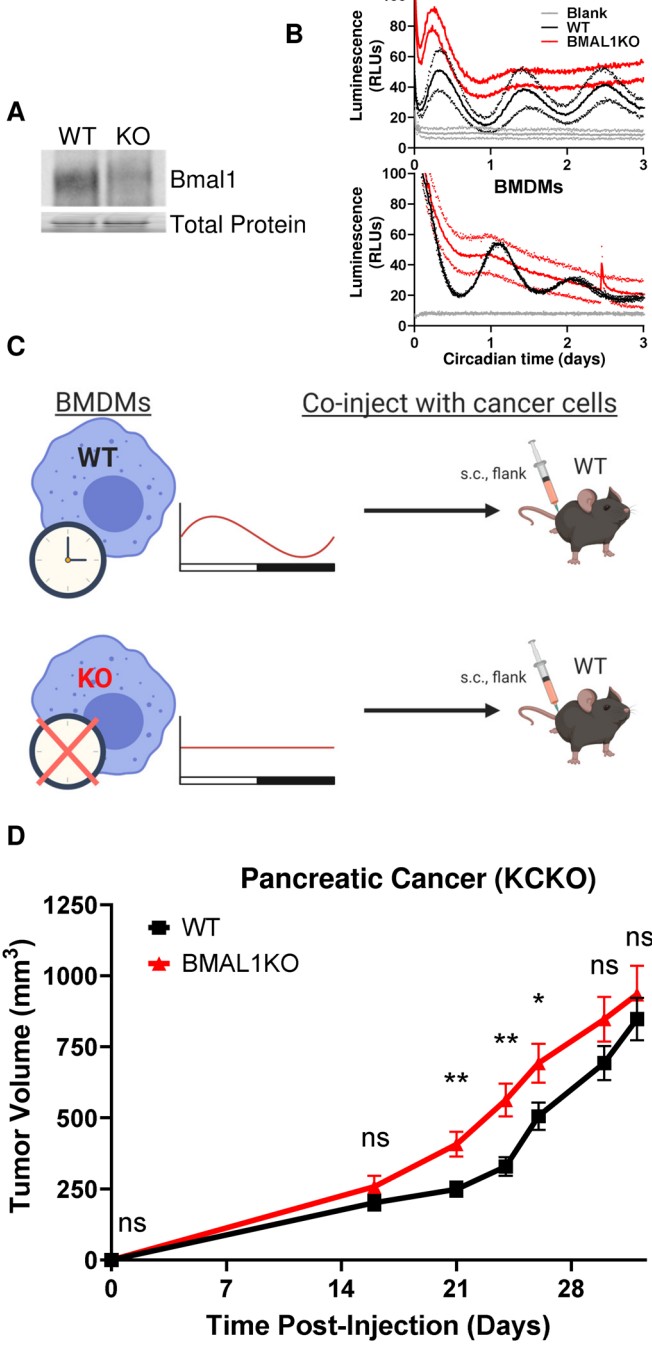

**Figure 10. A functional circadian clock in macrophages can influence tumor growth in a murine model of PDAC.**

(A) Levels of Bmal1 in bone marrow-derived macrophages (BMDMs) from WT or BMAL1 KO mice were assessed by immunoblot. (B) To confirm functional disruption of the circadian clock, peritoneal macrophages or BMDMs were obtained from WT or BMAL1 KO mice expressing PER2-Luc and cultured in vitro with D-luciferin. Luciferase activity was monitored in real time by LumiCycle, $n = 2$ biological replicates. (C, D) (C) Illustration and (D) results of in vivo tumor growth experiment. Bone marrow-derived macrophages (BMDMs) obtained from WT or BMAL1 KO mice were subcutaneously co-injected with KCKO cells into the flank of WT mice. Tumor growth was measured by caliper, $n = 20$ individual mice, 10 male and 10 female. Data information: For (A, B), experiment was replicated twice, and for (B), shown are individual points and mean. For (D), shown is the mean and SEM. Statistical significance determined at each time point by multiple Mann–Whitney tests with two-stage step-up (Benjamini, Krieger, and Yekutieli) correction for multiple testing; *$p < 0.05$; **$p < 0.005$; ***$p < 0.0005$ with $q < 0.05$ for all tests; ns: not significant. Experiment was replicated twice. Exact $p$ values: (D) 0.00321 (16 days), 0.000696 (24 days), 0.016898 (26 days).

murine model of PDAC (Besmer et al, 2011; Kidiyoor et al, 2014). To this end, we used a genetic disruption of the circadian clock in macrophages by deletion of BMAL1 (Fig. 10A,B). Myeloid-specific genetic mouse models are not macrophage-specific, so co-injection experiments are commonly used to determine macrophage-specific roles (Alexander et al, 2020; Colegio et al, 2014; Lee et al, 2018; Mills et al, 2019). Thus, we co-injected BMDMs from WT or BMAL1 KO mice along with KCKO cells into WT mice, and tumor growth was measured (Fig. 10C). We saw a significant increase in the growth of tumors co-injected with BMAL1 KO macrophages compared to those co-injected with WT macrophages (Fig. 10D). These results suggest intact circadian rhythms of macrophages can restrain tumor growth, in agreement with similar published findings in a murine model of melanoma (Alexander et al, 2020).

## Discussion

Macrophages experience altered environmental conditions within the tumor microenvironment, but how these may affect macrophage circadian rhythms remains unclear. Here we assessed whether circadian rhythms are altered in macrophages within the TME. To this end, we investigated whether conditions commonly associated with the tumor microenvironment could influence circadian rhythms in macrophages. As TAMs are phenotypically heterogenous, we first assessed circadian rhythms in macrophages polarized toward different phenotypes. We found that polarization state affected circadian rhythmicity, with pro-inflammatory macrophages exhibiting far weaker rhythms that pro-resolution macrophages (Fig. 1). We then modeled acidic conditions in the TME by exposing macrophages to pH and lactate levels similar to those found in the TME, and found that low pH in particular dramatically altered the rhythms of macrophages (Figs. 2–4, EV3 and EV4). Changes in cAMP signaling may contribute to these changes in rhythmicity, but low pH induced alterations far beyond what is observed by enhancing cAMP signaling pharmacologically (Fig. 5A,B). While the adenylyl-cyclase inhibitor MDL-12 largely rescued the changes in amplitude observed in low pH, our data suggest that a pathway other than canonical cAMP signaling may be involved in this effect (Fig. 5C–F). Finally, we assessed the status of the

*Crem* low groups, suggesting that acidic pH is not the only factor in the TME that can alter the circadian clock. Collectively, these data suggest that there is heterogeneity in the circadian clock of macrophages within the TAM population that is driven in part by acidic pH.

### Circadian rhythms of macrophages can influence tumor growth in a murine model of pancreatic cancer

We next sought to determine how circadian rhythms in tumor-associated macrophages may influence tumor growth in KCKO, a

circadian clock in tumor-associated macrophages, the potential contribution of heterogeneity in circadian rhythms to population-level rhythms, and assessed whether the circadian regulation of macrophages impacts tumor growth. Our results indicated that macrophage rhythms are disordered within tumors (Figs. 6 and 7), and that heterogeneity in rhythms within the tumor-associated macrophage population may underlie this observed circadian disorder (Figs. 8 and EV5), which was supported by our observations of heterogeneity in circadian clock gene expression within the TAM population from scRNA-sequencing data (Fig. 9). We further demonstrated that the intact macrophage circadian clock can suppress tumor growth (Fig. 10). Overall, our results for the first time demonstrate that exposure of macrophages to conditions associated with the tumor microenvironment can influence circadian rhythms, a key aspect of macrophage biology (Synopsis).

A critical question in understanding the role of circadian rhythms in macrophage biology is how different polarization states of macrophages affect their internal circadian rhythms. This is especially important considering that tumor-associated macrophages are a highly heterogeneous population. Our data indicate that compared to unstimulated macrophages, rhythms are enhanced in pro-resolution macrophages, characterized by increased amplitude and improved ability to maintain synchrony; in contrast, rhythms are suppressed in pro-inflammatory macrophages, characterized by decreased amplitude and impaired ability to maintain synchrony (Fig. 1). These agree with previously published work showing that polarizing stimuli alone and in combination with each other can alter rhythms differently in macrophages (Chen et al, 2020; Lellupitiyage Don et al, 2022). In a tumor, macrophages exist along a continuum of polarization states and phenotypes (Chevrier et al, 2017; Cuccarese et al, 2017; Huang et al, 2019; Laviron et al, 2022; Mantovani et al, 2022). Thus, while our characterizations of rhythms in in vitro-polarized macrophages provide a foundation for understanding how phenotype affects circadian rhythms of macrophages, further experiments will be needed to assess macrophages across the full spectrum of phenotypes. Indeed, alteration of rhythms may be just as highly variable and context-dependent as phenotype itself.

In addition to polarizing stimuli, tumor-associated macrophages are exposed to a variety of conditions within the tumor microenvironment that may alter their circadian rhythms. We observed that exposure to acidic pH altered rhythms in macrophages, increasing amplitude of pro-resolution macrophages but suppressing amplitude of pro-inflammatory macrophages (Fig. 2). This indicates that pH affects rhythms differently depending on phenotype, hinting at additional layers of complexity in how the environment could contribute to changes in circadian rhythms. Even further changes in rhythms were observed when macrophages were exposed to stimuli such as lactate or cancer cell supernatant in conjunction with acidic pH (Figs. 4 and EV3). These observations suggest that the combination of stimuli present in the microenvironment such as lactate and low pH, as well as various polarizing stimuli, can each contribute to modulate rhythms, resulting in highly context-dependent changes in circadian rhythms of macrophages based on the microenvironment. As macrophages are highly plastic and exquisitely capable of sensing and responding to their environment, one could reason that changes in circadian rhythms, and downstream circadian regulation, are a mechanism by which macrophages can adopt different programs to respond to their environment.

Elucidating the role of circadian rhythms in regulation of macrophage biology necessitates a better understanding of the crosstalk between phenotype, altered environmental conditions, and circadian rhythms. The differences in response to acidic pH between pro-inflammatory and pro-resolution macrophages (Figs. 1 and 2) may be due to signaling pathways downstream of polarization that directly influence circadian clock gene expression. One previously published finding that may offer mechanistic insight into how phenotype can influence circadian rhythms is the suppression of BMAL1 by LPS-inducible miR-155 (Curtis et al, 2015). It has also been observed that RORα-mediated activation of BMAL1 transcription is enhanced by PPARγ co-activation (Liu et al, 2007). In macrophages, PPARγ expression is induced upon stimulation with IL-4 and plays a key role in alternative activation of macrophages, promoting a pro-resolution macrophage phenotype, and supporting resolution of inflammation (Daniel et al, 2018; Gautier et al, 2012; Odegaard et al, 2007). In addition, lactate polarizes macrophages toward a pro-resolution phenotype similar to acidic pH (Colegio et al, 2014; Jiang et al, 2021b), and we found that exposure to lactate had different effects on circadian rhythms —and in some cases, circadian clock gene expression—than exposure to acidic pH (Fig. 4). Sensing of lactate occurs through different pathways than acid-sensing, which may contribute to the different ways in which these two stimuli modulate circadian rhythms of macrophages (Certo et al, 2022). Such observations prompt the question of whether there are yet-unidentified factors induced downstream of various polarizing stimuli that can modulate expression of circadian genes at the transcriptional and protein levels. Further work is required to understand the interplay between macrophage phenotype, altered environmental conditions, and circadian rhythms.

It was previously observed that acidic pH can disrupt circadian rhythms in cell lines (Walton et al, 2018). However, while acidic pH altered rhythms in macrophages, it did not ablate them. This suggests that the influence of acidic pH on circadian rhythms can vary between cell types. pH-induced circadian disruption was found to be driven by inhibition of mTORC1 activity in cell lines, and there was evidence to suggest that mTORC1 activity was sensitive to pH in T cells (Walton et al, 2019). Thus, the role of mTORC1 activity in mediating pH-driven changes in circadian rhythms of macrophages will be a topic of future investigation.

The mechanism through which acidic pH can modulate the circadian clock in macrophages remains unclear. Evidence in the literature suggests that acidic pH promotes a pro-resolution phenotype in macrophages by driving signaling through the cAMP pathway (Bohn et al, 2018). It has previously been shown that cAMP signaling can modulate the circadian clock (O'Neill et al, 2008). However, our data indicated that cAMP signaling was not fully sufficient to confer pH-mediated changes in circadian rhythms of macrophages (Fig. 5A,B). Treatment with MDL-12, commonly known as an inhibitor of adenylyl cyclase (Bohn et al, 2018; Guellaen et al, 1977), resulted in suppression of pH-induced changes in amplitude of circadian rhythms but did not inhibit signaling through the cAMP signaling pathway (Fig. 5C,D). While MDL-12 is commonly used as an adenylyl cyclase inhibitor, it has also been documented to have inhibitory activity toward phosphodiesterases (PDEs) and the import of calcium into the cytosol through various mechanisms (Hunt and Evans, 1980; van Rossum et al, 2000). This is of particular interest, as calcium signaling has

also been shown to be capable of modulating the circadian clock (O'Neill and Reddy, 2012). Furthermore, while acid-sensing through GPCRs has been the most well-characterized pathway in macrophages, there remain additional ways in which acidic pH can be sensed by macrophages such as acid-sensing ion channels (Ni et al, 2018; Selezneva et al, 2022). Further work is required to understand the signaling pathways through which pH can influence macrophage phenotype and circadian rhythms.

We observed that acidic pH appears to enhance circadian rhythms of unstimulated and pro-resolution macrophages, and we and others have shown evidence that macrophages are exposed to an acidic environment within the TME (Bohn et al, 2018; El-Kenawi et al, 2019). Interestingly, analysis of TAMs by clock correlation distance (CCD) presents evidence that rhythms are disordered in bulk TAMs compared to other macrophage populations (Fig. 6). CCD is one of the most practical tools currently available to assess circadian rhythms due to its ability to assess rhythms independent of time of day and without the need for a circadian time series, which is often not available in publicly available data from mice and humans (Shilts et al, 2018). However, CCD is limited in that it is a measure of population-level circadian rhythms. Our data indicate that heterogeneity of circadian rhythms within a given population can underlie circadian disorder observed by CCD (Figs. 8 and EV5). Indeed, we observed differences in the circadian clock of *Crem* low human TAM samples compared to *Crem* high human TAM samples, suggesting that acidic pH influences circadian disorder in TAMs (Fig. 7). Interestingly, *Crem* low TAM samples exhibited a trend toward disorder while *Crem* high TAM samples did not. This is of particular interest, as we have observed that acidic pH can enhance circadian rhythms in macrophages, raising the question of whether acidic pH promotes or protects against circadian disorder. We have shown that various stimuli can alter rhythms of macrophages in a complex and contributing manner, including polarizing stimuli, acidic pH, and lactate. TGFβ is produced by a variety of cells within the TME, and was recently identified as a signal that can modulate circadian rhythms (Finger et al, 2021; Massagué, 2008). In addition, when we exposed macrophages to cancer cell-conditioned media, rhythms were modulated in a manner distinct from acidic pH or lactate, with these changes in rhythms partially reversed by neutralization of the cancer cell-conditioned media pH (Fig. EV3). It is conceivable that, in addition to acidic pH, other stimuli in the TME are influencing circadian rhythms to drive population-level disorder that we observed by CCD.

Supporting the notion that population-level disorder may exist in TAMs, we used scRNA-sequencing data and found evidence of heterogeneity between the expression of circadian clock genes in different TAM subpopulations (Fig. 9A,B). Phenotypic heterogeneity of TAMs in various types of cancer has previously been shown (Chevrier et al, 2017; Lavin et al, 2017; Laviron et al, 2022; Storrs et al, 2023), and we have identified distinct TAM subpopulations by unbiased clustering (Fig. 9A). Within those TAM subpopulations, we identified differential expression of circadian clock genes encoding transcription factors that bind to different consensus sequences (Fig. 9B): DEC1 and DEC2 bind to E-boxes, NFIL3 and DBP binds to D-boxes, and RORα and REV-ERBβ binds to retinoic acid-related orphan receptor elements (ROREs) (Kato et al, 2014; Takahashi, 2017). While little is known about regulation of macrophages by D-box elements beyond the

circadian clock, aspects of macrophage function have been shown to be subject to transcriptional regulation through RORE- and E-box-binding transcription factors (Billon et al, 2019; Pastore et al, 2016; Pello et al, 2012; Rehli et al, 2005; Sato et al, 2014). Thus, we speculate that variations in the circadian transcription factors that were identified to be differentially expressed between TAM subpopulations (Fig. 9B) may exert influence on expression of genes to drive diversity within the TAM population. Differential expression of circadian clock genes between TAM subpopulations was also associated with *Crem* expression (Fig. 9C–E), suggesting that exposure of TAMs to acidic pH within the TME can alter the circadian clock. However, there remained significant variation in expression of circadian clock genes within the *Crem* high and *Crem* low groups (Fig. 9B), suggesting that acidic pH is not the only factor in the TME that can alter the circadian clock. Together, these data implicate the TME in driving heterogeneity in TAM circadian rhythms just as it drives heterogeneity in TAM phenotype.

Interestingly, in contrast to our observations of circadian disorder in TAMs isolated from LLC tumors (Fig. 6), rhythmicity in expression of circadian genes was observed in bulk TAMs isolated from B16 tumors (Wang et al, 2024b). This suggests that circadian rhythms of TAMs are maintained differently in different types of cancer. Notably, both of these observations were at the population level. Upon separation of the B16 TAM population into subsets by unbiased clustering of single-cell RNA-sequencing data, we measured differences in expression of circadian clock genes between TAM subpopulations (Fig. 9A,B). This suggests that even within a rhythmic TAM population, there is heterogeneity in the circadian clock of TAM subpopulations.

Considering our observations that conditions associated with the TME can alter circadian rhythms in macrophages, it becomes increasingly important to understand the relevance of macrophage rhythms to their function in tumors. It has been shown that acidic pH and lactate can each drive functional polarization of macrophages toward a phenotype that promotes tumor growth, with acidic pH modulating phagocytosis and suppressing inflammatory cytokine secretion and cytotoxicity (Colegio et al, 2014; El-Kenawi et al, 2019; Jiang et al, 2021b). However, how the changes in circadian rhythms of macrophages driven by these conditions contributes to their altered function remains unknown. Current evidence suggests that circadian rhythms confer a time-of-day-dependency on macrophage function by gating the macrophage response to inflammatory stimuli based on time-of-day. As such, responses to inflammatory stimuli such as LPS or bacteria are heightened during the active phase while the inflammatory response is suppressed during the inactive phase. An important future direction will be to determine how changes in circadian rhythms of macrophages, such as those observed under acidic pH or high lactate, influences the circadian gating of their function. Data from our lab and others suggest that disruption of the macrophage-intrinsic circadian clock accelerates tumor growth, indicating that circadian regulation of macrophages is tumor-suppressive in models of PDAC (Fig. 10) and melanoma (Alexander et al, 2020). This agrees with complementary findings that behavioral disruption of circadian rhythms in mice (through chronic jetlag) disrupts tumor macrophage circadian rhythms and accelerates tumor growth (Aiello et al, 2020). It remains unclear whether this is through pro-tumorigenic functions of macrophages such as extracellular matrix remodeling or angiogenesis, through suppression of the anti-tumor immune response, or a combination of both functions. Further work will be needed to tease apart these distinctions.

Whereas much work has been done to characterize how macrophages are regulated within the TME, the impact of the TME on circadian rhythms of macrophages remained elusive. Our work uncovers a novel way in which conditions associated with the TME can influence macrophage biology through modulation of circadian rhythms. While the majority of studies investigating the circadian regulation of macrophages have been conducted studying macrophages under homeostatic conditions or in response to acute inflammation (Geiger et al, 2019; Gibbs et al, 2012; Keller et al, 2009; Kitchen et al, 2020; Nguyen et al, 2013), our work contributes to an emerging body of evidence that the tissue microenvironment can influence circadian rhythms (Finger et al, 2021). This is increasingly important when considering the role of circadian rhythms in immune responses at sites of ongoing, chronic inflammation where the microenvironment is altered, such as within tumors. In identifying factors within the TME that can modulate circadian rhythms of macrophages and uncovering evidence of circadian disorder within tumor-associated macrophages, our work lays the foundation for further studies aimed at understanding how the TME can influence the function of tumor-associated macrophages through modulation of circadian rhythms.

## Limitations of the study

Our observations of rhythms in macrophages of different phenotypes are limited by in vitro polarization models. It is important to note that while our data suggest that pro-inflammatory macrophages have suppressed rhythms and increased rate of desynchrony, it remains unclear the extent to which these findings apply to the range of pro-inflammatory macrophages found in vivo. We use IFNγ and LPS co-treatment in vitro to model a pro-inflammatory macrophage phenotype that is commonly referred to as 'M1', but under inflammatory conditions in vivo, macrophages are exposed to a variety of stimuli that result in a spectrum of phenotypes, each highly context-dependent. The same is true for 'M2'; different tissue microenvironment are different and pro-resolution macrophages exist in a spectrum. Rhythms were heavily suppressed in pro-inflammatory macrophages, which made analysis of rhythm parameters in pro-inflammatory macrophages more challenging as amplitude and signal reached limit of detection. Our observations of changes in amplitude and period in pro-inflammatory macrophages compared to unstimulated macrophages agrees with the literature, where these changes in rhythms have been observed using LumiCycle as well as by mRNA (Chen et al, 2020; Lellupitiyage Don et al, 2022). This supports the validity and reproducibility of our observations despite the challenges of observing and analyzing rhythms of pro-inflammatory macrophages.

## Methods

### Reagents and tools table

| Reagent/Resource | Reference or Source | Identifier or Catalog Number |
| --- | --- | --- |
| **Experimental Models** | | |
| LysM-cre mice on C57B6 background | Jackson Labs | Strain # 004781 |
| Bmal1^flox/flox mice on C57B6 background | Jackson Labs | Strain # 007668 |
| PER2-Luciferase mice on C57B6 background | Jackson Labs | Strain # 006852 |

| Reagent/Resource | Reference or Source | Identifier or Catalog Number |
| --- | --- | --- |
| KCKO Pancreatic cancer mouse cells from C57B6 strain P48-Cre; LSL-KRAS^G12D; Muc1KO | (Besmer et al, 2011) | Not commercially available |
| L929 mouse cell line | ATCC | CAT#CCL-1 |
| **Recombinant DNA** | | |
| N/A | | |
| **Antibodies** | | |
| Rabbit anti-p-CREB (Ser133, Ser129) | Invitrogen | CAT#44-297G |
| Goat anti-rabbit Alexa Fluor 680 | Invitrogen | CAT#A21109 |
| **Oligonucleotides and other sequence-based reagents** | | |

| qPCR primer name | Source and/or Catalog Number | Primer Sequences |
| --- | --- | --- |
| Arg1 | (Liu et al, 2017) | 5'-CTCCAAGCCAAAGTCCTTAGAG-3', 5'-AGGAGCTGTCATTAGGGACATC-3' |
| Chil3 | IDT Mm.PT.58.33370435 | 5'-AGAAGCAATCCTGAAGACACC-3', 5'-ACTGGTATAGTAGCACATCAGC-3' |
| Clec10a | IDT Mm.PT.56a.19092703 | 5'-TGACTGAGTTCCTGCCTCT-3', 5'-GACCAAGGAGAGTGCTAGAAG-3' |
| Cry1 | (Huber et al, 2016a) | 5'-GCTATGCTCCTGGAGAGAACG T-3', 5'-TGTCCCCGTGAGCATAGTGTAA-3' |
| Icer | (Bohn et al, 2018) | 5'-ATGGCTGTAACTGGAGATGAA-3', 5'-GTGGCAAAGCAGTAGTAGGA-3' |
| Il1b | (Liu et al, 2017) | 5'-TACGGACCCCAAAAGATGA-3', 5'-TGCTGCTGCGAGATTTGAAG-3' |
| Il6 | (Liu et al, 2017) | 5'-TAGTCCTTCCTACCCCAATTTCC-3', 5'-TTGGTCCTTAGCCACTCCTTC-3' |
| Mrc1 | IDT Mm.PT.58.42560062 | 5'-CAAGTTGCCGTCTGAACTGA-3', 5'-TATCTCTGTCATCCCTGTCTCT-3' |
| Nos2 | IDT Mm.PT.58.43705194 | 5'-GCTTCTGGTCGATGTCATGAG-3', 5'-TCCACCAGGAGATGTTGAAC-3' |
| Nr1d1 | IDT Mm.PT.58.17472803 | 5'-GAGCCACTAGAGCCAATGTAG-3', 5'-CCAGTTTGAATGACCGCTTTC-3' |
| Per2 | IDT Mm.PT.58.5594166 | 5'-TGAGGTAGATAGCCCAGGAG-3', 5'-GCTATGAAGCGCCTAGAATCC-3' |
| Retnla | (Liu et al, 2017) | 5'-CTGGGTTCTCCACCTCTTCA-3', 5'-TGCTGGGATGACTGCTACTG-3' |
| Tbp | IDT Mm.PT.39a.22214839 | 5'-CCAGAACTGAAAATCAACGCAG-3', 5'-TGTATCTACCGTGAATCTTGGC-3' |
| Tnfa | (Liu et al, 2017) | 5'-ACGGCATGGATCTCAAAGAC-3', 5'-AGATAGCAAATCGGCTGACG-3' |
| Vegf | (Colegio et al, 2014) | 5'-CCACGACAGAAGGAGAGCAGAAGTCC-3', 5'-CGTTACAGCAGCCTGCACAGCG-3' |
| **Chemicals, Enzymes and other reagents** | | |
| RPMI media | Corning | CAT#MT10040CV |
| RPMI powdered media | Corning | CAT#50-020-PC |
| Low endotoxin fetal bovine serum | Cytiva | CAT#SH30396.03 |
| Penicillin-Streptomycin | Gibco | CAT# 15140122 |
| Horse serum | Corning | CAT#35030CV |
| HEPES | Gibco | CAT#15630080 |
| PIPES | Sigma | CAT#P1851 |
| NaOH | EMD Millipore | CAT#SX0593-1 |
| Mouse IL-4 recombinant protein | Thermo/Peprotech | CAT#214-14 |
| Mouse IL-13 recombinant protein | Thermo/Peprotech | CAT#210-13 |
| Mouse IFNγ recombinant protein | Thermo/Peprotech | CAT#315-05 |
| Lipopolysaccharide (LPS) | Invitrogen | CAT#00497693 |
| Sodium-L-lactate | Sigma | CAT#L7022 |
| MDL-12330A | Sigma | CAT#M182 |
| 3-Isobutyl-1-methylxanthine (IBMX) | Sigma | CAT#I5879 |

| Reagent/Resource | Reference or Source | Identifier or Catalog Number |
|---|---|---|
| Forskolin | Sigma | CAT#344270 |
| Beetle D-luciferin | Promega | CAT#E1602 |
| Oligo d(T)16 | Invitrogen | CAT#N8080128 |
| PerfeCTa SYBR Green FastMix | QuantBio | CAT#95074-05K |
| M-Per lysis reagent | Thermo Scientific | CAT#78501 |
| Protease and phosphatase inhibitor cocktail | Sigma | CAT#PPC1010 |
| Phosphatase inhibitor cocktail 2 | Sigma | CAT#D9533 |
| Hoechst Nucleic Acid Stain | Thermo Scientific | CAT#62249 |
| Isoflurane (Fluriso™) | VetOne Fluriso | CAT#501017 |
| Normal Saline 0.9% | Medline | CAT#RDI30296 |
| **Software** | | |
| Ubuntu v18.04 | Microsoft Store | |
| Bioconda v23.1.0 | https://bioconda.github.io/ | |
| R v3.6.3 or v4.0.5 | https://cran.r-project.org/ | |
| R Studio v2024.04.2 | https://posit.co/ | |
| GraphPad Prism v10.3.0 | https://www.graphpad.com/ | |
| **Other** | | |
| EZNA HP Total RNA Kit | Omega Biotek | CAT#R6812-02 |
| Taqman Reverse Transcription Reagents | Invitrogen | CAT#N8080234 |
| QuantStudio 5 384-well qPCR machine | Applied Biosystems | CAT#A28140 |
| Bio-Rad DC Protein Assay Kit | Bio-Rad | CAT#5000112 |
| Bio-Rad Criterion 4–15% Criterion TGX Stain-Free 26-well gradient gel | Bio-Rad | CAT#5678095 |
| TransBlot Turbo Tranfer System | Bio-Rad | CAT#1704150 |
| Trans-Blot Turbo RTA Midi 0.2 μm Nitrocellulose Transfer Kit | Bio-Rad | Bio-Rad CAT#1704271 |

## Animals

Mice were maintained in individually ventilated cages with bedding and nesting material in a temperature-controlled, pathogen-free environment in the animal care facility at the University of Rochester. All animal protocols were approved by the University of Rochester Committee of Animal Resources (UCAR, approval #UCAR-2019-014). The University of Rochester Medical Center Animal Facility is fully accredited by AAALAC International and in compliance with state law, federal statute and NIH policy. All experiments were performed in compliance with the NIH- and University of Rochester-approved guidelines for the use and care of animals, as well as recommendations in the Guide for the Care and Use of Laboratory Animals of the National Research Council (Committee, 2011). Mice were housed on a 12:12 light dark cycle. In some cases, to ease timepoint collection, mice were housed under reverse lighting conditions in a 12:12 dark light cycle for at least 2 weeks prior to use in experiments. Mice used for experiments were between the ages of 8–14 weeks old; both male and female mice were used. Mice were euthanized humanely prior to harvesting peritoneal macrophages or bone marrow.

Previously characterized mice with a myeloid-specific deletion of BMAL1 (LysM-cre$^{+/-}$Bmal1$^{flox/flox}$; referred to as BMAL1 KO mice) (Gibbs et al, 2012) in a C57BL/6 background were generated by crossing LysM-cre mice (Clausen et al, 1999) with Bmal1$^{flox/flox}$ mice (Storch et al, 2007). These mice were further crossed with PER2-Luc mice (Yoo et al, 2004) to generate BMAL1 KO or wild-type control mice (LysM-cre$^{-/-}$Bmal1$^{flox/flox}$; referred to as WT) expressing PER2-Luc. PER2-Luc (strain #006852), LysM-cre (strain #004781), and Bmal1$^{flox/flox}$ (strain #007668) mice used for breeding to generate WT and BMAL1 KO mice were purchased from the Jackson Laboratory.

## Differentiation and culture of bone marrow-derived macrophages

Bone marrow-derived macrophages (BMDMs) were generated from bone marrow isolated from WT mice using a well-established protocol for differentiation of BMDMs over 7 days (Gonçalves and Mosser, 2015; Trouplin et al, 2013). In brief, bone marrow cells were seeded at 200,000 cells/mL on non-tissue culture treated-plates in BMDM Differentiation Media: RPMI (Corning, CAT#MT10040CV) supplemented with 20% (v/v) L929 supernatant, 10% (v/v) heat-inactivated (HI) fetal bovine serum (FBS) (Cytiva, CAT#SH30396.03), and 100 U/mL Penicillin-Streptomycin (Gibco, CAT# 15140122). Cells were grown at 37 °C in air enriched with 5% $CO_2$. On day 3, additional BMDM Differentiation Media was added to cells. On day 6 of the differentiation protocol, BMDMs were seeded at $1.2 \times 10^6$ cells/mL and left in BMDM Differentiation Media, and kept at 37 °C in air enriched with 5% $CO_2$. On day 7, BMDM Differentiation Media was removed and BMDMs were synchronized.

To synchronize BMDMs, we adapted a recently published method (Collins et al, 2021). Briefly, BMDMs were first serum starved for 24 h in serum-free media (RPMI, supplemented with 100 U/mL Penicillin-Streptomycin); BMDMs were then subjected to serum shock by replacing serum-free media with RPMI supplemented with 50% (v/V) HI horse serum (Corning, CAT#35030CV) at 37 °C in air enriched with 5% $CO_2$. At the end of this synchronization protocol, media was replaced with Atmospheric Media, which has been formulated for use at atmospheric $CO_2$ levels and enhanced pH stability by increasing buffering capacity at low pH (Walton et al, 2018): RPMI (Corning, CAT#50-020-PC), 25 mM HEPES (Gibco, CAT#15630080), 25 mM PIPES (Sigma, CAT#P1851), supplemented with 10% (v/v) HI FBS and 100 U/mL Penicillin-Streptomycin (Walton et al, 2018). Atmospheric Media was adjusted to pH 7.4, 6.8, or 6.5 with NaOH (EMD Millipore CAT#SX0593-1) and filter-sterilized.

BMDMs cultured in Atmospheric Media at pH 7.4, 6.8, or 6.5 were either left unstimulated or were polarized toward a pro-resolution ('M2') or pro-inflammatory phenotype ('M1') by addition of 10 ng/mL IL-4 (PeproTech, CAT#214-14) and 10 ng/mL IL-13 (PeproTech, CAT#210-13), or 50 ng/mL IFNγ (PeproTech, CAT#315-05) and 100 ng/mL LPS (Invitrogen, CAT#00497693), respectively. For lactate experiments, sodium-L-lactate (Sigma, CAT#L7022) or vehicle was added to Atmospheric Media for a final concentration of 25 mM or 0 mM. For interrogation of cAMP signaling pathway, BMDMs were cultured in Atmospheric Media at pH 7.4 or 6.5 with vehicle or 5, 10, or 15 μM MDL-12330A (Sigma, CAT#M182). For phenocopy experiments (Fig. 5; Appendix Fig. S4), BMDMs were not synchronized prior to the experiment. BMDMs were cultured in Atmospheric

Media at pH 7.4 or 6.5 with vehicle or 20, 40, or 80 μM IBMX (Sigma, CAT#I5879) or forskolin (Sigma, CAT#344270). For LumiCycle experiments, 100 mM D-luciferin was added to Atmospheric Media at 1:1000 for 100 μM D-luciferin (Promega, CAT#E1602). Cells cultured in Atmospheric Media were kept at 37 °C in atmospheric conditions and were either monitored over time by LumiCycle or harvested for RNA or protein at the time points indicated.

## Isolation and culture of peritoneal macrophages

Peritoneal exudate cells were harvested from mice as previously published (Ray and Dittel, 2010). To isolate peritoneal macrophages, peritoneal exudate cells were seeded at $1.2 \times 10^6$ cells/mL in RPMI/10% HI FBS supplemented with 100 U/mL Penicillin-Streptomycin and left at 37 °C for 1 h, after which non-adherent cells were rinsed off (Gonçalves and Mosser, 2015). Isolation of peritoneal macrophages using this method has been shown to yield a population that is over 90% pure macrophages (De Jesus et al, 2022; Layoun et al, 2015). Peritoneal macrophages were then cultured in Atmospheric Media at pH 7.4 or 6.5 with 100 μM D-luciferin and kept at 37 °C in atmospheric conditions.

## Quantification of circadian rhythm parameters

Using the Lumicycle Analysis program version 2.701 (Actimetrics), raw data was fitted to a linear baseline, and the baseline-subtracted data was fitted to a damped sine wave from which period and damping were calculated (Ramanathan et al, 2014). Amplitude was calculated from baseline-subtracted data by subtracting the bioluminescent values of the first peak from the first trough as previously published (Chen et al, 2020).

## Production of KCKO cancer cell supernatant

KCKO cells were seeded at 300,000 cells/mL in pH 7.4 Atmospheric Media (RPMI buffered for use at atmospheric $CO_2$ levels and enhanced buffering capacity at low pH—see "Differentiation and culture of bone marrow-derived macrophages" section) and cultured at 37 °C in atmospheric conditions for 5 days. Supernatant was then collected, pH-adjusted to pH 7.4 or 6.5, and filter-sterilized. For the experiment, bone marrow-derived macrophages were cultured in pH 7.4 or pH 6.5 KCKO supernatant, or pH 7.4 or pH 6.5 Atmospheric Media. Media was supplemented with 100 μM D-luciferin, and cells were kept at 37 °C in atmospheric conditions and monitored over time by LumiCycle.

## Quantitative PCR

Cells were lysed and RNA was isolated using the E.Z.N.A. HP Total RNA Kit (Omega BioTek, CAT#R6812-02). RNA was reverse transcribed to cDNA using the Taqman Reverse Transcription Reagents system (Invitrogen CAT#N8080234), using oligo dT for priming (Invitrogen CAT#N8080128). qPCR was performed with cDNA using PerfeCTa SYBR Green FastMix (QuantaBio, CAT#95074-05K) and with the Quant Studio 5 quantitative PCR machines (Applied Biosystems). Triplicate technical replicates were performed, outlier replicates (defined as being more than 1 Ct away from other two replicates) were discarded, and relative mRNA was normalized to *Tbp* and assessed by the ΔΔCt. See Reagents and Tools Table for primer sources and sequences.

## Immunoblot

Cells were lysed using the M-Per lysis reagent (Thermo Scientific, CAT#78501), supplemented with protease and phosphatase inhibitor cocktail (1:100; Sigma, CAT#PPC1010) and phosphatase inhibitor cocktail 2 (1:50; Sigma, CAT#P5726), with 200 μM deferoxamine (Sigma, CAT#D9533). M-Per is formulated to lyse the nucleus and solubilize nuclear and chromatin-bound proteins, allowing isolation of nuclear proteins as well as cytosolic proteins. Lysates were incubated on ice for 1 h, then centrifuged at $17,000 \times g$ to pellet out debris; supernatant was collected. Protein was quantified using the Bio-Rad DC Protein Assay Kit (Bio-Rad, CAT#5000112), and lysates of equal concentration were prepared and run by SDS-PAGE on Bio-Rad Criterion 4–15% Criterion TGX Stain-Free 26-well gradient gel (Bio-Rad, CAT#5678095). Gels were transferred using the Trans-Blot Turbo system (Bio-Rad CAT#1704150) to nitrocellulose membranes (Bio-Rad CAT#1704271).

The following primary antibody was used: rabbit anti-p-CREB (Ser133, Ser129) (Invitrogen, CAT#44-297G). The following secondary antibody was used: goat anti-rabbit Alexa Fluor 680 (Invitrogen, CAT#A21109). Of note, two different anti-CREB antibodies were tested (Cell Signaling, CAT#9197 and Invitrogen, CAT#35-0900) in combination with appropriate secondary antibodies but neither revealed bands at the correct molecular weight for CREB protein. Membranes were digitally imaged using a ChemiDoc MP (Bio-Rad) and uniformly contrasted. Total protein was imaged by Stain-Free imaging technology (Bio-Rad) and used as loading control. To visualize total protein, image of entire membrane was shrunk to match the size of pCREB.

## Survival under acidic pH

BMDMs were seeded, in triplicate, at $1.2 \times 10^6$ cells/mL in a 96-well plate. BMDMs were synchronized, then cultured in Atmospheric Media at pH 7.4, 6.8, or 6.5 containing 10 ng/mL IL-4 and 10 ng/mL IL-13, or 50 ng/mL IFNγ and 100 ng/mL LPS, or vehicle for unstimulated control. BMDMs were fixed at 1, 2, and 3 days later. BMDMs were stained with Hoechst (Thermo Scientific, CAT#62249), and plates were imaged using a Celigo S. Number of nuclei per well was enumerated using Celigo software to quantify the number of adherent BMDMs after time in culture under acidic conditions as a readout of survival.

## Tumor growth

Mice were anesthetized via inhalation of 4 vol% isoflurane (Fluriso™, VetOne, CAT#501017) in 100% oxygen at a flow rate of 4 L/min prior to injection. Following application of 70% ethanol to the site of injection, with $1 \times 10^6$ WT or BMAL1 KO macrophages and $1 \times 10^6$ KCKO cells in 100 μL sterile 0.9% normal saline (Medline, CAT#RDI30296) were subcutaneously co-injected in the flank of WT mice. In line with previously published co-injection tumor experiments, mice were injected with macrophages at a 1:1 ratio (Alexander et al, 2020; Colegio et al, 2014). Tumor growth was measured by caliper, and volume was calculated by the modified ellipsoidal formula: tumor volume = $0.5 \times (\text{length} \times \text{width}^2)$ (Tomayko and Reynolds, 1989). When tumors did not engraft and grow, these animals were excluded. Mice were euthanized when there was ulcer formation or when tumor size reached a diameter of 20 mm.

## Processing and analysis of publicly available bulk gene expression data

FASTQ files from a previously published analysis of peritoneal macrophages cultured under neutral pH 7.4 or acidic pH 6.8 conditions were downloaded from NCBI GEO (Data ref: accession #GSE164697) (Jiang et al, 2021b, a). FASTQ files from a previously published analysis of peritoneal macrophages from WT or BMAL1 KO mice were downloaded from EMBL- European Bioinformatics Institute Array Express (Data ref: accession #E-MTAB-8411) (Kitchen et al, 2019, 2020). FASTQ files from a previously published analysis of bone marrow-derived macrophages were downloaded from NCBI GEO (Data ref: accession #GSE157878) (Collins et al, 2020, 2021). FASTQ files from a previously published analysis of macrophages from tumor and tumor-adjacent tissue from NSCLC patients were downloaded from NCBI GEO (Data ref: accession #GSE116946) (Garrido-Martin et al, 2020a, b). FASTQ files from a previous published study of tumor-associated macrophages were downloaded from NCBI GEO (Data ref: accession #GSE188549) (Geeraerts et al, 2021a, b). Where applicable, multiple FASTQ files of the same run were concatenated before processing and mapping. CEL files from a previously published microarray time series analysis of peritoneal macrophages from WT mice were downloaded from NCBI GEO (Data ref: accession #GSE25585) (Keller et al, 2009, 2010). CEL files from a previously published microarray analysis of alveolar macrophages from healthy human donors were downloaded from NCBI GEO (Data ref: accession #GSE13896) (Shaykhiev et al, 2009a, b).

All FASTQ files were processed with FASTP using default parameters to trim adapters and remove reads that were low quality or too short (Chen et al, 2018). Cleaned FASTQ files from mouse data were mapped to transcripts using Salmon 1.4.0 in mapping-based mode using a decoy-aware transcription built from the Gencode M27 GRCm39 primary assembly mouse genome and M27 mouse transcriptome, and from human data were mapped to transcripts using Salmon v1.4.0 in mapping-based mode using a decoy-aware transcription built from the Gencode v43 GRCh38 primary assembly human genome and v43 human transcriptome (Patro et al, 2017). Single-end mapping was used for GSE188549, GSE116946 samples and paired-end mapping was used for E-MTAB-8411 and GSE164697. All transcripts were then collapsed to gene-level using Tximport v1.14.2 with the Gencode M27 transcriptome for mouse and the v43 transcriptome for human, and genes were annotated with symbols using the Ensembl GRCm39.104 transcriptome annotations for mouse and Ensembl GRCh38.111 transcriptome annotations for human (Soneson et al, 2015). Transcripts per million (TPM) outputted from Tximport were used for downstream analyses. Microarray data was imported and analyzed from CEL files using the packages affy v1.68.0 and Limma v3.46.0, and genes were annotated with symbols using the University of Michigan Brain Array Custom CDF v25.0 for the Mouse Gene 1.0 ST Array or Human Gene 133 plus 2 Array (Dai et al, 2005; Ritchie et al, 2015).

Clock correlation distance (CCD) analysis was performed as previously described (Shilts et al, 2018), using v1.0.4. Briefly, the default universal 12-gene molecular circadian clock reference correlation was used. Genes outside this 12-gene reference that were of zero variance across at least one of the sample groups were discarded prior to analysis. For delta CCD, which directly compares between each group, the sample group with the lowest CCD score (corresponding to the most ordered clock) was set as the control group, and $p < 0.05$ was deemed significantly different from the control group. DeltaCCD determines significance by permuting the sample labels 1000 times, keeping the reference sample group the same, and uses the method of Phipson and Smith to determine an exact one-sided $p$ value.

WCGNA v1.72-5 analysis to determine significant covariance between pairs of clock genes was performed as previously described, using an expanded list of core circadian clock genes (Geeraerts et al, 2021a). Results were calculated from Pearson correlations of log-transformed TPM values. For data presented in Fig. 6, the gene-specific mean for TAMs within each group in the original publication (MHCII-high and MHCII-low, (Geeraerts et al, 2021a, b)) were subtracted out prior to analysis. Significance was determined by $p < 0.05$ (Fig. 6) or $p < 0.01$ (Figs. 7 and EV5). Note that WGCNA does not output exact $P$ values.

## Processing and analysis of publicly available single-cell gene expression data

A merged Seurat Object of single-cell RNA-sequencing data of immune cells infiltrating B16-F10 melanoma was downloaded from NCBI GEO (Data ref: accession #GSE260641) (Wang et al, 2024a, b). Data were analyzed using Seurat 5.0.3 (Butler et al, 2018). Macrophages, which were previously annotated, were isolated, gene expression normalized, and the following factors regressed out: cell cycle state, mitochondrial content, and circadian timepoint. 9 clusters were identified with the FindNeighbors and FindClusters commands. UMAP was used to visualize macrophage clusters (Becht et al, 2019). *Crem* high groups were identified from violin plot and confirmed with differential expression compared to *Crem* low groups. All differential expression was performed using the FindMarkers command only considering genes expressed within at least 10% of the comparison populations and with a minimum log fold change of 0.25 between populations (min.pct = 0.1 and logfc.theshold = 0.25). Significantly differentially expressed genes were determined by adjusted p(adj.p) < 0.05 by the limma implementation of the Wilcoxon Rank-Sum test (test.use = "wilcox_limma") (Ritchie et al, 2015; Smyth, 2005). A differentially expressed genes shown are p(adj.p) < 0.005.

## Software

Analyses in the Windows BASH Shell were run using Ubuntu version 18.04 and Bioconda version 23.1.0. Analyses in R were run using R versions 3.6.3 or 4.0.5 in R Studio version 2024.04.2. Analyses in GraphPad Prism were run using GraphPad Prism 10.3.0.

## Power calculations and blinding

Sample sizes were determined from power calculations using data from small pilot experiments. Researchers were not blinded to conditions, but were blinded to results while taking measurements. For tumor assays, tumor measurements were not graphed and analyzed until after completion of experiment.

## Graphics

The following Figures were created using Biorender.com: Fig. 8A, Fig. 10C, Fig. EV1A, and the Summary Figure.

# Data availability

All raw data, analyses, and code used for analyses are available at FigShare at the following link: https://rochester.figshare.com/projects/Source_data_for_Circadian_rhythms_of_macrophages_are_altered_by_the_acidic_pH_of_the_tumor_microenvironment/210625.

The source data of this paper are collected in the following database record: biostudies:S-SCDT-10_1038-S44319-024-00288-2.

# Peer review information

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

## Acknowledgements

We would like to thank Dr. Jim Miller and Dr. Scott Gerber (Department of Microbiology and Immunology, University of Rochester) and Dr. Paul Brookes (Department of Anesthesiology and Perioperative Medicine, University of Rochester) for their intellectual contributions. We also thank Dr. Christoph Scheiermann (University of Geneva, Geneva, Switzerland) for assistance with scRNA-seq data, and the University of Rochester Genomics Research Center (GRC) with assistance with data analysis and preparation. The project described was supported by Grant Number T32AI007285 (to AMKC) from the National Institute of Allergy and Infectious Diseases of the National Institutes of Health (NIH), Grant Number T32GM135134 (to AMKC) from the National Institute of General Medical Sciences of the National Institutes of Health, Grant Numbers R00CA204593 and R01CA282225 (to BJA) from the National Cancer Institute of the National Institutes of Health, the University of Rochester University Research Award (to BJA), and the Wilmot Predoctoral Cancer Research Fellowship (to AMKC) from the Wilmot Cancer Institute at the University of Rochester. The contents of this paper are solely the responsibility of the Authors and do not necessarily represent the official views of the NIH.

## Author contributions

**Amelia M Knudsen-Clark**: Conceptualization; Data curation; Software; Formal analysis; Funding acquisition; Validation; Investigation; Visualization; Methodology; Writing—original draft; Writing—review and editing. **Daniel Mwangi**: Software; Investigation; Writing—review and editing. **Juliana Cazarin**: Investigation; Writing—review and editing. **Kristina Morris**: Software; Formal analysis; Methodology. **Cameron Baker**: Data curation; Software; Formal analysis; Visualization; Writing—review and editing. **Lauren M Hablitz**: Formal analysis; Writing—review and editing. **Matthew N McCall**: Data curation; Software; Formal analysis. **Minsoo Kim**: Methodology; Writing—review and editing. **Brian J Altman**: Conceptualization; Resources; Data curation; Software; Formal analysis; Supervision; Funding acquisition; Visualization; Methodology; Writing—review and editing.

Source data underlying figure panels in this paper may have individual authorship assigned. Where available, figure panel/source data authorship is listed in the following database record: biostudies:S-SCDT-10_1038-S44319-024-00288-2.

## Disclosure and competing interests statement

BJA is a member of *The EMBO Journal* Catalysts program. The authors declare no competing interests.

# Expanded View Figures

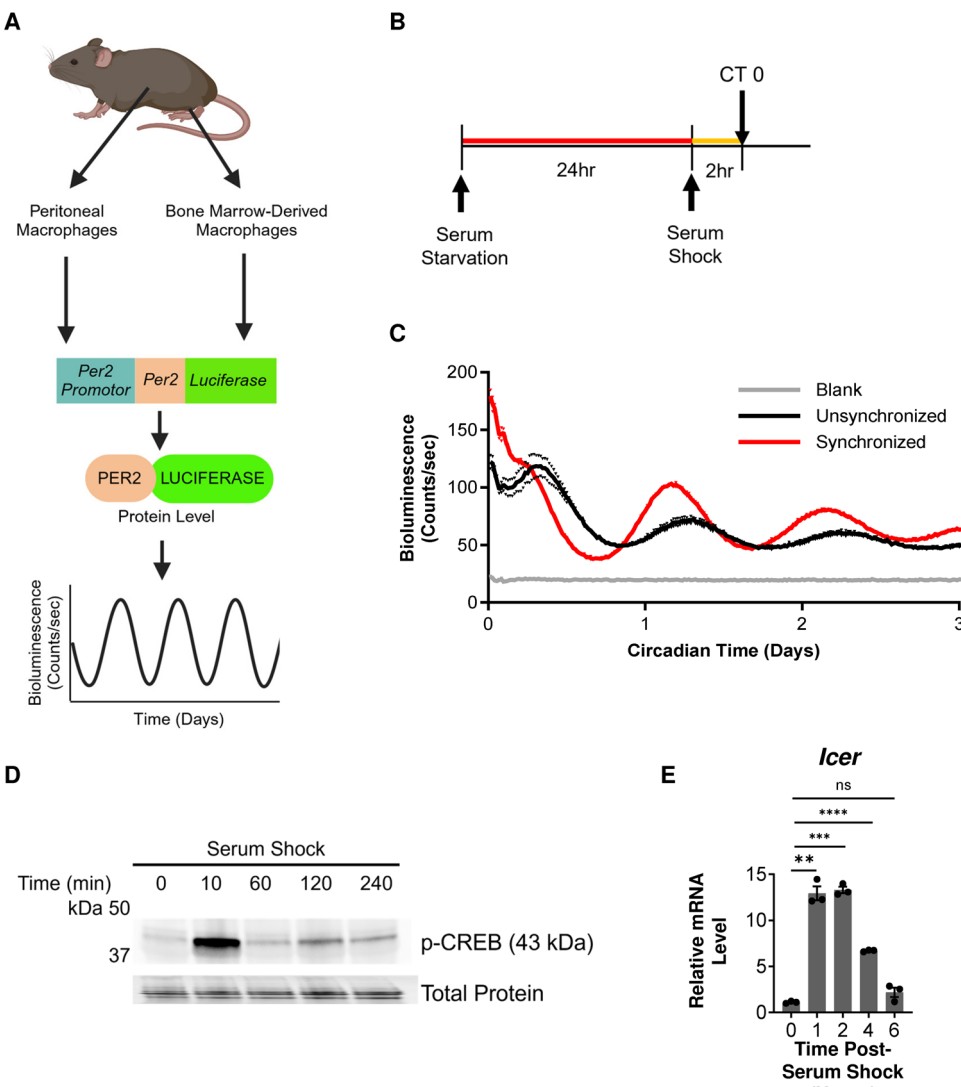

**Figure EV1. The PER2-Luciferase reporter system enables real-time monitoring of circadian rhythms of macrophages.**

(A) A schematic of the Per2-Luciferase (Per2-Luc) luciferase reporter system. (B) A schematic of the synchronization protocol in which the circadian clocks of bone marrow-derived macrophages (BMDMs) derived from C57BL/6 mice expressing Per2-Luc were synchronized by a 24-h period of serum starvation in media with 0% serum, followed by a 2-h period of serum shock in media with 50% serum. (C) BMDMs were then cultured in RPMI/10% FBS supplemented with D-luciferin at circadian time (CT) 0. Luciferase activity of BMDMs was monitored in real time by LumiCycle, $n = 2$ biological replicates. (D, E) Protein and RNA were collected at the indicated times post-serum shock to assess (D) cAMP signaling by p-CREB levels and (E) expression of *Icer*, $n = 3$ biological replicates. Data information: For (C–E), experiments were replicated twice. For (C), shown are individual points and mean. For (E), mean and SEM are shown, and statistical significance determined by unpaired two-tailed t-test with Welch's correction; *$p < 0.05$; **$p < 0.005$; ***$p < 0.0005$; ****$p < 0.0001$; ns: not significant. Exact $p$ values: (E) 0.0039 (0 h vs 1 h), 0.0006 (0 h vs 2 h), <0.0001 (0 h vs 4 h).

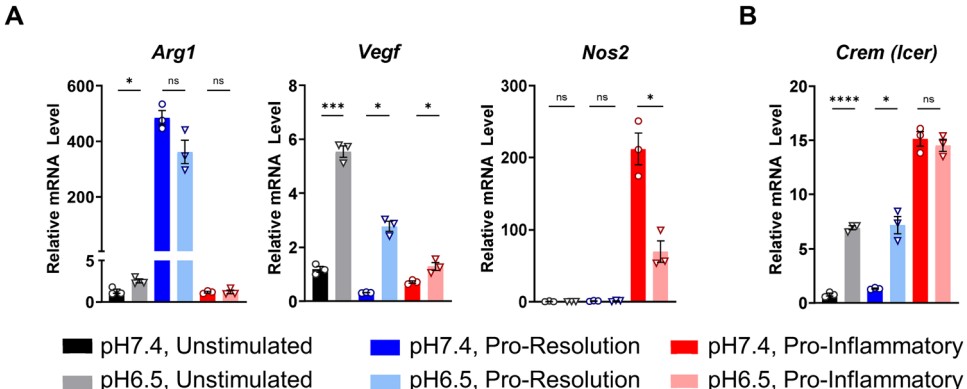

**Figure EV2. Macrophages sense and respond to an acidic extracellular environment when cultured in vitro in media with acidic pH.**

(A, B) Bone marrow-derived macrophages (BMDMs) were obtained from C57BL/6 mice expressing PER2-Luc. BMDMs were cultured in media with pH 7.4 or acidic media with pH 6.5, and stimulated with either 10 ng/mL IL-4 and 10 ng/mL IL-13 (pro-resolution), or 50 ng/mL IFNγ and 100 ng/mL LPS (pro-inflammatory); or left unstimulated. RNA was collected at 2 h post-treatment, and qt-PCR was performed to assess expression of genes associated with (A) phenotype or (B) acid sensing in macrophages. For both panels, $n = 3$ biological replicates. Data information: Shown are mean and SEM. Statistical significance determined by two-tailed t-test with Welch's correction. The Holm-Šídák correction for multiple t-tests were applied; *$p < 0.05$; **$p < 0.005$; ***$p < 0.0005$; ****$p < 0.0001$; ns: not significant. Experiments were replicated twice. Exact *p* values: (A) *Arg1* 0.024372 (pH 7.4 vs pH 6.5 unstimulated); *Vegf* 0.000403 (pH 7.4 vs pH 6.5 unstimulated), 0.011077 (pH 7.4 vs pH 6.5 pro-resolution), 0.048413 (pH 7.4 vs pH 6.5 pro-Inflammatory); *Nos2* 0.008509 (pH 7.4 vs pH 6.5 pro-inflammatory). (B) *Icer* < 0.0001 (pH 7.4 vs pH 6.5 unstimulated), 0.01749 (pH 7.4 vs pH 6.5 pro-resolution).

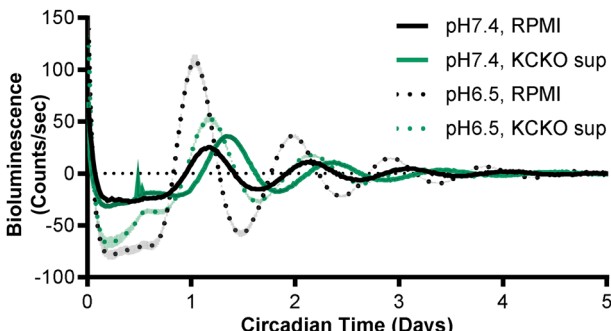

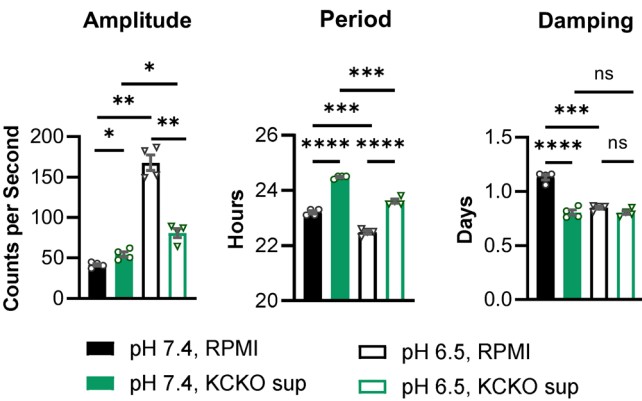

**Figure EV3. Exposure to cancer cell supernatant further modulates circadian rhythms in addition to pH-driven changes.**

Bone marrow-derived macrophages (BMDMs) were obtained from C57BL/6 mice expressing Per2-Luc. The circadian clocks of BMDMs were synchronized by a 24-h period of serum starvation in media with 0% serum, followed by a 2-h period of serum shock in media with 50% serum. BMDMs were then cultured in RPMI with neutral pH 7.4 or acidic pH 6.5, or in KCKO supernatant at pH 6.5 or pH-adjusted to pH 7.4. Luciferase activity was monitored in real time by LumiCycle. Data was baseline-subtracted using the running average, and oscillation parameters were measured by LumiCycle Analysis, $n = 4$ biological replicates. Data information: Shown is the mean and SEM. Statistical significance determined by unpaired two-tailed t-test with Welch's correction; *$p < 0.05$; **$p < 0.005$; ***$p < 0.0005$; ****$p < 0.0001$; ns: not significant. Exact $p$ values: Amplitude 0.0216 (pH 7.4 RPMI vs pH 7.4 KCKO sup), 0.0008 (pH 7.4 RPMI vs pH 6.5 RPMI), 0.0105 (pH 7.4 KCKO sup vs pH 6.5 KCKO sup), 0.0007 (pH 6.5 RPMI vs pH 6.5 KCKO sup); Period <0.0001 (pH 7.4 RPMI vs pH 7.4 KCKO sup), 0.0002 (pH 7.4 RPMI vs pH 6.5 RPMI), 0.0004 (pH 7.4 KCKO sup vs pH 6.5 KCKO sup), <0.0001 (pH 6.5 RPMI vs pH 6.5 KCKO sup); Damping <0.0001 (pH 7.4 RPMI vs pH 7.4 KCKO sup), 0.0002 (pH 7.4 RPMI vs pH 6.5 RPMI).

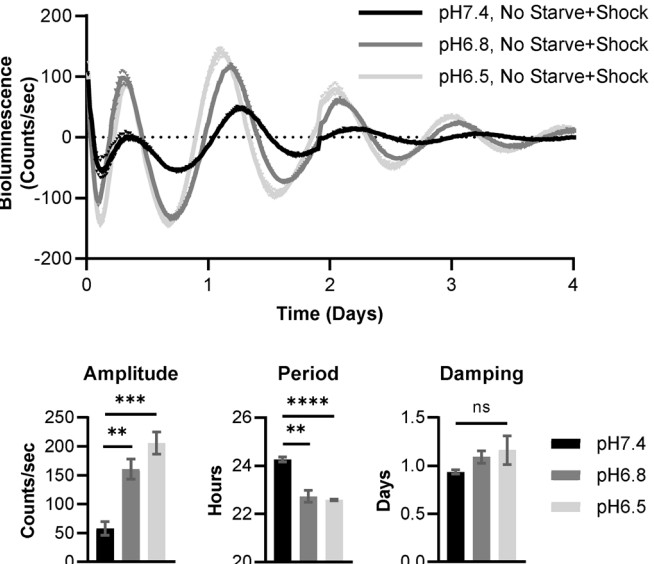

**Figure EV4. Acidic pH alters circadian rhythms in macrophages in the absence of prior serum starvation followed by serum shock.**

Bone marrow-derived macrophages (BMDMs) were obtained from C57BL/6 mice expressing Per2-Luc. BMDMs were cultured in media with neutral pH 7.4 or acidic pH 6.8 or 6.5. Luciferase activity was monitored in real time by LumiCycle (pH 7.4: $n = 3$. pH 6.8: $n = 2$. pH 6.5: $n = 3$). Data was baseline-subtracted using the running average, and oscillation parameters were measured by LumiCycle Analysis (pH 7.4: $n = 5$. pH 6.8: $n = 4$. pH 6.5: $n = 5$, data pooled from 2 individual experiments). Data information: For LumiCycle traces, shown are individual points and mean. For LumiCycle analysis, shown is the mean and SEM. Statistical significance determined by unpaired two-tailed t-test with Welch's correction (pH 6.8 and 6.5 comparison not tested); **$p < 0.005$; ***$p < 0.0005$; ****$p < 0.0001$. Exact $p$ values: Amplitude 0.0035 (pH 7.4 vs pH 6.8), 0.0004 (pH 7.4 vs pH 6.5); Period 0.0044 (pH 7.4 vs pH 6.8), <0.0001 (pH 7.4 vs pH 6.5).

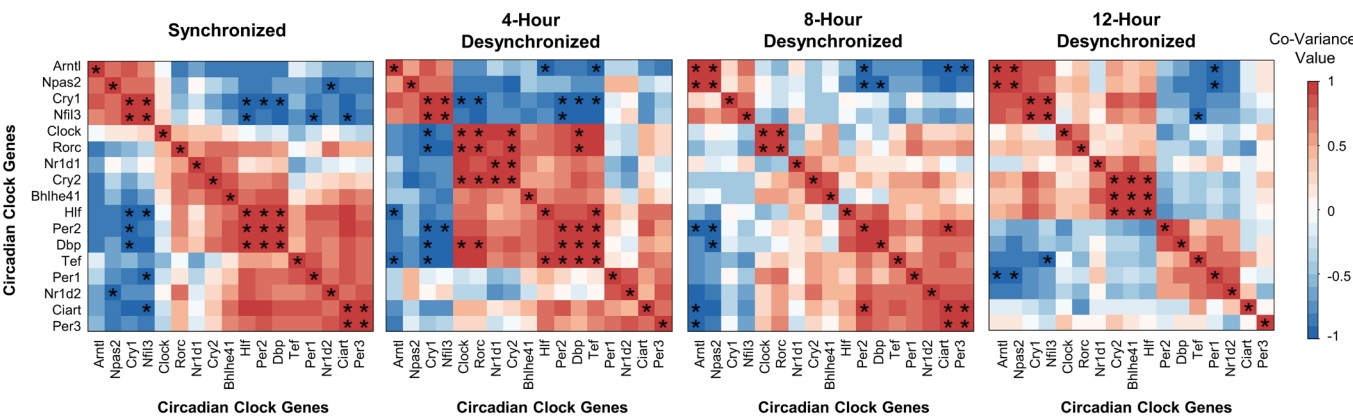

**Figure EV5. Heterogeneity in circadian rhythms of cells within a population can lead to an altered circadian clock gene network in samples.**

Increasingly desynchronized populations were modeled using an RNA-seq data set of WT peritoneal macrophages taken at 4-h intervals across two days, $n = 12$ biological replicates (see Fig. 8 and Methods). Weighted gene co-expression network analysis (WGCNA) was performed. Data information: *$p < 0.01$ by WGCNA for significant covariance by Pearson correlation, and blank squares are not significant.

