## [Peer Review File · EMBO Reports]

Circadian rhythms of macrophages are altered by the acidic tumor microenvironment

Amelia Knudsen-Clark, Daniel Mwangi, Juliana Cazarin, Kristina Morris, Cameron Baker, Lauren Hablitz, Matthew McCall, Minsoo Kim, and Brian Altman

Corresponding author(s): Brian Altman (brian_altman@urmc.rochester.edu)

Review Timeline:

Submission Date:	1st Aug 24
Editorial Decision:	15th Aug 24
Revision Received:	7th Sep 24
Editorial Decision:	19th Sep 24
Revision Received:	30th Sep 24
Accepted:	1st Oct 24

Editor: Achim Breiling

Transaction Report: This manuscript was transferred to EMBO reports following peer review at Review Commons.

**Review
COMMONS**

Review #1

1. Evidence, reproducibility and clarity:

Evidence, reproducibility and clarity (Required)

The manuscript by Knudsen-Clark et al. investigates the novel topic of circadian rhythms in macrophages and their role in tumorigenesis. The authors explore how circadian rhythms of macrophages may be influenced by the tumor microenvironment (TME). They utilize a system of bone marrow-derived macrophages obtained from transgenic mice carrying PER2-Luciferase (PER2-Luc), a trackable marker of rhythmic activity. The study evaluates how conditions associated with the TME, such as polarizing stimuli (to M1 or M2 subtype), acidic pH, and elevated lactate, can each alter circadian rhythms in macrophages. The authors employ several approaches to explore macrophage functions in cancer-related settings. While the manuscript presents interesting findings and may be the first to demonstrate that tumor stimuli alter circadian rhythms in macrophages and impact tumor growth, it lacks a clear conclusion regarding the role of altered circadian rhythms in suppressing tumor growth. Several discrepancies need to be addressed before publication, therefore, the manuscript requires revision before publication, addressing the following comments:

****Major comments:****

1. It is well known that pro-inflammatory macrophages primarily rely on glycolysis during inflammation, exhibiting dysregulated tricarboxylic acid (TCA) cycle activity. These pro-inflammatory macrophages are commonly referred to as 'M1' or pro-inflammatory, as noted in the manuscript. In contrast, M2 macrophages, or pro-resolution macrophages, are highly dependent on active mitochondrial respiration and oxidative phosphorylation (OXPHOS). Given that M1 macrophages favor glycolysis, they create an acidic environment due to elevated lactate levels and other acidifying metabolites. However, the study does not address this effect. The authors' hypothesis revolves around the acidic environment created by glycolytic tumors, yet they overlook the self-induced acidification of media when culturing M1 macrophages. This raises the question of how the authors explain the reduced circadian rhythms observed in pro-inflammatory macrophages in their study, while low pH and higher lactate levels enhance the amplitude of circadian rhythms. I would encourage the authors to incorporate the glycolytic activity of pro-inflammatory macrophages into their experimental setup. Otherwise the data look contradictory and misleading in some extent.

2. The article examines the role of circadian rhythms in tumor-associated macrophages, yet it lacks sufficient compelling data to support this assertion. Two figures, Figure 7 and Figure 9, are presented in relation to cancer. In Figure 7, gene expression analysis of Arg1 (an M2 marker) and Crem (a potential circadian clock gene) is conducted in wild-type macrophages, BMAL1-knockout macrophages with dysregulated circadian rhythms, and using publicly available data on tumor-associated macrophages from a study referenced as 83. However, it is noted that this referenced study is actually a review article by Geeraerts et al. (2017) titled "Macrophage Metabolism as Therapeutic Target for Cancer, Atherosclerosis, and Obesity" published in *Frontiers in Immunology*. This raises concerns about the reliability of the results. Furthermore, comparing peritoneal macrophages from healthy mice with macrophages isolated from lung tumors is deemed

inaccurate. It is suggested that lung macrophages from healthy mice and those from mice with lung tumors should be isolated separately for a more appropriate comparison. Consequently, Figure 7B is further questioned regarding how the authors could compare genes from the circadian rhythm pathway between these non-identical groups. As a result, the conclusion drawn from these data, suggesting that tumor-associated macrophages exhibit a gene expression pattern similar to BMAL1-KO macrophages, is deemed incorrect, affecting the interpretation of the data presented in Figure 8.

3. If the authors aim to draw a clear conclusion regarding the circadian rhythms of tumor-associated macrophages (TAMs), they may need to analyze single-sorted macrophages from tumors and corresponding healthy tissues. Such data are publicly available (of course not in #83)

4. Additionally, it is widely acknowledged that human and mouse macrophages exhibit distinct gene expression profiles, both in vitro and in vivo. While assuming that genes involved in circadian rhythms are conserved across species, the authors could consider extending their funding to include analyses of single-sorted macrophages from cancer patients, such as those with lung cancer or pancreatic ductal adenocarcinoma (PDAC). These experiments would provide relevant insights into TAM biology.

****Minor comments:****

1. Figure 2C needs clarification. It's unclear why pro-inflammatory macrophages treated with lactic acid would have a shorter amplitude and period, while acidic pH would increase amplitude and period in M2 macrophages.

2. The scale in Figure 2C should be equal for all conditions (e.g., -200).

3. Absolute values of amplitude, damping, and period differ between Figure 1 and Figure 2A, B, C. The authors should explain these discrepancies.

4. The authors should consider modulating the acidic environment of macrophages in settings more representative of cancer. For example, by adding conditioned media from tumor cells with pronounced glycolysis.

5. Arg1 alone is not sufficient as an M2 polarization marker. The authors should include additional markers.

2. Significance:

Significance (Required)

While the manuscript provides valuable insights and has obvious novelty, it requires a significant revision

3. How much time do you estimate the authors will need to complete the suggested revisions:

Estimated time to Complete Revisions (Required)

(Decision Recommendation)

Cannot tell / Not applicable

4. Review Commons values the work of reviewers and encourages them to get credit for their work. Select 'Yes' below to register your reviewing activity at Web of Science Reviewer

Recognition Service (formerly Publons); note that the content of your review will not be visible on Web of Science.

Yes

Review #2

1. Evidence, reproducibility and clarity:

Evidence, reproducibility and clarity (Required)

Knudsen-Clark et al. showed that the circadian rhythm of bone marrow-derived macrophages (BMDM) can be affected by polarization stimuli, pH of the microenvironment, and by the presence of sodium-lactate. Mechanistically, the acidic pH of cell microenvironment is partly regulated by intracellular cAMP-mediated signaling events in BMDM. The authors also showed that the circadian clock of peritoneal macrophages is also modified by the pH of the cell microenvironment. Using publicly available data, the authors showed that the circadian rhythm of tumor-associated macrophages is similar to that of Bmal1-KO peritoneal macrophages. In a murine model of pancreatic cancer, the authors showed that the tumor growth is accelerated in C57BL/6 mice co-injected with cancer cells and Bmal1-KO BMDM as compared to mice co-injected with cancer cell and wild type BMDM.

Major points of criticism:

1. Nine main figures include different experimental models on a non-systematic manner in the manuscript, and only literature-based correlation is used to link the results each other. The authors used in vitro BMDM and peritoneal cell-based model systems to study the effects of IL4+IL13, IFN γ +LPS, low pH, sodium-lactate, adenylate cyclase inhibitors on the circadian clock of macrophages. The link between these microenvironment conditions of the cells is still correlative with the tumor microenvironment: publicly available data were used to correlate the increased expression level of cAMP-activated signaling events with the presence of acidic pH of tumor microenvironment. Notably, the cell signaling messenger molecule cAMP is produced by not only low extracellular pH by activated GPCRs, but also starvation of the cell. The starvation is also relevant to this study, since the BMDM used in the in vitro culture system were starving for 24 hours before the measurement of Per2-Luc expression to monitor circadian rhythm.
2. The definition of pre-resolution macrophages (MF) used across the manuscript could be argued. The authors defined BMDM polarized with IL-4 and IL-13 as pre-resolution MF. Resolution is followed by inflammation, but the IL-4 secretion does not occur in every inflammatory setting. Moreover, IL-4 and IL-13 are secreted during specific tissue environment and immunological settings involving type 2 inflammation or during germinal center reactions of the lymph nodes.
 - What are the characteristics of pre-resolution macrophages (MF)? The authors indicated that IL-4 and IL-13 cytokines were used to model the pre-resolution macrophages. In which pathological

context are these cytokines produced and induce pre-resolution macrophages? IL-4 polarized BMDM can also produce pro-inflammatory protein and lipid mediators as compared to LPS-stimulated BMDM, and IL-4 polarized BMDM still have potent capacity to recruit immune cells and to establish type 2 inflammation.

- The authors showed Arg1 and Vegfa qPCR data from BMDM only. Based on the literature, these MFs are anti-inflammatory cells particularly. Resolution-related MFs followed by acute inflammation are a specific subset of MFs, and the phenotype of pre-resolution MF should be described, referred, and measured specifically.

3. The authors used IFN γ and LPS, or IL-4 and IL-13 and co-treatments to polarize BMDM in to type 1 (referred as pro-inflammatory MF) and type 2 (referred as pre-resolution MF) activation state. The comparison between these BMDM populations has limitations, since LPS induces a potent inflammatory response in MF. The single treatment with MF-polarizing cytokines enable a more relevant comparison to study the circadian clock in classically and alternatively activated MF.

4. There are missing links between the results of showing the circadian rhythm of polarized BMDM, sodium-lactate treated BMDM, and tumor growth. Specifically, do the used pancreatic ductal adenocarcinoma cells produce IL-4 and sodium-lactate?

In the LLC-based experimental in silico analysis of tumors, the LLC do not produce IL-4.

5. How can the circadian rhythm affect the function of BMDM? The Authors should provide evidence that circadian rhythm affects the function of polarized MF.

6. In Figure 3, the authors show data from peritoneal cells. The isolated peritoneal cells are not pure macrophage populations. Based on the referred article in the manuscript, the peritoneal cavity contains more than 50% of lymphocytes, and the myeloid compartment contains 80% macrophages.

7. The figure legend of Figure 3 describes the effects of pH on the circadian rhythm of bone marrow-derived macrophages ex vivo. Peritoneal macrophages involve tissue resident peritoneal macrophages with yolk sac and fetal liver origin, and also involve small peritoneal MF with bone marrow origin. The altered description of results and figure legends makes confusion.

8. In Figure 6C, one single Western blot is shown with any quantification. The authors should provide data of the relative protein level of p-CREB from at least 3 independent experiments. In the Western-blot part of the methods, the authors described that the pellet was discarded after cell lysis. The p-CREB is the activated form of the transcription factor CREB and there is increased binding to the chromatin to regulate gene expression. By discarding the pellet after cell lysis, the chromatin-bound p-CREB could be also removed at the same time.

9. It is confusing that adenylate-cyclase inhibitor MDL-12 elevated the phospho-CREB levels in BMDM. How can the authors exclude any other inducers of CREB phosphorylation?

10. It is described in the methods that BMDM were starving for 24 hours in serum-free culture media followed by serum shock (50% FBS). The cAMP production can be induced during cell starvation which should be considered for the data representation.

11. How can the authors explain and prove that the wild type and Bmal1-KO BMDM co-injected with pancreatic cancer cells subcutaneously survive, present, and have effector functions at the same extent in the subcutaneous tissue, before and during tumor growth (Figure 9)? In other words, what kind of MF-derived parameters could be modified by disrupting the circadian rhythm of MF during tumor development? The production of MF-derived regulatory enzymes, cytokines, growth factors are affected by the disrupted circadian clock in MF?

Minor points of criticism:

1. The figure legends of the graphs and diagrams are missing in Figure 2D,E,F
2. The BMAL1-based in vivo murine model of circadian rhythm is not introduced in the manuscript.

2. Significance:

Significance (Required)

Knudsen-Clark et al. showed that the circadian rhythm of bone marrow-derived macrophages (BMDM) can be affected by polarization stimuli, pH of the microenvironment, and by the presence of sodium-lactate. Mechanistically, the acidic pH of cell microenvironment is partly regulated by intracellular cAMP-mediated signaling events in BMDM. The authors also showed that the circadian clock of peritoneal macrophages is also modified by the pH of the cell microenvironment. Using publicly available data, the authors showed that the circadian rhythm of tumor-associated macrophages is similar to that of Bmal1-KO peritoneal macrophages. In a murine model of pancreatic cancer, the authors showed that the tumor growth is accelerated in C57BL/6 mice co-injected with cancer cells and Bmal1-KO BMDM as compared to mice co-injected with cancer cell and wild type BMDM.

****Major points of criticism:****

1. Nine main figures include different experimental models on a non-systematic manner in the manuscript, and only literature-based correlation is used to link the results each other. The authors used in vitro BMDM and peritoneal cell-based model systems to study the effects of IL4+IL13, IFN γ +LPS, low pH, sodium-lactate, adenylylate cyclase inhibitors on the circadian clock of macrophages. The link between these microenvironment conditions of the cells is still correlative with the tumor microenvironment: publicly available data were used to correlate the increased expression level of cAMP-activated signaling events with the presence of acidic pH of tumor microenvironment. Notably, the cell signaling messenger molecule cAMP is produced by not only low extracellular pH by activated GPCRs, but also starvation of the cell. The starvation is also relevant to this study, since the BMDM used in the in vitro culture system were starving for 24 hours before the measurement of Per2-Luc expression to monitor circadian rhythm.
2. The definition of pre-resolution macrophages (MF) used across the manuscript could be argued. The authors defined BMDM polarized with IL-4 and IL-13 as pre-resolution MF. Resolution is followed by inflammation, but the IL-4 secretion does not occur in every inflammatory setting. Moreover, IL-4 and IL-13 are secreted during specific tissue environment and immunological settings involving type 2 inflammation or during germinal center reactions of the lymph nodes.
 - What are the characteristics of pre-resolution macrophages (MF)? The authors indicated that IL-4 and IL-13 cytokines were used to model the pre-resolution macrophages. In which pathological context are these cytokines produced and induce pre-resolution macrophages? IL-4 polarized BMDM can also produce pro-inflammatory protein and lipid mediators as compared to LPS-stimulated BMDM, and IL-4 polarized BMDM still have potent capacity to recruit immune cells and to establish type 2 inflammation.
 - The authors showed Arg1 and Vegfa qPCR data from BMDM only. Based on the literature, these MFs are anti-inflammatory cells particularly. Resolution-related MFs followed by acute

inflammation are a specific subset of MFs, and the phenotype of pre-resolution MF should be described, referred, and measured specifically.

3. The authors used IFN γ and LPS, or IL-4 and IL-13 and co-treatments to polarize BMDM in to type 1 (referred as pro-inflammatory MF) and type 2 (referred as pre-resolution MF) activation state. The comparison between these BMDM populations has limitations, since LPS induces a potent inflammatory response in MF. The single treatment with MF-polarizing cytokines enable a more relevant comparison to study the circadian clock in classically and alternatively activated MF.

4. There are missing links between the results of showing the circadian rhythm of polarized BMDM, sodium-lactate treated BMDM, and tumor growth. Specifically, do the used pancreatic ductal adenocarcinoma cells produce IL-4 and sodium-lactate?

In the LLC-based experimental in silico analysis of tumors, the LLC do not produce IL-4.

5. How can the circadian rhythm affect the function of BMDM? The Authors should provide evidence that circadian rhythm affects the function of polarized MF.

6. In Figure 3, the authors show data from peritoneal cells. The isolated peritoneal cells are not pure macrophage populations. Based on the referred article in the manuscript, the peritoneal cavity contains more than 50% of lymphocytes, and the myeloid compartment contains 80% macrophages.

7. The figure legend of Figure 3 describes the effects of pH on the circadian rhythm of bone marrow-derived macrophages ex vivo. Peritoneal macrophages involve tissue resident peritoneal macrophages with yolk sac and fetal liver origin, and also involve small peritoneal MF with bone marrow origin. The altered description of results and figure legends makes confusion.

8. In Figure 6C, one single Western blot is shown with any quantification. The authors should provide data of the relative protein level of p-CREB from at least 3 independent experiments. In the Western-blot part of the methods, the authors described that the pellet was discarded after cell lysis. The p-CREB is the activated form of the transcription factor CREB and there is increased binding to the chromatin to regulate gene expression. By discarding the pellet after cell lysis, the chromatin-bound p-CREB could be also removed at the same time.

9. It is confusing that adenylate-cyclase inhibitor MDL-12 elevated the phospho-CREB levels in BMDM. How can the authors exclude any other inducers of CREB phosphorylation?

10. It is described in the methods that BMDM were starving for 24 hours in serum-free culture media followed by serum shock (50% FBS). The cAMP production can be induced during cell starvation which should be considered for the data representation.

11. How can the authors explain and prove that the wild type and Bmal1-KO BMDM co-injected with pancreatic cancer cells subcutaneously survive, present, and have effector functions at the same extent in the subcutaneous tissue, before and during tumor growth (Figure 9)? In other words, what kind of MF-derived parameters could be modified by disrupting the circadian rhythm of MF during tumor development? The production of MF-derived regulatory enzymes, cytokines, growth factors are affected by the disrupted circadian clock in MF?

****Minor points of criticism:****

1. The figure legends of the graphs and diagrams are missing in Figure 2D,E,F

2. The BMAL1-based in vivo murine model of circadian rhythm is not introduced in the manuscript.

3. How much time do you estimate the authors will need to complete the suggested revisions:

Estimated time to Complete Revisions (Required)

(Decision Recommendation)

Between 3 and 6 months

4. Review Commons values the work of reviewers and encourages them to get credit for their work. Select 'Yes' below to register your reviewing activity at Web of Science Reviewer Recognition Service (formerly Publons); note that the content of your review will not be visible on Web of Science.

Yes

Review #3

1. Evidence, reproducibility and clarity:

Evidence, reproducibility and clarity (Required)

Review for Knudsen-Clark et al.

"Circadian rhythms of macrophages are altered by the acidic pH of the tumor microenvironment"

Knudsen-Clark and colleagues explore the impact of TME alterations on macrophage circadian rhythms. The authors find that both acidic pH and lactate modulate circadian rhythms which alter macrophage phenotype. Importantly, they define circadian disruption of tumor-associated macrophages within the TME and show that circadian disruption in macrophages promotes tumor growth using a PDAC line. This represents an important understanding of the crosstalk between cancer cells and immune cells as well as the understanding of how the TME disrupts circadian rhythms. The study is well-done, however, authors need to address several important points below.

****Major comments:****

1. In Figures 3 and 4, the authors can include additional clock genes that can be run by qPCR. This was done in Figure 2 and was a nice addition to the data.
2. There are far too many figures with minimal data in each. Please consolidate the figures. For example, Figures 1-3 can be fully combined, Figures 4-6 can be combined, and Figures 7-8 can be combined. Additionally, it is unclear if Figure 5 needs to be in the main, it can be moved to the supplement.
3. The observation that conditions like pH and lactate alter macrophage phenotype and rhythmicity are important. However, macrophage phenotype via gene expression does not always correlate to function. It is important for authors to demonstrate the effect of pH or lactate on macrophage function. This can be done using co-culture assays with cancer cells. If these experiments cannot be performed, it is suggested that authors discuss these limitations and consideration in the discussion.

4. On line 119-122, authors describe a method for polarization of macrophages. They then reference one gene to confirm each macrophage polarization state. To more definitively corroborate proper macrophage polarization, authors should perform qPCR for additional target genes that are associated with each phenotype. For example, Socs3, CD68, or CD80 for M1, and CD163 or VEGF for M2. Alternatively, the authors should cite previous literature validating this in vitro polarization model.
5. Several portions of the manuscript are unnecessarily long, including the intro and discussion. Please consolidate the text. The results section is also very lengthy, please consider consolidation.
6. The authors find that macrophage phenotype impacts rhythmicity. However, there is no mechanistic understanding of why this occurs. The authors should provide some mechanistic insight on this topic in the discussion.
7. The data presented in Figure 9 is very intriguing and arguably the strongest aspect of the paper. To strengthen the point, the authors could repeat this experiment with an additional cell model, another PDAC line or a different cancer line.

****Minor Comments:****

1. Data in Figure 2 is interesting and the impact on circadian rhythms is clear based on changes in amplitude and period. However, though the impact on period and amplitude is clear from Figures 2A-C, the changes in circadian gene expression are less clear. For instance, though amplitude is up in 2B, amplitude is suppressed in 2C. However, that does not appear to be reflected in the gene expression data in Figures 2E and F. The authors should comment on this.
2. On line 156-158, authors describe damping rate. I believe the authors are trying to say that damping rate increases as the time it takes cells to desynchronize decreases and vice versa. However, this point needs to be better explained.
3. Data presented in Figures 3 and 4 are different in terms of the impact of changing the pH. The source of the macrophages is different, but the authors could clarify this further.
4. For heatmaps shown in Figures 7 and 8, please calculate covariance and display asterisks where $P < 0.001$. This will more clearly demonstrate the loss of co-variance between the clock network as a result of clock disruption, the TME, and cell desynchrony.

2. Significance:

Significance (Required)

This is an important study that is well-done. It is the feeling of the reviewer that the study warrants a revision, at the discretion of the editor. The study represents an important understanding of the crosstalk between cancer cells and immune cells as well as the understanding of how the TME disrupts circadian rhythms.

3. How much time do you estimate the authors will need to complete the suggested revisions:

Estimated time to Complete Revisions (Required)

(Decision Recommendation)

Between 1 and 3 months

No

Full Revision

Manuscript number: RC-2024-02394

Corresponding author(s): Altman, Brian J

[Please use this template only if the submitted manuscript should be considered by the affiliate journal as a full revision in response to the points raised by the reviewers.]

*If you wish to submit a preliminary revision with a revision plan, please use our "Revision Plan" template. **It is important to use the appropriate template to clearly inform the editors of your intentions.**]*

1. General Statements [optional]

We thank all three Reviewers for their insightful and helpful feedback and suggestions. We strongly believe that addressing these comments has now resulted in a much-improved manuscript. We appreciate that the Reviewers found the manuscript "interesting" with "valuable insights and... obvious novelty", "an important study that is well-done", and "an important understanding of the crosstalk between cancer cells and immune cells as well as the understanding of how the TME disrupts circadian rhythms". All three Reviewers requested a significant revision, which we provide here. We carefully and completely responded to each Reviewer question or suggestion, in most cases with new experiments and text, and in a very few cases with changes or additions to the Discussion section. This includes new data in seven of the original Figures and Supplementary Figures, and one new main Figure and three new Supplementary Figures. Highlights of these new data include testing the role of low pH in cancer cell supernatant on macrophage rhythms, and analysis of single-cell RNA-sequencing data for heterogeneity in macrophage circadian gene expression. Additional experiments were also performed that were not included in the manuscript, and these data are presented in this Response. A detailed point-by-point response to each comment is included below with excerpts of the data and updated text for the reviewers.

Reviewer #1

Evidence, reproducibility and clarity

The manuscript by Knudsen-Clark et al. investigates the novel topic of circadian rhythms in macrophages and their role in tumorigenesis. The authors explore how circadian rhythms of macrophages may be influenced by the tumor microenvironment (TME). They utilize a system of bone marrow-derived macrophages obtained from transgenic mice carrying PER2-Luciferase (PER2-Luc), a trackable marker of rhythmic activity. The study evaluates how conditions associated with the TME, such as polarizing stimuli (to M1 or M2 subtype), acidic pH, and elevated lactate, can each alter circadian rhythms in macrophages. The authors employ several approaches to explore macrophage functions in cancer-related settings. While the manuscript presents interesting findings and may be the first to demonstrate that tumor stimuli alter circadian rhythms in macrophages and impact tumor growth, it lacks a clear conclusion

Full Revision

regarding the role of altered circadian rhythms in suppressing tumor growth. Several discrepancies need to be addressed before publication, therefore, the manuscript requires revision before publication, addressing the following comments:

We thank Reviewer #1 for the comments regarding the quality of our work and are pleased that the Reviewer finds that this manuscript “presents interesting findings and may be the first to demonstrate that tumor stimuli alter circadian rhythms in macrophages and impact tumor growth”. We have addressed all comments and critiques from Reviewer #1 below. To summarize, we added new data on how different macrophage polarization states affect media pH (Supplementary Figure 4), further characterized gene expression in our distinct macrophage populations (Supplementary Figure 1), provided clarity in the data and text on the universal nature of Clock Correlation Distance (CCD) across macrophage populations (Figure 6), included human tumor-associated macrophage (TAM) data for CCD (Figure 7) analyzed single-cell RNA-sequencing data of TAMs to demonstrate heterogeneity in circadian gene expression (Figure 9), and used tumor-conditioned media to show that low pH still affects macrophage rhythms in this context *Supplementary Figure 5”. Thanks to the helpful suggestions of the Reviewer, we also made numerous clarifications and fixed a critical referencing error that the Reviewer identified.

Major comments:

1. It is well known that pro-inflammatory macrophages primarily rely on glycolysis during inflammation, exhibiting dysregulated tricarboxylic acid (TCA) cycle activity. These pro-inflammatory macrophages are commonly referred to as 'M1' or pro-inflammatory, as noted in the manuscript. In contrast, M2 macrophages, or pro-resolution macrophages, are highly dependent on active mitochondrial respiration and oxidative phosphorylation (OXPHOS). Given that M1 macrophages favor glycolysis, they create an acidic environment due to elevated lactate levels and other acidifying metabolites. However, the study does not address this effect. The authors' hypothesis revolves around the acidic environment created by glycolytic tumors, yet they overlook the self-induced acidification of media when culturing M1 macrophages. This raises the question of how the authors explain the reduced circadian rhythms observed in pro-inflammatory macrophages in their study, while low pH and higher lactate levels enhance the amplitude of circadian rhythms. I would encourage the authors to incorporate the glycolytic activity of pro-inflammatory macrophages into their experimental setup. Otherwise the data look contradictory and misleading in some extent.

We appreciate the important point Reviewer #1 made that macrophages polarized toward a pro-inflammatory phenotype such as those stimulated with IFN γ and LPS (M1 macrophages) prioritize metabolic pathways that enhance glycolytic flux, resulting in increased release of protons and lactate as waste products from the glycolysis pathway. In this way, polarization of macrophages toward the pro-inflammatory phenotype can lead to acidification of the media, which may influence our observations given that we are studying the effect of extracellular pH on rhythms in macrophages. To address this point, we have performed additional experiments in which we measured pH of the media to capture changes in media pH that occur during the time in which we observe changes in rhythms of pro-inflammatory macrophages.

In line with the documented enhanced glycolytic activity of pro-inflammatory macrophages, the media of pro-inflammatory macrophages is acidified over time, in contrast to media of unstimulated or pro-resolution macrophages. Notably, while pH decreased over time in the pro-inflammatory group, the pH differential between the pH7.4, pH6.8, and pH6.5 sample groups was maintained over the period in which we observe and measure changes in circadian rhythms of pro-inflammatory macrophages. Additionally, media that began at pH 7.4 was acidified only to pH 7 by day 2, above the acidic pH of 6.8 or 6.5. As a result, there remained a difference in pH between the two groups (pH 7.4 and pH 6.5) out to 2 days consistent with the changes in rhythms that we observe between these two groups. This indicates that the difference in circadian rhythms observed in pro-inflammatory macrophages cultured at pH 7.4 compared to pH 6.5 were indeed due to the difference in extracellular pH between the two conditions. We have incorporated these data, shown below, into Supplementary Figure 4 and added the following discussion of these data to the Results section:

“In line with their documented enhanced glycolytic capacity, pro-inflammatory macrophages acidified the media over time (Supplementary Figure 4C). Notably, while pH of the media the pro-inflammatory macrophages were cultured in decreased over time pH, the pH differential between the pH 7.4, pH 6.8, and pH 6.5 samples groups of pro-inflammatory macrophages was maintained out to 2 days, consistent with the changes in rhythms that we observe and measure between these groups.”

2. The article examines the role of circadian rhythms in tumor-associated macrophages, yet it lacks sufficient compelling data to support this assertion. Two figures, Figure 7 and Figure 9, are presented in relation to cancer. In Figure 7, gene expression analysis of Arg1 (an M2 marker) and Crem (a potential circadian clock gene) is conducted in wild-type macrophages, BMAL1-knockout macrophages with dysregulated circadian rhythms, and using publicly available data on tumor-associated macrophages from a study referenced as 83. However, it is noted that this referenced study is actually a review article by Geeraerts et al. (2017) titled "Macrophage Metabolism as Therapeutic Target for Cancer, Atherosclerosis, and Obesity" published in *Frontiers in Immunology*. This raises concerns about the reliability of the results. Furthermore, comparing peritoneal macrophages from healthy mice with macrophages isolated from lung tumors is deemed inaccurate. It is suggested that lung macrophages from healthy mice and those from mice with lung tumors should be isolated separately for a more appropriate comparison. Consequently, Figure 7B is further questioned regarding how the authors could

Full Revision

compare genes from the circadian rhythm pathway between these non-identical groups. As a result, the conclusion drawn from these data, suggesting that tumor-associated macrophages exhibit a gene expression pattern similar to BMAL1-KO macrophages, is deemed incorrect, affecting the interpretation of the data presented in Figure 8.

We thank Reviewer #1 for pointing out our error in the reference provided as the source of the TAM data used for CCD in Figure 7. While we took care to provide the GEO ID for the data set (GSE188549) in the Methods section, we mistakenly cited Geeraerts (2017) *Front Immunol* when we should have cited Geeraerts (2021) *Cell Rep*. We have corrected this citation error in the main text.

We also appreciate Reviewer #1's concern that we are comparing circadian gene expression of peritoneal macrophages to tumor-associated macrophages derived from LLC tumors, which are grown ectopically in the flank for the experiment from which the data set was produced. To ensure an accurate comparison of gene expression, we downloaded the raw FASTQ files from each dataset and processed them in identical pipelines. Our main comparison between these cell types is Clock Correlation Distance (CCD), which compares the pattern of co-expression of circadian genes (Shilts et al *PeerJ* 2018). CCD was built from multiple mouse and human tissues to be a "universal" tool to compare circadian rhythms, and designed to compare between different tissues and cell types. Each sample is compared to a reference control built from these multiple tissues. To better convey this concept to readers to give confidence the suitability of CCD for comparing data sets across different tissues, we have added the reference control to Figure 7 (now Figure 6B), We have also expanded our analysis to include bone marrow-derived macrophages, to further demonstrate that the organization of clock gene co-expression is not specific to peritoneal macrophages; we have added this data to Figure 7 (now Figure 6C,D). Finally, we have included an abbreviated explanation of the points made above in the results section.

Due to the universal nature of the CCD tool, we disagree with Reviewer #1's assertion that "the conclusion drawn from these data, suggesting that tumor-associated macrophages exhibit a gene expression pattern similar to BMAL1-KO macrophages, is deemed incorrect". Indeed, this finding mirrors findings in the original CCD paper, which showed that tumor tissues universally exhibit a disordered molecular clock as compared to normal tissue. Notably, the original CCD paper also compared across cell and tumor types.

As an additional note to the review, we would like to clarify that nowhere in the manuscript do we propose that *Crem* is a potential circadian clock gene. We are clear throughout the manuscript that we are using *Crem* as a previously established biomarker for acidic pH-sensing in macrophages. Please see below for the modified Figure and text.

“To understand the status of the circadian clock in TAMs, we performed clock correlation distance (CCD) analysis. This analysis has previously been used to assess functionality of the circadian clock in whole tumor and in normal tissue[102]. As the circadian clock is comprised of a series of transcription/translation feedback loops, gene expression is highly organized in a functional, intact clock, with core clock genes existing in levels relative to each other irrespective of the time of day. In a synchronized population of cells, this ordered relationship is maintained at the population level, which can be visualized in a heatmap. CCD is designed to compare circadian clock gene co-expression patterns between different tissues and cell types. To accomplish this, CCD was built using datasets from multiple different healthy tissues from mouse and human to be a universal tool to compare circadian rhythms. Each sample is compared to a reference control built from these multiple tissues (Figure 6B)[102]. To validate the use of this analysis for assessing circadian disorder in macrophages, we performed CCD analysis using publicly available RNA-sequencing data from bone marrow-derived macrophages and wild type peritoneal macrophages, as a healthy control for functional

Full Revision

rhythms in a synchronized cell population, and BMAL1 KO peritoneal macrophages, as a positive control for circadian disorder[44].”

And in the Discussion:

“Interestingly, analysis of TAMs by clock correlation distance (CCD) presents evidence that rhythms are disordered in bulk TAMs compared to other macrophage populations (Figure 6). CCD is one of the most practical tools currently available to assess circadian rhythms due to its ability to assess rhythms independent of time of day and without the need for a circadian time series, which is often not available in publicly available data from mice and humans[102].”

3. If the authors aim to draw a clear conclusion regarding the circadian rhythms of tumor-associated macrophages (TAMs), they may need to analyze single-sorted macrophages from tumors and corresponding healthy tissues. Such data are publicly available (of course not in #83)

We agree with Reviewer #1 that while our interpretation of the data is that there may be heterogeneity in circadian rhythms of tumor-associated macrophages, we cannot prove this without assessing circadian rhythms at the single cell level. While single-cell RNA-sequencing data of freshly isolated tumor associated macrophages of sufficient read depth for circadian gene expression analysis has historically been unavailable, fortunately a dataset was released recently (May 2024) which we were able to use to address this point. We have analyzed publicly available single-cell RNAseq data of tumor-associated macrophages (GSE260641, Wang 2024 Cell) to determine whether there are differences in expression of circadian clock genes between different TAM populations. We have added these data as a new Figure 9. Please see the figure and updated text below.

“Tumor-associated macrophages exhibit heterogeneity in circadian clock gene expression.”

Our findings suggested that heterogeneity of the circadian clock may lead to disorder in bulk macrophage populations, but did not reveal if specific gene expression changes exist in tumor-associated macrophages at the single-cell level. To determine whether heterogeneity exists within the expression of circadian clock genes of the tumor-associated macrophage population, we analyzed publicly available single-cell RNA sequencing data of macrophages isolated from B16-F10 tumors[107]. To capture the heterogeneity of macrophage subsets within the TAM population, we performed unbiased clustering (Figure 9A). We then performed differential gene expression to determine if circadian clock genes were differentially expressed within the TAM subpopulations. The circadian clock genes Bhlhe40 (DEC1), Bhlhe41 (DEC2), Nfil3 (E4BP4), Rora (ROR α), Dbp (DBP), and Nr1d2 (REV-ERB β) were significantly ($adj.p < 0.005$) differentially expressed between TAM clusters (Figure 9B). This indicates

that there is heterogeneity in expression of circadian clock genes within the TAM population.

We next sought to determine whether differences in circadian clock gene expression between TAM subpopulations were associated with exposure to acidic pH in the TME. To this end, we first assessed Crem expression in the TAM subpopulations that were identified by unbiased clustering. Crem expression was significantly higher in TAM clusters 4, 5, and 6 compared to TAM clusters 1-3 and 7-9 (Figure 9C). Clusters were subset based on Crem expression into Crem high (clusters 4-6) and Crem low (clusters 1-3, 7-9) (Figure 9D), and differential gene expression analysis was performed. The circadian clock genes Nfil3, Rora, Bhlhe40, and Cry1 (CRY1) were significantly ($\text{adj.}p < 0.005$) differentially expressed between Crem high and Crem low TAMs (Figure 9E). This suggests that acidity within the TME is associated with heterogeneity in expression of circadian clock genes within the TAM population. Interestingly, expression of circadian clock genes varied between clusters designated as Crem high or Crem low (Figure 9B); for instance, Nfil3 was more highly expressed in cluster 1 than cluster 3, both of which had low Crem expression. This indicates that there is diversity in circadian clock gene expression within the Crem high and Crem low groups, suggesting that acidic pH is not the only factor in the TME that can alter the circadian clock. Collectively, these data suggest that there is heterogeneity in the circadian clock of macrophages within the TAM population that is driven in part by acidic pH.”

And in the Discussion:

“Supporting the notion that population-level disorder may exist in TAMs, we used scRNA-sequencing data and found evidence of heterogeneity between the expression of circadian clock genes in different TAM subpopulations (Figure 9A, B). Phenotypic heterogeneity of TAMs in various types of cancer has previously been shown[20, 21, 125, 126], and we have identified distinct TAM subpopulations by unbiased clustering (Figure 9A). Within those TAM subpopulations, we identified differential expression of circadian clock genes encoding transcription factors that bind to different consensus sequences: DEC1 and DEC2 bind to E-boxes, NFIL3 and DBP binds to D-boxes, and ROR α and REV-ERB β binds to retinoic acid-related orphan receptor elements (ROREs)[127, 128]. While little is known about regulation of macrophages by E-box and D-box elements beyond the circadian clock, aspects of macrophage function have been shown to be subject to transcriptional regulation through ROREs[129, 130]. Thus, we speculate that variations in these transcription factors may exert influence on expression of genes to drive diversity between TAM subpopulations. Differential expression of circadian clock genes between TAM subpopulations was also associated with Crem expression (Figure 9C-E), suggesting that exposure of TAMs to acidic pH within the TME can alter the circadian clock. However, there remained significant variation in expression of circadian clock genes within the Crem high and Crem low groups (Figure

9B), suggesting that acidic pH is not the only factor in the TME that can alter the circadian clock. Together, these data implicate the TME in driving heterogeneity in TAM circadian rhythms just as it drives heterogeneity in TAM phenotype.

Interestingly, in contrast to our observations of circadian disorder in TAMs isolated from LLC tumors (Figure 6), rhythmicity in expression of circadian genes was observed in bulk TAMs isolated from B16 tumors[107]. This suggests that circadian rhythms of TAMs are maintained differently in different types of cancer. Notably, both of these observations were at the population level. Upon separation of the B16 TAM population into subsets by unbiased clustering of single-cell RNA sequencing data, we measured differences in expression of circadian clock genes between TAM subpopulations (Figure 9A,B). This suggests that even within a rhythmic TAM population, there is heterogeneity in the circadian clock of TAM subpopulations.”

4. Additionally, it is widely acknowledged that human and mouse macrophages exhibit distinct gene expression profiles, both in vitro and in vivo. While assuming that genes involved in circadian rhythms are conserved across species, the authors could consider extending their funding to include analyses of single-sorted macrophages from cancer patients, such as those with lung cancer or pancreatic ductal adenocarcinoma (PDAC). These experiments would provide relevant insights into TAM biology.

We agree that with Reviewer #1 that ultimately, being able to relate findings in mice to humans is critical. It is important to assess if circadian disorder is observed in TAMs in human cancers as it is for LLC tumor-derived macrophages in mice. To address this point, we have performed CCD using a human data set (GSE116946; Garrido 2020 *J Immunother Cancer*) suitable for use with CCD (wherein macrophages were isolated from bulk tumor in humans, with a high enough samples size, and not cultured prior to sequencing). We have added these data as a new Figure 7, shown below. Please see the added data and updated text below.

“We next assessed the status of the circadian clock in human TAMs from NSCLC patients. We performed CCD with publicly available RNA-seq data of tumor-adjacent macrophages and tumor-associated macrophages from NSCLC patients, using alveolar macrophages from healthy donors as a control[104, 105]. To assess the contribution of the acidic TME to circadian disorder, we subset TAM NSCLC patient samples into groups (*Crem* high TAMs and *Crem* low TAMs) based on median *Crem* expression. Notably, in macrophages from human NSCLC there was a trend toward disorder in *Crem* low but not *Crem* high TAM samples (Figure 7A,B). Additionally, the co-variance among core clock genes observed in alveolar macrophages from healthy donors was absent within *Crem* low and *Crem* high TAM samples (Figure 7C). In all, these data indicate that there is population-level disorder in the circadian rhythms of tumor-associated macrophages in humans and

mice, suggesting that circadian rhythms are indeed altered in macrophages within the TME.”

And in the Discussion:

“Indeed, we observed differences in the circadian clock of Crem low human TAM samples compared to Crem high human TAM samples, suggesting that acidic pH influences circadian disorder in TAMs (Figure 7). Interestingly, Crem low TAM samples exhibited a trend toward disorder while Crem high TAM samples did not. This is of particular interest, as we have observed that acidic pH can enhance circadian rhythms in macrophages, raising the question of whether acidic pH promotes or protects against circadian disorder.”

Minor comments:

1. Figure 2C needs clarification. It's unclear why pro-inflammatory macrophages treated with lactic acid would have a shorter amplitude and period, while acidic pH would increase amplitude and period in M2 macrophages.

We thank Reviewer #1 for this important observation. Based on the comment, it is our understanding that the Reviewer is referring to the data in Figure 2 (low pH) compared to Figure 4 (lactate). We also find it very interesting that lactate alters rhythms in a manner distinct from the way in which acidic pH alters rhythms. Reviewer 3 asked for clarification on how lactate affected circadian gene expression in pH 7.4 or 6.5. We have added these data as Figure 4C (data and text below). It is notable that lactate opposing effects on circadian gene expression in pH 6.5, enhancing the effects of low pH in some cases (*Nr1d1*) while blunting them in other cases (*Cry1*). This is mentioned in the text.

*“Lactate was also observed to alter expression of the circadian clock genes *Per2*, *Cry1*, and *Nr1d1* over time in BMDMs cultured at pH 6.5, while having more subtle effects at pH 7.4 (Figure 4C). Notably, lactate blunted the effect of pH 6.5 on *Cry1* expression, while enhancing the effect of low pH on *Nr1d1* expression.”*

Why these two stimuli alter rhythms differently remains an open question that is discussed in the Discussion section and is prime to be a topic of future investigation. We have added to the Discussion section potential reasons why these conditions may alter rhythms

differently, such as the different pathways downstream of sensing these two different conditions. Please see the updated text, below.

*“Although lactate polarizes macrophages toward a pro-resolution phenotype similar to acidic pH[30, 93], exposure to lactate had different effects on circadian rhythms – and in some cases, circadian clock gene expression – than exposure to acidic pH (Figure 4). Sensing of lactate occurs through different pathways than acid-sensing, which may contribute to the different ways in which these two stimuli modulate circadian rhythms of macrophages[111]. One previously published finding that may offer mechanistic insight into how phenotype can influence circadian rhythms is the suppression of *Bmal1* by LPS-inducible miR-155[54]. It has also been observed that ROR α -mediated activation of *Bmal1* transcription is enhanced by PPAR γ co-activation[112]. In macrophages, PPAR γ expression is induced upon stimulation with IL-4 and plays a key role in alternative activation of macrophages, promoting a pro-resolution macrophage phenotype, and supporting resolution of inflammation[113-115]. Such observations prompt the question of whether there are yet-unidentified factors induced downstream of various polarizing stimuli that can modulate expression of circadian genes at the transcriptional and protein levels. Further work is required to understand the interplay between macrophage phenotype and circadian rhythms.”*

2. The scale in Figure 2C should be equal for all conditions (e.g., -200).

We appreciate Reviewer #1's preference for the axes to be scaled similarly to enable cross-comparison between graphs. However, due to the different amplitude of pro-inflammatory macrophages compared to the others, we feel that making all axes the same will make it hard to see the rhythms of pro-inflammatory macrophages, hindering the reader's ability to observe the data. Thus, we have put the matched-axis plots, shown below, in Supplementary Figure 4A.

3. Absolute values of amplitude, damping, and period differ between Figure 1 and Figure 2A, B, C. The authors should explain these discrepancies.

As with many experimental approaches, there is slight variation in absolute values between independent experiments, which Reviewer #1 correctly notes. However, while the absolute values vary slightly, the relationship between the values in each of these conditions remains the same across the panels mentioned by Reviewer #1.

4. The authors should consider modulating the acidic environment of macrophages in settings more representative of cancer. For example, by adding conditioned media from tumor cells with pronounced glycolysis.

We appreciate Reviewer #1's desire to more closely mimic the tumor microenvironment. To address Reviewer #1's point, we cultured macrophages in RPMI or cancer cell (KCKO) supernatant at pH 6.5 or pH-adjusted to pH 7.4 and assessed rhythms by measuring rhythmic activity of Per2-Luc with LumiCycle analysis. We then compared changes in rhythms between macrophages cultured normal media to cancer cell supernatant in pH-matched conditions to assess how cancer cell-conditioned media may influence circadian rhythms of macrophages, and the contribution of acidic pH. We have added these data, shown below, as a new Supplementary Figure 5, and included a discussion of these data in the manuscript. Please see the new Figure and updated text below.

“Cancer cell supernatant alters circadian rhythms in macrophages in a manner partially reversed by neutralization of pH.”

We have observed that polarizing stimuli, acidic pH, and lactate can alter circadian rhythms. However, the tumor microenvironment is complex. Cancer cells secrete a variety of factors and deplete nutrients in the environment. To model this, we cultured BMDMs in RPMI or supernatant collected from KCKO cells, which are a murine model of pancreatic ductal adenocarcinoma (PDAC)[94, 95], at pH 6.5 or neutralized to pH 7.4 (Supplementary Figure 5). Circadian rhythms of BMDMs cultured in cancer cell supernatant at pH 7.4 or pH 6.5 exhibited increased amplitude and lengthened period compared to RPMI control at pH 7.4 or 6.5, respectively, indicating that cancer cell supernatant contains factors that can alter

circadian rhythms of BMDMs. Notably, BMDMs cultured in cancer cell supernatant at pH 6.5 had increased amplitude and shortened period compared to BMDMs cultured in cancer cell-conditioned media at pH7.4, indicating that pH-driven changes in rhythms were maintained in BMDMs cultured in cancer cell supernatant. When the pH of cancer cell supernatant was neutralized to pH7.4, the increased amplitude was decreased, and the shortened period was lengthened, indicating that neutralizing acidic pH partially reverses the changes in rhythms observed in macrophages cultured in cancer cell supernatant at pH 6.5. These data further support our observations that acidic pH can alter circadian rhythms of macrophages both alone and in combination with various factors in the TME.”

And, in the Discussion:

“We have shown that various stimuli can alter rhythms of macrophages in a complex and contributing manner, including polarizing stimuli, acidic pH, and lactate. TGF β is produced by a variety of cells within the TME, and was recently identified as a signal that can modulate circadian rhythms[123, 124]. Additionally, when we exposed macrophages to cancer cell-conditioned media, rhythms were modulated in a manner distinct from acidic pH or lactate, with these changes in rhythms partially reversed by neutralization of the cancer cell-conditioned media pH (Supplementary Figure 5). It is conceivable that, in addition to acidic pH, other stimuli in the TME are influencing circadian rhythms to drive population-level disorder that we observed by CCD.”

5. Arg1 alone is not sufficient as an M2 polarization marker. The authors should include additional markers.

We thank Reviewer #1 for bringing up this critical point in experimental rigor. While Arg1 is a commonly-used marker for M2 polarization, Reviewer #1 points out that polarization of macrophages is typically assessed by a full panel of markers characteristic of the M2 state. To address this point, we have expanded our panel to include several other markers of M2 polarization in mice such as Retnla, Ym1, MGL1, and CD206. In response to Reviewer 2’s major point 2 and Reviewer 3’s major point 4 below, we have also expanded our panel of markers used to assess the M1 polarization state with Tnfa, Il1b, and Il6. We have added these data, shown below, to Supplementary Figure 1 and updated the text appropriately. Please see the new Figure and updated text below.

“Consistent with previous studies, we found that genes associated with anti-inflammatory and pro-resolution programming characteristic of IL-4 and IL-13-stimulated macrophages such as Arg1, Retnla, Chil3 (Ym1), Clec10a (MGL1), and Mrc1 (CD206) were induced in IL-4 and IL-13-stimulated macrophages, but not IFN γ and LPS-stimulated macrophages. In contrast, genes associated with pro-inflammatory activity characteristic of IFN γ and LPS-stimulated macrophages such as Nos2 (iNOS), Tnfa, Il1b, and Il6 were induced in IFN γ and LPS-stimulated macrophages, but not IL-4 and IL-13-stimulated macrophages (Supplementary Figure 1)[28, 30, 65, 71, 74, 75]. This indicates that macrophages stimulated with IL-4 and IL-13 were polarized toward a pro-resolution phenotype, while macrophages stimulated with IFN γ and LPS were polarized toward a pro-inflammatory phenotype.”

Significance

While the manuscript provides valuable insights and has obvious novelty, it requires a significant revision

We thank Reviewer #1 for their deep read of our manuscript, and their helpful feedback and suggestions. As shown by the comments above, we are confident we have fully addressed each of the points that were made to result in a much-improved revised manuscript.

Reviewer #2

Evidence, reproducibility and clarity

Knudsen-Clark et al. showed that the circadian rhythm of bone marrow-derived macrophages (BMDM) can be affected by polarization stimuli, pH of the microenvironment, and by the presence of sodium-lactate. Mechanistically, the acidic pH of cell microenvironment is partly regulated by intracellular cAMP-mediated signaling events in BMDM. The authors also showed that the circadian clock of peritoneal macrophages is also modified by the pH of the cell microenvironment. Using publicly available data, the authors showed that the circadian rhythm of tumor-associated macrophages is similar to that of Bmal1-KO peritoneal macrophages. In a murine model of pancreatic cancer, the authors showed that the tumor growth is accelerated in

C57BL/6 mice co-injected with cancer cells and Bmal1-KO BMDM as compared to mice co-injected with cancer cell and wild type BMDM.

We thank Reviewer #2 for their insightful and helpful comments and feedback. Their Review guided key clarifying experiments and additions to the Discussion and Methods. To summarize, we added new data to Supplementary Figure 1 to characterize distinct gene expression in our different polarized macrophage populations, showed in Supplementary Figure 2 that serum shock independently induces cAMP and *Icer*, discussed the limitations of the artificial polarization models more clearly, and updated our Methods to better explain how macrophages were isolated from the peritoneum. We also quantified multiple immunoblots of pCREB, provided clarity in the Methods and Reviewer-only data on how our protein-extraction protocol isolates nuclear protein, better introduced the BMAL1-KO mouse model, and showed in Supplementary Figure 6 that low pH can induce oscillations in the absence of a serum shock.

Major points of criticism:

1. Nine main figures include different experimental models on a non-systematic manner in the manuscript, and only literature-based correlation is used to link the results each other. The authors used in vitro BMDM and peritoneal cell-based model systems to study the effects of IL4+IL13, IFN γ +LPS, low pH, sodium-lactate, adenylate cyclase inhibitors on the circadian clock of macrophages. The link between these microenvironment conditions of the cells is still correlative with the tumor microenvironment: publicly available data were used to correlate the increased expression level of cAMP-activated signaling events with the presence of acidic pH of tumor microenvironment. Notably, the cell signaling messenger molecule cAMP is produced by not only low extracellular pH by activated GPCRs, but also starvation of the cell. The starvation is also relevant to this study, since the BMDM used in the in vitro culture system were starving for 24 hours before the measurement of Per2-Luc expression to monitor circadian rhythm.

We agree with the important point that Reviewer #2 makes that our synchronization protocol of serum starvation followed by serum shock can impact the cAMP signaling pathway. Indeed, it has previously been shown that serum shock induces phosphorylation of CREM in rat fibroblasts, which is indicative of signaling through the cAMP pathway. To address this point, we have added a schematic of our synchronization protocol to Supplementary Figure 2B for additional clarity. We have also performed additional experiments to test whether cAMP signaling is induced in macrophages by our synchronization protocol. For this, we assessed downstream targets of the cAMP signaling pathway, *Icer* and pCREB, after serum starvation but before serum shock, and at several time points post-treatment with serum shock (Supplementary Figures 2D,E). We observed that *Icer* and phosphorylation of Creb are induced rapidly in macrophages upon exposure to serum shock, as early as 10 minutes for pCREB and 1 hour post-exposure for *Icer*. Notably, this signaling is transient and rapidly returns to baseline, with pCREB levels fully returned to baseline by 2 hours post-treatment, at which time media is replaced and the experiment begins (CT 0). These data, shown below, have been added to Supplementary Figure 2 and a discussion of these data has been added to the manuscript – please see the modified text below.

“The synchronization protocol we use to study circadian rhythms in BMDMs involves a 24-hour period of serum starvation followed by 2 hours of serum shock. It has previously been shown that serum shock can induce signaling through the cAMP pathway in rat fibroblasts[98]. To determine whether the synchronization protocol impacts cAMP signaling in macrophages, we harvested macrophages before and after serum shock. We then assessed *Icer* expression and phosphorylation of cyclic AMP-response element binding protein (CREB), which occur downstream of cAMP and have been used as readouts to assess induction of cAMP signaling in macrophages[29, 96, 100]. Serum shock of macrophages following serum starvation led to rapid phosphorylation of CREB and *Icer* expression that quickly returned to baseline (Supplementary Figure 2D,E). This indicates that serum starvation followed by serum shock in the synchronization protocol we use to study circadian rhythms in BMDMs induces transient signaling through the cAMP signaling pathway.”

2. The definition of pre-resolution macrophages (MF) used across the manuscript could be argued. The authors defined BMDM polarized with IL-4 and IL-13 as pre-resolution MF. Resolution is followed by inflammation, but the IL-4 secretion does not occur in every inflammatory setting. Moreover, IL-4 and IL-13 are secreted during specific tissue environment and immunological settings involving type 2 inflammation or during germinal center reactions of the lymph nodes.

- What are the characteristics of pre-resolution macrophages (MF)? The authors indicated that IL-4 and IL-13 cytokines were used to model the pre-resolution macrophages. In which pathological context are these cytokines produced and induce pre-resolution macrophages? IL-4 polarized BMDM can also produce pro-inflammatory protein and lipid mediators as compared to LPS-stimulated BMDM, and IL-4 polarized BMDM still have potent capacity to recruit immune cells and to establish type 2 inflammation.
- The authors showed Arg1 and Vegfa qPCR data from BMDM only. Based on the literature, these MFs are anti-inflammatory cells particularly. Resolution-related MFs followed by acute inflammation are a specific subset of MFs, and the phenotype of pre-resolution MF should be described, referred, and measured specifically.

We thank Reviewer #2 for bringing up this important point that clarity is required in describing our *in vitro* macrophage models. We chose the most commonly used models of *in vitro* macrophage polarization in the tumor immunology field, M2 (IL-4+IL-13) and M1 (IFN γ +LPS). These polarization conditions have been used for over two decades in the field, and have been well-characterized to drive a pro-inflammatory (for M1) and pro-resolution or anti-inflammatory (for M2) macrophage phenotype (Murray 2017 *Annu Rev Phys*). Each of these cell states have similarities in phenotype to pro-inflammatory and pro-resolution (pro-tumorigenic) macrophages found in tumors. In fact, in the literature, pro-inflammatory and pro-resolution TAMs will frequently be categorized as “M1” or “M2”, respectively, even though this is a gross oversimplification (Ding 2019 *J Immunol*, Garrido-Martin 2020 *J Immunother Cancer*).

As Reviewer #2 points out, IL-4 and IL-13 play a role in inflammatory settings, mediating protective responses to parasites and pathological responses to allergens. Importantly, IL-4 and IL-13 are also key regulators and effectors of resolution and wound repair (Allen 2023 *Annu Rev Immunol*). In line with this, M2 macrophages show many of the characteristics of pro-resolution programming in their gene expression profile, expressing genes associated with wound healing (ex. *Vegf*) and immunoregulation (ex. *Arg1*) (Mantovani 2013 *J Pathol*). These cells have frequently been used as a model for studying TAMs *in vitro*, due to the similarity in pro-resolution programming that is dysregulated/hijacked in TAMs (Biswas 2006 *Blood*). M2 macrophages have also been referred to as anti-inflammatory, and this is in line with their role in the type 2 response driven by IL-4 and IL-13, as this is primarily a response induced by allergy or parasites where tissue damage drives an anti-inflammatory and pro-resolution phenotype in macrophages (Pesce 2009 *Plos Pathogens* and Allen 2023 *Annu Rev Immunol*).

We do not assert that these *in vitro* models recapitulate the macrophage polarization cycle that Reviewer #2 astutely describes, and indeed, stimuli polarizing macrophages in tumor are much more diverse and complex (Laviron 2022 *Cell Rep*). We also fully agree with Reviewer #2 that, while IL4 and IL13 may exist in the tumor and be secreted by Th2 CD4 T cells (see Shiao 2015 *Cancer Immunol Res*), there may be multiple reasons why macrophages may be polarized to a pro-resolution, M2-like state in a tumor (in fact, exposure to low pH and lactate each independently do this, as we show in Supplementary Figure 2 and Figure 4, and was previously shown in Jiang 2021 *J Immunol* and Colegio 2014 *Nature*). Nonetheless, using the well-described M1 and M2 *in vitro* models allows our findings to be directly comparable to the vast literature that also uses these models, and to understand how distinct polarization states respond to low pH.

We fully agree with Reviewer #2 that these cells must be defined more clearly in the text. We have taken care to discuss the limitations of using *in vitro* polarization models to study macrophages in our Limitations of the Study section. To better address Reviewer #2's concern, we have more thoroughly introduced the M2 macrophages as a model, and are clear that that these are type 2-driven macrophages that share characteristics of pro-resolution macrophages. We have also added additional citations to the manuscript, including those highlighted above in our response. Finally, we have expanded our panel to better characterize the IL-4/IL-13 stimulated macrophages using more markers that have been characterized in the literature, in line with both Reviewer #2's comments and that of Reviewer #1 and Reviewer #3. Please see the updated data and text, below.

“As macrophages are a phenotypically heterogeneous population in the TME, we first sought to understand whether diversity in macrophage phenotype could translate to diversity in circadian rhythms of macrophages. To this end, we used two well-established in vitro polarization models to study distinct macrophage phenotypes[5, 60-63]. For a model of pro-inflammatory macrophages, we stimulated macrophages with IFN γ (interferon γ) and LPS (lipopolysaccharide) to elicit a pro-inflammatory phenotype[60, 64]. These macrophages are often referred to as ‘M1’ and are broadly viewed as anti-tumorigenic, and we will refer to them throughout this paper as pro-inflammatory macrophages[65, 66]. For a model at the opposite end of the phenotypic spectrum, we stimulated macrophages with IL-4 and IL-13[60, 67]. While these type 2 stimuli play a role in the response to parasites and allergy, they are also major drivers of wound healing; in line with this, IL-4 and IL-13-stimulated macrophages have been well-characterized to adopt gene expression profiles associated with wound-healing and anti-inflammatory macrophage phenotypes[68-71]. As such, these macrophages are often used as a model to study pro-tumorigenic macrophages in vitro and are often referred to as ‘M2’ macrophages; throughout this paper, we will refer to IL-4 and IL-13-stimulated macrophages as pro-resolution macrophages[66, 72, 73]. Consistent with previous

studies, we found that genes associated with anti-inflammatory and pro-resolution programming characteristic of IL-4 and IL-13-stimulated macrophages such as Arg1, Retnla, Chil3 (Ym1), Clec10a (MGL1), and Mrc1 (CD206) were induced in IL-4 and IL-13-stimulated macrophages, but not IFN γ and LPS-stimulated macrophages. In contrast, genes associated with pro-inflammatory activity characteristic of IFN γ and LPS-stimulated macrophages such as Nos2 (iNOS), Tnfa, Il1b, and Il6 were induced in IFN γ and LPS-stimulated macrophages, but not IL-4 and IL-13-stimulated macrophages (Supplementary Figure 1)[28, 30, 65, 71, 74, 75]. This indicates that macrophages stimulated with IL-4 and IL-13 were polarized toward a pro-resolution phenotype, while macrophages stimulated with IFN γ and LPS were polarized toward a pro-inflammatory phenotype.

In the Limitations of the Study section, we now write the following:

“Our observations of rhythms in macrophages of different phenotypes are limited by in vitro polarization models. It is important to note that while our data suggest that pro-inflammatory macrophages have suppressed rhythms and increased rate of desynchrony, it remains unclear the extent to which these findings apply to the range of pro-inflammatory macrophages found in vivo. We use IFN γ and LPS co-treatment in vitro to model a pro-inflammatory macrophage phenotype that is commonly referred to as ‘M1’, but under inflammatory conditions in vivo, macrophages are exposed to a variety of stimuli that result in a spectrum of phenotypes, each highly context-dependent. The same is true for ‘M2’; different tissue microenvironment are different and pro-resolution macrophages exist in a spectrum.”

3. The authors used IFN γ and LPS, or IL-4 and IL-13 and co-treatments to polarize BMDM in to type 1 (referred as pro-inflammatory MF) and type 2 (referred as pre-resolution MF) activation state. The comparison between these BMDM populations has limitations, since LPS induces a potent inflammatory response in MF. The single treatment with MF-polarizing cytokines enable a more relevant comparison to study the circadian clock in classically and alternatively activated MF.

We thank Reviewer #2 for bringing up this important point to provide additional clarity on our polarization conditions. The use of IFN γ and LPS to polarize macrophages toward a pro-inflammatory, M1 phenotype, and the use of IL-4 and IL-13 to polarize macrophages toward a pro-resolution, M2 phenotype have been commonly used for over two decades, and thus are well-characterized in the literature (please see Murray 2017 *Annu Rev Phys* for an extensive review on the history of these polarization models, as well as Hörhold 2020 *PLOS Computational Biology*, Binger 2015 *JCI*, McWhorter 2013 *PNAS*, Ying 2013 *J Vis Exp* for more recent studies using these models). The use of LPS alone or in combination with IFN γ , and IL-13 along with IL-4, was introduced in 1998 (Munder 1998 *J Immunol*). This approach was originally designed to mimic what could happen when macrophages were exposed to CD4+ Th1 cells, which produce IFN γ , or Th2 cells, which produce IL-4 and IL-13 (Munder 1998 *J Immunol*,

Murray 2017 *Annu Rev Phys*). As Reviewer #2 points out, these stimuli induce potent responses, driving macrophages to adopt pro-inflammatory or pro-resolution/anti-inflammatory phenotypes that are two extremes at opposite ends of the spectrum of macrophage phenotypes (Mosser 2008 *Nat Rev Immunol*). Since our goal was to study rhythms of distinct macrophage phenotypes *in vitro*, and how TME-associated conditions such as acidic pH and lactate affect their rhythms, these cell states were appropriate for our questions. Thus, the polarization models used in this paper allowed us to achieve this goal. We include a section in the Discussion on the limitations of *in vitro* polarization models.

*“A critical question in understanding the role of circadian rhythms in macrophage biology is determining how different polarization states of macrophages affect their internal circadian rhythms. This is especially important considering that tumor-associated macrophages are a highly heterogeneous population. Our data indicate that compared to unstimulated macrophages, rhythms are enhanced in pro-resolution macrophages, characterized by increased amplitude and improved ability to maintain synchrony; in contrast, rhythms are suppressed in pro-inflammatory macrophages, characterized by decreased amplitude and impaired ability to maintain synchrony (Figure 1). These agree with previously published work showing that polarizing stimuli alone and in combination with each other can alter rhythms differently in macrophages[80, 81]. In a tumor, macrophages exist along a continuum of polarization states and phenotypes[18-21, 24]. Thus, while our characterizations of rhythms in *in vitro*-polarized macrophages provide a foundation for understanding how phenotype affects circadian rhythms of macrophages, further experiments will be needed to assess macrophages across the full spectrum of phenotypes. Indeed, alteration of rhythms may be just as highly variable and context-dependent as phenotype itself.”*

4. There are missing links between the results of showing the circadian rhythm of polarized BMDM, sodium-lactate treated BMDM, and tumor growth. Specifically, do the used pancreatic ductal adenocarcinoma cells produce IL-4 and sodium-lactate? In the LLC-based experimental *in silico* analysis of tumors, the LLC do not produce IL-4.

Reviewer #2 raises important points about the source of lactate and IL-4 in tumors as relevance for our investigation of how these factors can alter rhythms in macrophages. Tumor-infiltrating Th2 CD4 T cells are potential sources of IL-4 and IL-13 in the tumor (see Shiao 2015 *Cancer Immunol Res*). Various cells in the tumor can produce lactate. We discuss this in both the Introduction and the Results: poor vascularization of tumors results in hypoxia areas, where cells are pushed toward glycolysis to survive and thus secrete increased glycolytic waste products such as protons and lactate. As lactate is lactic acid, ionized it is sodium l-lactate.

5. How can the circadian rhythm affect the function of BMDM? The Authors should provide evidence that circadian rhythm affects the function of polarized MF.

We agree with Reviewer #2 that the next step is to determine how altered rhythms influence function of macrophages. This will be the topic of future work, but is outside the scope of this paper. Our contribution with this paper is providing the first evidence that rhythms are

altered in the TME and the TME-associated conditions can alter rhythms in macrophages. We have added what is currently known about how circadian rhythms influence macrophages function to the discussion section to facilitate a conversation about this important future direction. Please see the updated text below.

“Considering our observations that conditions associated with the TME can alter circadian rhythms in macrophages, it becomes increasingly important to understand the relevance of macrophage rhythms to their function in tumors. It has been shown that acidic pH and lactate can each drive functional polarization of macrophages toward a phenotype that promotes tumor growth, with acidic pH modulating phagocytosis and suppressing inflammatory cytokine secretion and cytotoxicity[28, 30, 93]. However, how the changes in circadian rhythms of macrophages driven by these conditions contributes to their altered function remains unknown. Current evidence suggests that circadian rhythms confer a time-of-day-dependency on macrophage function by gating the macrophage response to inflammatory stimuli based on time-of-day. As such, responses to inflammatory stimuli such as LPS or bacteria are heightened during the active phase while the inflammatory response is suppressed during the inactive phase. An important future direction will be to determine how changes in circadian rhythms of macrophages, such as those observed under acidic pH or high lactate, influences the circadian gating of their function.”

6. In Figure 3, the authors show data from peritoneal cells. The isolated peritoneal cells are not pure macrophage populations. Based on the referred article in the manuscript, the peritoneal cavity contains more than 50% of lymphocytes, and the myeloid compartment contains 80% macrophages.

Reviewer #2 raises important concerns about the purity of the peritoneal population used in our experiments. We enrich for peritoneal macrophages from the peritoneal exudate cells by removing non-adherent cells in culture. This is described in our Methods section and is a method of isolation that is commonly used in the field, as lymphocytes are non-adherent. In addition to the source cited in the paper within our Methods section (Goncalves 2015 *Curr Prot Immunol*), please see Layoun 2015 *J Vis Exp*, de Jesus 2022 STAR Protocols, and Harvard HLA Lab protocol – macrophages enriched in this manner have been shown to be over 90% pure. We have modified our Methods section to make this clear, and added the additional references in this response to this section of our Methods. Please see the modified text below.

“Peritoneal exudate cells were harvested from mice as previously published[137]. To isolate peritoneal macrophages, peritoneal exudate cells were seeded at 1.2×10^6 cells/mL in RPMI/10% HI FBS supplemented with 100U/mL Penicillin-Streptomycin and left at 37°C for 1 hour, after which non-adherent cells were rinsed off[136]. Isolation of peritoneal macrophages using this method has been shown to yield a population that is over 90% in purity[138, 139]. Peritoneal

Full Revision

macrophages were then cultured in Atmospheric Media at pH 7.4 or 6.5 with 100µM D-luciferin, and kept at 37°C in atmospheric conditions.”

7. The figure legend of Figure 3 describes the effects of pH on the circadian rhythm of bone marrow-derived macrophages *ex vivo*. Peritoneal macrophages involve tissue resident peritoneal macrophages with yolk sac and fetal liver origin, and also involve small peritoneal MF with bone marrow origin. The altered description of results and figure legends makes confusion.

We are very grateful to Reviewer #2 for pointing out our typo. We have fixed the caption of Figure 3 to properly describe the data as “peritoneal macrophages *ex vivo*”.

8. In Figure 6C, one single Western blot is shown with any quantification. The authors should provide data of the relative protein level of p-CREB from at least 3 independent experiments. In the Western-blot part of the methods, the authors described that the pellet was discarded after cell lysis. The p-CREB is the activated form of the transcription factor CREB and there is increased binding to the chromatin to regulate gene expression. By discarding the pellet after cell lysis, the chromatin-bound p-CREB could be also removed at the same time.

We thank Reviewer 2 for bringing up this point. We agree that quantification is an important aspect of western blot. We have repeated the experiment again for n=3 and provide quantification of pCREB normalized to total protein. We have added these data, shown below, to Figure 5.

Reviewer #2 also expressed concern that we may not be capturing all of the CREB due to nuclear localization and chromatin binding. We specifically chose the lysis buffer M-Per, which is formulated to lyse the nucleus and solubilize nuclear and chromatin-bound proteins. To demonstrate this, we show in the below Figure to the Reviewer that the nuclear protein p85 is solubilized and readily detectable by western blot using our protein extraction method.

Full Revision

We have also added an additional sentence in the Methods section for clarity – please see the modified text below.

“Cells were lysed using the M-Per lysis reagent (Thermo Scientific, CAT#78501), supplemented with protease and phosphatase inhibitor cocktail (1:100; Sigma, CAT#PPC1010) and phosphatase inhibitor cocktail 2 (1:50; Sigma, CAT#P5726), with 200µM deferoxamine (Sigma, CAT#D9533). M-Per is formulated to lyse the nucleus and solubilize nuclear and chromatin-bound proteins, allowing isolation of nuclear proteins as well as cytosolic proteins. Lysates were incubated on ice for 1 hour, then centrifuged at 17,000 xg to pellet out debris; supernatant was collected.”

9. It is confusing that adenylyl cyclase inhibitor MDL-12 elevated the phospho-CREB levels in BMDM. How can the authors exclude any other inducers of CREB phosphorylation?

We agree with Reviewer #2 that it is surprising pCREB was elevated with MDL-12 treatment alone, and we do indeed think that there are other pathways contributing to this. We have addressed this point in the Discussion – please see the text below.

“The mechanism through which acidic pH can modulate the circadian clock in macrophages remains unclear. Evidence in the literature suggests that acidic pH promotes a pro-resolution phenotype in macrophages by driving signaling through the cAMP pathway[29]. It has previously been shown that cAMP signaling can modulate the circadian clock[99]. However, our data indicated that cAMP signaling was not fully sufficient to confer pH-mediated changes in circadian rhythms of macrophages (Figure 5A,B). Treatment with MDL-12, commonly known as an inhibitor of adenylyl cyclase[29, 117], resulted in suppression of pH-induced changes in amplitude of circadian rhythms but did not inhibit signaling through the cAMP signaling pathway (Figure 5C,D). While MDL-12 is commonly used as an adenylyl cyclase inhibitor, it has also been documented to have inhibitory activity toward phosphodiesterases (PDEs) and the import of calcium into the cytosol through various mechanisms[118, 119]. This is of particular interest, as calcium signaling has also been shown to be capable of modulating the circadian clock[120]. Furthermore, while acid-sensing through GPCRs have been the most well-characterized pathways in macrophages, there remain additional ways in which acidic pH can be sensed by macrophages such as acid-sensing ion channels[121, 122]. Further work is required to understand the signaling pathways through which pH can influence macrophage phenotype and circadian rhythms.”

10. It is described in the methods that BMDM were starving for 24 hours in serum-free culture media followed by serum shock (50% FBS). The cAMP production can be induced during cell starvation which should be considered for the data representation.

We appreciate that Reviewer #2 points out that our synchronization protocol of serum starvation followed by serum shock may impact the cAMP signaling pathway in macrophages, as serum shock has been shown to induce pCREB, a downstream mediator of cAMP signaling,

in rat fibroblasts. Indeed, we show in additional experiments performed (in response to Reviewer #2's major comment 1) evidence that cAMP signaling is induced in macrophages following the serum shock phase of our synchronization protocol, as indicated by elevation of Icer and pCREB. As we note above, this induction is transient and returns to baseline by 2 hours post-serum shock, the time at which we replace media and begin our experiments (CT 0).

Despite the transient nature of cAMP induction by our synchronization protocol, we agree wholeheartedly with Reviewer #2 that this must be considered in light of our experimental system in which we are studying the effect of acidic pH on circadian rhythms of macrophages, which in itself induces signaling through the cAMP signaling pathway. To address Reviewer #2's point, we have performed experiments in which we culture unstimulated BMDMs in neutral pH 7.4 or acidic pH 6.5, without prior serum starvation and serum shock (i.e. we do not submit these BMDMs to the synchronization protocol). We then observed circadian rhythms of Per2-Luc by LumiCycle to determine whether acidic pH alters circadian rhythms of BMDMs in the absence of prior serum starvation followed by serum shock. Similar to our observations in Figure 2, circadian rhythms of macrophages at pH 6.5 had increased amplitude and shortened period compared to rhythms of macrophages at pH 7.4. This indicates that pH-driven changes in circadian rhythms observed in our system are not due to the synchronization protocol. The data, shown below, have been placed in a new Supplementary Figure 6, and a discussion of these results has been added to the Results section – please see the updated text below.

“As acidic pH induces signaling through the cAMP pathway, we sought to determine whether acidic pH independently contributed to the pH-driven changes in circadian rhythms we observe in BMDMs. To test this, we omitted the synchronization step and observed BMDM rhythms by LumiCycle when cultured in neutral pH 7.4 or acidic pH 6.8 or pH 6.5 (Supplementary Figure 6). Circadian rhythms of BMDMs cultured at pH 6.5 exhibited similar changes as previously observed, with enhanced amplitude and shortened period relative to BMDMs at pH 7.4. This indicates pH-driven changes observed in circadian rhythms of BMDMs occur in the absence of prior serum starvation

and serum shock. “As acidic pH independently induces signaling through the cAMP pathway, we sought to determine whether acid pH could also independently contribute to the pH-driven changes in circadian rhythms we observe in BMDMs. To test this, we omitted the synchronization step and observed BMDM rhythms by LumiCycle when cultured in neutral pH 7.4 or acidic pH 6.8 or pH 6.5 (Supplementary Figure 6). Circadian rhythms of BMDMs cultured at pH 6.5 exhibited similar changes as previously observed, with enhanced amplitude and shortened period relative to BMDMs at pH 7.4. This indicates pH-driven changes observed in circadian rhythms of BMDMs occur in the absence of prior serum starvation and serum shock.”

11. How can the authors explain and prove that the wild type and Bmal1-KO BMDM co-injected with pancreatic cancer cells subcutaneously survive, present, and have effector functions at the same extent in the subcutaneous tissue, before and during tumor growth (Figure 9)? In other words, what kind of MF-derived parameters could be modified by disrupting the circadian rhythm of MF during tumor development? The production of MF-derived regulatory enzymes, cytokines, growth factors are affected by the disrupted circadian clock in MF?

Review #2 poses the very important question of why we see differences in tumor growth in our co-injection model, and what might be driving it. Of note, co-injection models of tumor growth are commonly used to determine macrophage-specific roles in tumor growth (Colegio 2014 *Nature*, Mills 2019 *Cell Rep*, Lee 2018 *Nat Comm*). We observed that tumor growth is altered when macrophages with disrupted circadian rhythms (BMAL1 KO) are co-injected compared to when macrophages with intact circadian rhythms (WT) are co-injected in a murine model of pancreatic cancer using KCKO cells. Our observation is supported by a previously published paper in which they used a co-injection model of melanoma, which we cite in the manuscript (Alexander 2020 *eLife*). What drives this difference in tumor growth remains an open question that is the subject of future work and is outside the scope of this paper, which focuses on our discovery that factors associated with the tumor microenvironment can alter circadian rhythms in macrophages. We have included a discussion on what is currently known about how circadian rhythms alter macrophage function, acknowledging that we have yet to answer these important questions and identifying it as interest for future work. Please see the text below.

“Considering our observations that conditions associated with the TME can alter circadian rhythms in macrophages, it becomes increasingly important to understand the relevance of macrophage rhythms to their function in tumors. It has been shown that acidic pH and lactate can each drive functional polarization of macrophages toward a phenotype that promotes tumor growth, with acidic pH modulating phagocytosis and suppressing inflammatory cytokine secretion and cytotoxicity [28, 30, 93]. However, how the changes in circadian rhythms of macrophages driven by these conditions contributes to their altered function remains unknown. Current evidence suggests that circadian rhythms confer a time-of-day-dependency on macrophage function by gating the macrophage response to inflammatory stimuli based on time-of-day. As such, responses to inflammatory stimuli such as LPS or bacteria are heightened during the active phase while the

inflammatory response is suppressed during the inactive phase. An important future direction will be to determine how changes in circadian rhythms of macrophages, such as those observed under acidic pH or high lactate, influences the circadian gating of their function. Data from our lab and others suggest that disruption of the macrophage-intrinsic circadian clock accelerates tumor growth, indicating that circadian regulation of macrophages is tumor-suppressive in models of PDAC (our work) and melanoma [109]. This agrees with complementary findings that behavioral disruption of circadian rhythms in mice (through chronic jetlag) disrupts tumor macrophage circadian rhythms and accelerates tumor growth[56]. It remains unclear whether this is through the pro-tumorigenic functions of macrophages such as extracellular matrix remodeling or angiogenesis, through suppression of the anti-tumor immune response, or a combination of both functions. Further work will be needed to tease apart these distinctions.”

Minor points of criticism:

1. The figure legends of the graphs and diagrams are missing in Figure 2D,E,F

We thank Reviewer #2 for pointing out that figure legends were missing. We have added legends for Figure 2D,E,F.

2. The BMAL1-based in vivo murine model of circadian rhythm is not introduced in the manuscript.

We thank Reviewer #2 for bringing to our attention that the BMAL1 KO macrophage model was not well-introduced in the manuscript. To address this point, we have modified the text to better introduce this model. Please see the modified text below.

“As a positive control for circadian clock disruption, we used data from BMAL1 KO peritoneal macrophages [44]. BMAL1 KO macrophages have a genetic disruption of the circadian clock due to the loss of Bmal1, the central clock gene. As a result, circadian rhythms of BMAL1 KO macrophages are disrupted, lacking rhythmicity and downstream circadian regulation of macrophage function (Supplementary Figure 8)[45, 54]. “As a positive control for circadian clock disruption, we used data from BMAL1 KO peritoneal macrophages [44]. BMAL1 KO macrophages have a genetic disruption of the circadian clock due to the loss of Bmal1, the central clock gene. As a result, circadian rhythms of BMAL1 KO macrophages are disrupted, lacking rhythmicity and downstream circadian regulation of macrophage function (Supplementary Figure 8)[45, 54].”

Significance

Knudsen-Clark et al. showed that the circadian rhythm of bone marrow-derived macrophages (BMDM) can be affected by polarization stimuli, pH of the microenvironment, and by the presence of sodium-lactate. Mechanistically, the acidic pH of cell microenvironment is partly regulated by intracellular cAMP-mediated signaling events in BMDM. The authors also showed that the circadian clock of peritoneal macrophages is also modified by the pH of the cell microenvironment. Using publicly available data, the authors showed that the circadian rhythm

of tumor-associated macrophages is similar to that of Bmal1-KO peritoneal macrophages. In a murine model of pancreatic cancer, the authors showed that the tumor growth is accelerated in C57BL/6 mice co-injected with cancer cells and Bmal1-KO BMDM as compared to mice co-injected with cancer cell and wild type BMDM.

We are grateful to Reviewer #2 for their very helpful comments and suggestions, which we believe have greatly enhanced the clarity and reproducibility of this manuscript.

Reviewer #3 (Evidence, reproducibility and clarity (Required)):

Review for Knudsen-Clark et al.

"Circadian rhythms of macrophages are altered by the acidic pH of the tumor microenvironment"

Knudsen-Clark and colleagues explore the impact of TME alterations on macrophage circadian rhythms. The authors find that both acidic pH and lactate modulate circadian rhythms which alter macrophage phenotype. Importantly, they define circadian disruption of tumor-associated macrophages within the TME and show that circadian disruption in macrophages promotes tumor growth using a PDAC line. This represents an important understanding of the crosstalk between cancer cells and immune cells as well as the understanding of how the TME disrupts circadian rhythms. The study is well-done, however, authors need to address several important points below.

We thank Reviewer #3 for their in-depth and insightful comments and suggestions, which have resulted in a much-improved manuscript. We were pleased that Reviewer #3 found the work to be "an important study that is well-done" and that it "represents an important understanding of the crosstalk between cancer cells and immune cells as well as the understanding of how the TME disrupts circadian rhythms." In response to Reviewer #3's comments, we have added several new key experiments and changes to the text. To summarize, we added new data to Supplementary Figure 1 to better characterize our macrophage polarization states, showed in Figure 3 that low pH affects peritoneal macrophage circadian gene expression in a similar fashion as bone marrow-derived macrophages, added new data in Figure 4 to show how lactate and low pH affect circadian gene expression over time, and new computational analysis to Figures 6, 7, and Supplementary Figure 9 to probe circadian gene covariance from publicly available data. We also made several key additions to the Discussion to discuss the functional implications of macrophage circadian rhythm disruption by low pH and potential mechanisms of this disruption. Finally, at the request of Reviewer #3, we consolidated several existing Figures and added new data, where appropriate, to existing figures, and we worked to describe new findings succinctly.

Major comments:

- 1) In Figures 3 and 4, the authors can include additional clock genes that can be run by qPCR. This was done in Figure 2 and was a nice addition to the data.

We agree with Reviewer #3's suggestion that an analysis of clock gene expression at the mRNA level would enhance our data in Figures 3 and 4. To address this point, we have performed short time course experiments to assess circadian clock gene expression over time

in BMDMs cultured with or without lactate at neutral or acidic pH (for Figure 4). In line with the difference in circadian rhythms of Per2-Luc levels between BMDMs cultured in the presence or absence of lactate which we observed by Lumicycle analysis, we measured changes in expression of the circadian clock genes Per2, Nr1d1, and Cry1 between macrophages cultured with 25 mM sodium-L-lactate compared to those cultured with 0 mM sodium-L-lactate at pH 6.5. We have added these data, shown below, to Figure 4, and updated the manuscript accordingly to discuss these results. Please see below for the new Figure Panel and modified text.

“Lactate was also observed to alter expression of the circadian clock genes Per2, Cry1, and Nr1d1 over time in BMDMs cultured at pH 6.5, while having more subtle effects at pH 7.4 (Figure 4C). Notably, lactate blunted the effect of pH 6.5 on Cry1 expression, while enhancing the effect of low pH on Nr1d1 expression. In all, these data indicate that concentration of lactate similar to that present in the TME can influence circadian rhythms and circadian clock gene expression of macrophages.”

As an additional measure to address Reviewer #3’s point about Figure 3 (peritoneal macrophages), we have compared expression of circadian clock genes in peritoneal macrophages cultured at neutral pH 7.4 or acidic pH 6.8 for 24 hours using a publicly available RNA-seq data set from Jiang 2021 *J Immunol* (GSE164697). In line with previous observations in macrophages cultured under acidic compared to neutral pH conditions, including the clock gene expression data from Figure 2 in BMDMs and the Per2-Luc levels observed in peritoneal macrophages in Figure 3, we found that peritoneal macrophages exhibited differences in expression of circadian clock genes when cultured at acidic pH 6.8 compared to neutral pH 7.4. We have added these data, shown below, as Figure 3B, and have updated the manuscript accordingly – please see below for the new Figure panel and modified text.

“To test whether pH-driven changes in circadian rhythms of peritoneal macrophages were reflected at the mRNA level, we compared expression of circadian clock genes in peritoneal macrophages cultured at neutral pH 7.4 or acidic pH 6.8 for 24 hours using publicly available RNA-sequencing data [30]. In line with altered circadian rhythms observed by Lumicycle, peritoneal macrophages cultured at pH 6.8 expressed different levels of circadian clock genes than peritoneal macrophages culture at pH 7.4 (Figure 3B). The trends in changes of gene expression in peritoneal macrophages cultured at pH 6.8 matched what we observed in BMDMs, where low pH generally led to higher levels of circadian clock gene expression (Figure 2D-F). These data support our observations by LumiCycle and indicate that acidic pH drives transcriptional changes in multiple components of the circadian clock. In all, these data are evidence that pH-dependent changes in circadian rhythms are relevant to *in vivo*-differentiated macrophages.”

We have also updated the Methods section appropriately

“FASTQ files from a previously published analysis of peritoneal macrophages cultured under neutral pH 7.4 or acidic pH 6.8 conditions were downloaded from NCBI GEO (accession #GSE164697) [30].”

2) There are far too many figures with minimal data in each. Please consolidate the figures. For example, Figures 1-3 can be fully combined, Figures 4-6 can be combined, and Figures 7-8 can be combined. Additionally, it is unclear if Figure 5 needs to be in the main, it can be moved to the supplement.

We appreciate the preference of Reviewer #3 to see some of the figures consolidated. We have combined Figures 5 and 6 into a single new Figure 5. Additionally, we have added new data from revisions to current figures to increase the amount of data in each figure and minimize the amount of new figures generated. In all, despite the large amount of new data added to the paper in response to Reviewer comments and suggestions (including additional data in Figure 4 and new Figures 6 and 8), our manuscript now contains 10 main Figures, only one more than the initial submission.

3) The observation that conditions like pH and lactate alter macrophage phenotype and rhythmicity are important. However, macrophage phenotype via gene expression does not always correlate to function. It is important for authors to demonstrate the effect of pH or lactate on macrophage function. This can be done using co-culture assays with cancer cells. If these experiments cannot be performed, it is suggested that authors discuss these limitations and consideration in the discussion.

Reviewer #3 correctly points out that changes in phenotype does not always correlate to changes in function. Others have shown that acidic pH and lactate can each alter macrophage phenotype, and also alter macrophage function and the ability to promote tumor growth (please see El-Kenawi 2019 *Br J Cancer*, Jiang 2021 *J Immunol*, Colegio 2014 *Nature*). How changes in rhythms influence macrophage function remains unknown and we agree with Reviewer #3 that this is an important future direction, We have added a section in the Discussion to facilitate the discussion of this important future direction. Please see the text below.

“Considering our observations that conditions associated with the TME can alter circadian rhythms in macrophages, it becomes increasingly important to understand the relevance of macrophage rhythms to their function in tumors. It has been shown that acidic pH and lactate can each drive functional polarization of macrophages toward a phenotype that promotes tumor growth, with acidic pH modulating phagocytosis and suppressing inflammatory cytokine secretion and cytotoxicity[28, 30, 93]. However, how the changes in circadian rhythms of macrophages driven by these conditions contributes to their altered function remains unknown. Current evidence suggests that circadian rhythms confer a time-of-day-dependency on macrophage function by gating the macrophage response to inflammatory stimuli based on time-of-day. As such, responses to inflammatory stimuli such as LPS or bacteria are heightened during the active phase while the inflammatory response is suppressed during the inactive phase. An important future direction will be to determine how changes in circadian rhythms of macrophages, such as those observed under acidic pH or high lactate, influences the circadian gating of their function.”

4) On line 119-122, authors describe a method for polarization of macrophages. They then reference one gene to confirm each macrophage polarization state. To more definitively corroborate proper macrophage polarization, authors should perform qPCR for additional target genes that are associated with each phenotype. For example, *Socs3*, *CD68*, or *CD80* for M1, and *CD163* or *VEGF* for M2. Alternatively, the authors should cite previous literature validating this in vitro polarization model.

We appreciate Reviewer #3's suggestion to better the phenotypic identity of our polarization models with additional canonical markers. To address this point, we have expanded our panel using transcriptional markers commonly used in the murine polarization model for M1 macrophages such as *Tnfa*, *Il6*, and *Il1b*. As discussed in the response to Reviewer #1's minor point 5 and Reviewer #2's major point 2, we have also expanded our panel to include additional markers for M2 such as *Vegf*, *Retnla*, *Ym1*, *Mgl1*, and *CD206*. We have added these new data to Supplementary Figure 1. Finally, we have added additional citations for the in vitro polarization models. Please see the modified text and new data, below.

“As macrophages are a phenotypically heterogeneous population in the TME, we first sought to understand whether diversity in macrophage phenotype could translate to diversity in circadian rhythms of macrophages. To this end, we used two well-established in vitro polarization models to study distinct macrophage phenotypes[5, 60-63]. For a model of pro-inflammatory macrophages, we stimulated macrophages with IFN γ (interferon γ) and LPS (lipopolysaccharide) to elicit a pro-inflammatory phenotype[60, 64]. These macrophages are often referred to as ‘M1’ and are broadly viewed as anti-tumorigenic, and we will refer to them throughout this paper as pro-inflammatory macrophages[65, 66]. For a model at the opposite end of the phenotypic spectrum, we stimulated macrophages with IL-4 and IL-13[60, 67]. While these type 2 stimuli play a role in the response to parasites and allergy, they are also major drivers of wound healing; in line with this, IL-4 and IL-13-stimulated macrophages have been well-characterized to adopt gene expression profiles associated with wound-healing and anti-inflammatory

macrophage phenotypes[68-71]. As such, these macrophages are often used as a model to study pro-tumorigenic macrophages in vitro and are often referred to as 'M2' macrophages; throughout this paper, we will refer to IL-4 and IL-13-stimulated macrophages as pro-resolution macrophages[66, 72, 73]. Consistent with previous studies, we found that genes associated with anti-inflammatory and pro-resolution programming characteristic of IL-4 and IL-13-stimulated macrophages such as Arg1, Retnla, Chil3 (Ym1), Clec10a (MGL1), and Mrc1 (CD206) were induced in IL-4 and IL-13-stimulated macrophages, but not IFN γ and LPS-stimulated macrophages. In contrast, genes associated with pro-inflammatory activity characteristic of IFN γ and LPS-stimulated macrophages such as Nos2 (iNOS), Tnfa, Il1b, and Il6 were induced in IFN γ and LPS-stimulated macrophages, but not IL-4 and IL-13-stimulated macrophages (Supplementary Figure 1)[28, 30, 65, 71, 74, 75]. This indicates that macrophages stimulated with IL-4 and IL-13 were polarized toward a pro-resolution phenotype, while macrophages stimulated with IFN γ and LPS were polarized toward a pro-inflammatory phenotype.

5) Several portions of the manuscript are unnecessarily long, including the intro and discussion. Please consolidate the text. The results section is also very lengthy, please consider consolidation.

We appreciate Reviewer #3's preference for a shorter manuscript. The revised manuscript, in response to the many Reviewer comments and requests, contains many new pieces of data, and we have taken care to describe these new data as briefly and simply as possible. In preparation for this Revision, we also removed and shortened several sections of the Results and Discussion where we felt extra explanation was not necessary. We will work with the editor of the journal we submit to ensure the length of the manuscript sections is compliant with the journal's guidelines.

6) The authors find that macrophage phenotype impacts rhythmicity. However, there is no mechanistic understanding of why this occurs. The authors should provide some mechanistic insight on this topic in the discussion.

We agree with Reviewer #3 that while the mechanism by which macrophage phenotype alters rhythms remains unknown, this is an important topic of discussion. While there is some literature on how circadian rhythms modulate inflammatory response (and hints at how it may influence phenotype) in macrophages, there is very little on the converse: how phenotype may influence circadian rhythms. We address this point by expanding on our Discussion – please see the modified text below.

“Elucidating the role of circadian rhythms in regulation of macrophage biology necessitates a better understanding of the crosstalk between phenotype and circadian rhythms. Although lactate polarizes macrophages toward a pro-resolution phenotype similar to acidic pH[30, 93], exposure to lactate had different effects on circadian rhythms – and in some cases, circadian clock gene expression – than exposure to acidic pH (Figure 4). Sensing of lactate occurs through different pathways than acid-sensing, which may contribute to the different ways in which

these two stimuli modulate circadian rhythms of macrophages[111]. One previously published finding that may offer mechanistic insight into how phenotype can influence circadian rhythms is the suppression of Bmal1 by LPS-inducible miR-155[54]. It has also been observed that ROR α -mediated activation of Bmal1 transcription is enhanced by PPAR γ co-activation[112]. In macrophages, PPAR γ expression is induced upon stimulation with IL-4 and plays a key role in alternative activation of macrophages, promoting a pro-resolution macrophage phenotype, and supporting resolution of inflammation[113-115]. Such observations prompt the question of whether there are yet-unidentified factors induced downstream of various polarizing stimuli that can modulate expression of circadian genes at the transcriptional and protein levels. Further work is required to understand the interplay between macrophage phenotype and circadian rhythms.”

7) The data presented in Figure 9 is very intriguing and arguably the strongest aspect of the paper. To strengthen the point, the authors could repeat this experiment with an additional cell model, another PDAC line or a different cancer line.

We appreciate Reviewer #3's comment about the impact of tumor growth data. Indeed, our finding that deletion of Bmal1 in co-injected macrophages accelerated PDAC growth has been recapitulated by others in different cancer models. This lends strength to our observations. We discuss and cite complementary work on macrophage rhythms and tumor growth in other models of cancer the Discussion, please see below.

“Data from our lab and others suggest that disruption of the macrophage-intrinsic circadian clock accelerates tumor growth, indicating that circadian regulation of macrophages is tumor-suppressive in models of PDAC (our work) and melanoma [109]. This agrees with complementary findings that behavioral disruption of circadian rhythms in mice (through chronic jetlag) disrupts tumor macrophage circadian rhythms and accelerates tumor growth[56].”

Minor Comments:

1) Data in Figure 2 is interesting and the impact on circadian rhythms is clear based on changes in amplitude and period. However, though the impact on period and amplitude is clear from Figures 2A-C, the changes in circadian gene expression are less clear. For instance, though amplitude is up in 2B, amplitude is suppressed in 2C. However, that does not appear to be reflected in the gene expression data in Figures 2E and F. The authors should comment on this.

Reviewer #3 correctly points out that there appear to be discrepancies between the LumiCycle data in Figure 2 and the circadian gene expression data in Figure 2. This discrepancy is perhaps unsurprising given that the gene expression data is only a short time course over 12 hours, while the LumiCycle data are collected over a course of 3 days. The gene expression data do not allow us to determine changes in period or rhythm. Another point of interest is that it's been shown that circadian regulation occurs on many different levels (transcriptional, post-transcriptional, translational, post-translational). As a result of this, circadian patterns observed in gene transcripts don't always match those of their encoded proteins; just

the same, circadian patterns of proteins aren't always reflected in their encoding gene transcripts (Collins 2021 *Genome Res*). Due to this multi-level regulation, we propose that the results of the LumiCycle analysis, which measures PER2-Luc levels, are a more robust readout of rhythms because they are further downstream of the molecular clock than transcriptional readouts. That said, observing changes at both the protein (by Lumicycle) and transcriptional level confirm that all components of the clock are altered by acidic pH, even if the way in which they are altered appears to differ. We have incorporated the points we raised above into the Results section.

Please see the modified text below.

“Low pH was also observed to alter the expression of the circadian clock genes Per2, Cry1, and Nr1d1 (REV-ERB α) over time across different macrophage phenotypes, confirming that multiple components of the circadian clock are altered by acidic pH (Figure 2D-F). Notably, the patterns in expression of circadian genes did not always match the patterns of PER2-Luc levels observed by LumiCycle. This is perhaps unsurprising, as circadian rhythms are regulated at multiple levels (transcriptional, post-transcriptional, translational, post-translational); as a result, circadian patterns observed in circadian proteins such as PER2-Luc do not always match those of their gene transcripts[77].”

2) On line 156-158, authors describe damping rate. I believe the authors are trying to say that damping rate increases as the time it takes cells to desynchronize decreases and vice versa. However, this point needs to be better explained.

We thank Reviewer #3 for bringing to our attention that this was not communicated clearly in the text. We have adjusted our explanation to be clearer. Please see the modified text below.

“Damping of rhythms in most free-running cell populations (defined as populations cultured in the absence of external synchronizing stimuli) occurs naturally as the circadian clocks of individual cells in the population become desynchronized from each other; thus, damping can be indicative of desynchrony within a population[84]. The damping rate increases as the time it takes for rhythms to dissipate decreases; conversely, as damping rate decreases as the time it takes for rhythms to dissipate increases.”

3) Data presented in Figures 3 and 4 are different in terms of the impact of changing the pH. The source of the macrophages is different, but the authors could clarify this further.

We thank Reviewer #3 for this comment. Our conclusion is that the impact of low pH is largely similar in Figure 3 (peritoneal macrophages) and Figure 4 (BMDMs exposed to low pH and lactate). In both Figures 3 and 4, exposure to acidic pH by culturing macrophages at pH 6.5 increased amplitude, decreased period, and increased damping rate compared to macrophages cultured at neutral pH 7.4.

4) For heatmaps shown in Figures 7 and 8, please calculate covariance and display asterisks where $P < 0.001$. This will more clearly demonstrate the loss of co-variance between the clock network as a result of clock disruption, the TME, and cell desynchrony.

We thank Reviewer #3 for the excellent suggestion to use an additional approach to assess circadian clock status in samples by measuring co-variance in the circadian clock gene network. To address this point, we have performed weighted gene co-expression network analysis (WGCNA) to calculate covariance, as was originally performed in Chun and Fortin et al Science Advances 2022. For the samples analyzed in Figure 7 (now Figure 6), we have added these data to the figure. We have applied this analysis to a new set of human data that we analyzed and added it to the new Figure 7. Finally, for the samples analyzed in Figure 8, we have added these data as a new Supplementary Figure 9. Please see the data and modified text below.

Figure 6

“Weighted gene co-expression network analysis (WGCNA) has been used as an alternate approach to measure the co-variance between clock genes and thus assess bi-directional correlations among the core clock gene network in healthy tissue and tumor samples [103]. In line with the circadian disorder observed by CCD, while many bi-directional correlations among the core clock gene network were significant and apparent in wild type peritoneal macrophages, these relationships were altered or abolished within BMAL1 KO peritoneal macrophages and TAM samples, and in some cases replaced by new relationships (Figure 6E). This indicates that there is population-level disorder in the circadian rhythms of tumor-associated macrophages in murine lung cancer.”

Figure 7

“We next assessed the status of the circadian clock in human TAMs from NSCLC patients. We performed CCD with publicly available RNA-seq data of tumor-adjacent macrophages and tumor-associated macrophages from NSCLC patients, using alveolar macrophages from healthy donors as a control[104, 105]. To assess the contribution of the acidic TME to circadian disorder, we subset TAM NSCLC patient samples into groups (Crem high TAMs and Crem low TAMs) based on median Crem expression. Notably, in macrophages from human NSCLC there was a trend toward disorder in Crem low but not Crem high TAM samples (Figure 7A,B). Additionally, the co-variance among core clock genes observed in alveolar macrophages from healthy donors was absent within Crem low and Crem high TAM samples (Figure 7C). In all, these data indicate that there is population-level disorder in the circadian rhythms of tumor-associated macrophages in humans and mice, suggesting that circadian rhythms are indeed altered in macrophages within the TME.”

Supplementary Figure 9

“CCD score worsened as populations became increasingly desynchronized, with the 12hr desynchronized population having a significantly worse CCD score than synchronized, homogenous macrophage population (Figure 8C). This indicates that as circadian rhythms of individual macrophages within a population become more different from each other, circadian disorder increases at the population-level. This is further supported by WGCNA, which revealed that the significant co-variance of circadian clock genes in the synchronized population was progressively altered and lost as the population is increasing desynchronized to 12 hours (Supplementary Figure 9).”

Reviewer #3 (Significance (Required)):

This is an important study that is well-done. It is the feeling of the reviewer that the study warrants a revision, at the discretion of the editor. The study represents an important understanding of the crosstalk between cancer cells and immune cells as well as the understanding of how the TME disrupts circadian rhythms.

We thank Reviewer #3 for their comments regarding the impact and significance of our work. As shown by the comments above, we are confident we have fully addressed each of the points that were made to result in a much-improved revised manuscript.

Dear Dr. Altman,

Thank you for the transfer of your revised manuscript from Review Commons to our editorial offices. I have now received the reports from the three referees that were asked to re-evaluate your study, you will find below. As you will see, the referees fully support publication of your study in EMBO reports. Referee #3 has some remaining concerns and suggestions to improve the manuscript I ask you to address in a final revised manuscript. Please also provide a final p-b-p-response addressing these points.

Moreover, the manuscript also needs formatting according to our journal style. Please carefully review the instructions that follow below.

When submitting your final revised manuscript, we will require:

1) a .docx formatted version of the final manuscript text (including legends for main figures, EV figures and tables), but without the figures included. Figure legends should be compiled at the end of the manuscript text.

2) individual production quality figure files as .eps, .tif, .jpg (one file per figure), of main figures and EV figures. Please upload these as separate, individual files upon re-submission.

The Expanded View format, which will be displayed in the main HTML of the paper in a collapsible format, has replaced the Supplementary information. You can submit up to 5 images as Expanded View. I would thus suggest combining the present supplementary figures to have 5 final figure files. Please follow the nomenclature Figure EV1, Figure EV2 etc. The figure legend for these should be included in the main manuscript document file in a section called Expanded View Figure Legends after the main Figure Legends section. Additional Supplementary material should be supplied as a single pdf file labeled Appendix. The Appendix should have page numbers and needs to include a table of content on the first page (with page numbers) and legends for all content. Please follow the nomenclature Appendix Figure Sx, Appendix Table Sx etc. throughout the text, and also label the figures and tables according to this nomenclature.

3) a complete author checklist, which you can download from our author guidelines (<https://www.embopress.org/page/journal/14693178/authorguide>). Please insert page numbers in the checklist to indicate where the requested information can be found in the manuscript. The completed author checklist will also be part of the RPF.

4) that primary datasets produced in this study (e.g. RNA-seq, ChIP-seq, structural and array data) are deposited in an appropriate public database. If no primary datasets have been deposited, please also state this in a dedicated section (e.g. 'No primary datasets have been generated and deposited'), see below.

The accession numbers and database should be listed in a formal "Data Availability" section (placed after Materials & Methods) that follows the model below. This is now mandatory (like the COI statement). Please note that the Data Availability Section is restricted to new primary data that are part of this study. This section is mandatory. As indicated above, if no primary datasets have been deposited, please state this in this section

Data availability

5) We now request the publication of original source data with the aim of making primary data more accessible and transparent to the reader. Our source data coordinator will contact you to discuss which figure panels we would need source data for and will also provide you with helpful tips on how to upload and organize the files.

6) Our journal encourages inclusion of *data citations in the reference list* to directly cite datasets that were re-used and obtained from public databases. Data citations in the article text are distinct from normal bibliographical citations and should directly link to the database records from which the data can be accessed. In the main text, data citations are formatted as follows: "Data ref: Smith et al, 2001" or "Data ref: NCBI Sequence Read Archive PRJNA342805, 2017". In the Reference list, data citations must be labeled with "[DATASET]". A data reference must provide the database name, accession number/identifiers and a resolvable link to the landing page from which the data can be accessed at the end of the reference. Further instructions are available at: <http://www.embopress.org/page/journal/14693178/authorguide#referencesformat>

7) Regarding data quantification and statistics, please make sure that the number "n" for how many independent experiments were performed, their nature (biological versus technical replicates), the bars and error bars (e.g. SEM, SD) and the test used to calculate p-values is indicated in the respective figure legends (also for potential EV and Appendix figures). Please also check that all the p-values are explained in the legend, and that these fit to those shown in the figure. Please provide statistical testing where applicable. Please avoid the phrase 'independent experiment', but clearly state if these were biological or technical replicates. Please also indicate (e.g. with n.s.) if testing was performed, but the differences are not significant. In case n=2, please show the data as separate datapoints without error bars and statistics. See also: <http://www.embopress.org/page/journal/14693178/authorguide#statisticalanalysis>

Please add to each legend (main and EV figures) a 'Data Information' section explaining the statistics used or providing information regarding replicates and scales.

8) Please add scale bars of similar style and thickness to microscopic images, using clearly visible black or white bars (depending on the background). Please place these in the lower right corner of the images themselves. Please do not write on or near the bars in the image but define the size in the respective figure legend.

9) Please also note our reference format:

10) We updated our journal's competing interests policy in January 2022 and request authors to consider both actual and perceived competing interests. Please review the policy <https://www.embopress.org/competing-interests> and update your competing interests if necessary. Please name this section 'Disclosure and Competing Interests Statement' and put it after the Acknowledgements section.

11) We now use CRediT to specify the contributions of each author in the journal submission system. CRediT replaces the author contribution section. Please use the free text box to provide more detailed descriptions and do NOT add an author contributions section to the manuscript text file. See also guide to authors:

<https://www.embopress.org/page/journal/14693178/authorguide#authorshippinguidelines>

12) Please add up to 5 keywords to the manuscript and order the manuscript sections like this, using these names:

Title page - Abstract - Keywords - Introduction - Results - Discussion - Methods - Data availability section - Acknowledgements - Disclosure and Competing Interests Statement - References - Figure legends - Expanded View Figure legends

13) All Materials and Methods need to be described in the main text using our 'Structured Methods' format, which is required for all research articles. According to this format, the Materials and Methods section should include a Reagents and Tools Table (listing key reagents, primers used, experimental models, software, and relevant equipment and including their sources and relevant identifiers), uploaded as separate file, followed by a Methods and Protocols section in which we encourage the authors

to describe their methods using a step-by-step protocol format with bullet points, to facilitate the adoption of the methodologies across labs. More information on how to adhere to this format as well as downloadable templates (.doc) for the Reagents and Tools Table can be found in our author guidelines (section 'Structured Methods'):

14) Please enter all the funding information also into our submission system during resubmission and make sure this is complete and similar to the one mentioned in the acknowledgements section of the manuscript text file.

15) Please provide a final abstract with not more than 175 words.

In addition, I would need from you:

- a short, two-sentence summary of the manuscript (not more than 35 words).
- three to four short (!) one sentence bullet points highlighting the key findings of your study.
- a schematic summary figure (synopsis image) in jpeg or tiff format with the exact width of 550 pixels and a height of not more than 400 pixels that can be used as a visual synopsis on our website.

I look forward to seeing a revised version of your manuscript when it is ready. Please let me know if you have questions or comments regarding the revision.

Referee #1:

In general, the Authors redesigned the figures and performed the Reviewers` requests experimentally. The Authors incorporated new experimental data into the manuscript to show the first evidence how low pH can affect the circadian rhythms of macrophage populations in tumor microenvironment. The methods and bibliography were extended significantly relevant to this study, which supports the reproducibility of the results.

Referee #2:

The authors have done an excellent job of addressing my comments. I am fully satisfied with the revised version of this manuscript. It is the opinion of this reviewer that the manuscript should be considered for publication, at the discretion of the editor.

Referee #3:

The manuscript has been significantly improved. However, there are still some aspects of the study that require clarification:

1. The manuscript presents a confusing situation regarding the self-induced lower pH in pro-inflammatory macrophages. Specifically, in Figures 1 and 2B, C, pro-inflammatory macrophages, which exhibit an acidic pH of 7.0, show lower amplitude, period, and damping compared to other conditions. However, in unstimulated macrophages, acidic pH appears to increase amplitude. How do the authors explain these contradictory results?
2. The study could be significantly strengthened by correlating clinical parameters of patients (NSCLC or PDAC for example) with high and low CREM levels in TAMs.

Rev_Com_number: RC-2024-02394
New_manu_number: EMBOR-2024-60120V1-T
Corr_author: Altman
Title: Circadian rhythms of macrophages are altered by the acidic pH of the tumor microenvironment

Response to editor and reviewers

1) a .docx formatted version of the final manuscript text (including legends for main figures, EV figures and tables), but without the figures included. Figure legends should be compiled at the end of the manuscript text.

This has been submitted.

2) individual production quality figure files as .eps, .tif, .jpg (one file per figure), of main figures and EV figures. Please upload these as separate, individual files upon re-submission. The Expanded View format, which will be displayed in the main HTML of the paper in a collapsible format, has replaced the Supplementary information. You can submit up to 5 images as Expanded View. I would thus suggest combining the present supplementary figures to have 5 final figure files. Please follow the nomenclature Figure EV1, Figure EV2 etc. The figure legend for these should be included in the main manuscript document file in a section called Expanded View Figure Legends after the main Figure Legends section. Additional Supplementary material should be supplied as a single pdf file labeled Appendix. The Appendix should have page numbers and needs to include a table of content on the first page (with page numbers) and legends for all content. Please follow the nomenclature Appendix Figure Sx, Appendix Table Sx etc. throughout the text, and also label the figures and tables according to this nomenclature.

Figures have been reorganized into 10 main Figures, 5 EV Figures, and 3 Appendix Figures. The Appendix is included with a Table of Contents, Figures, and Legends. Publication licenses for images made with Biorender have also been uploaded.

3) a complete author checklist, which you can download from our author guidelines (<https://www.embopress.org/page/journal/14693178/authorguide>). Please insert page numbers in the checklist to indicate where the requested information can be found in the manuscript. The completed author checklist will also be part of the RPF.

The completed checklist is included, and lists page numbers where appropriate.

4) Primary datasets produced in this study (e.g. RNA-seq, ChIP-seq, structural and array data) are deposited in an appropriate public database. If no primary datasets have been deposited, please also state this in a dedicated section (e.g. 'No primary datasets have been generated and deposited'), see below.

The accession numbers and database should be listed in a formal "Data Availability" section (placed after Materials & Methods) that follows the model below. This is now mandatory (like the COI statement). Please note that the Data Availability Section is restricted to new primary data that are part of this study. This section is mandatory. As indicated above, if no primary datasets have been deposited, please state this in this section

Data availability

A data availability section with a link to complete raw and processed data is now included as a separate section.

5) We now request the publication of original source data with the aim of making primary data more

accessible and transparent to the reader. Our source data coordinator will contact you to discuss which figure panels we would need source data for and will also provide you with helpful tips on how to upload and organize the files.

Original source data are included in the FigShare link under Data Availability.

6) Our journal encourages inclusion of *data citations in the reference list* to directly cite datasets that were re-used and obtained from public databases. Data citations in the article text are distinct from normal bibliographical citations and should directly link to the database records from which the data can be accessed. In the main text, data citations are formatted as follows: "Data ref: Smith et al, 2001" or "Data ref: NCBI Sequence Read Archive PRJNA342805, 2017". In the Reference list, data citations must be labeled with "[DATASET]". A data reference must provide the database name, accession number/identifiers and a resolvable link to the landing page from which the data can be accessed at the end of the reference. Further instructions are available at:

Data citations for public datasets that were downloaded are now included in the References.

7) Regarding data quantification and statistics, please make sure that the number "n" for how many independent experiments were performed, their nature (biological versus technical replicates), the bars and error bars (e.g. SEM, SD) and the test used to calculate p-values is indicated in the respective figure legends (also for potential EV and Appendix figures). Please also check that all the p-values are explained in the legend, and that these fit to those shown in the figure. Please provide statistical testing where applicable. Please avoid the phrase 'independent experiment', but clearly state if these were biological or technical replicates. Please also indicate (e.g. with n.s.) if testing was performed, but the differences are not significant. In case n=2, please show the data as separate datapoints without error bars and statistics. See also:

<http://www.embopress.org/page/journal/14693178/authorguide#statisticalanalysis>

Please add to each legend (main and EV figures) a 'Data Information' section explaining the statistics used or providing information regarding replicates and scales.

All figures have been changed to adhere to the above guidelines. Each Figure Legend now includes a information on biological replicates and exact n. As a note: for Figures 2D-F and Appendix Figure S1C, GraphPad Prism was unable to graph individual points along with mean and SEM on the same graph. Therefore, we created new panels in Appendix Figure 3 (for Figures 2D-F) and Appendix Figure 2C (right panel) to show individual data points with the mean (but not SEM).

8) Please add scale bars of similar style and thickness to microscopic images, using clearly visible black or white bars (depending on the background). Please place these in the lower right corner of the images themselves. Please do not write on or near the bars in the image but define the size in the respective figure legend.

Not applicable.

9) Please also note our reference format:

References have been updated to adhere to the Journal format.

10) We updated our journal's competing interests policy in January 2022 and request authors to consider both actual and perceived competing interests. Please review the policy

<https://www.embopress.org/competing-interests> and update your competing interests if necessary.

Please name this section 'Disclosure and Competing Interests Statement' and put it after the Acknowledgements section.

Updated to reflect that BJA is a member of The EMBO Journal Catalysts program.

11) We now use CRediT to specify the contributions of each author in the journal submission system. CRediT replaces the author contribution section. Please use the free text box to provide more detailed descriptions and do NOT add an author contributions section to the manuscript text file. See also guide to authors:

<https://www.embopress.org/page/journal/14693178/authorguide#authorshipguidelines>

The CREDIT Taxonomy was used to assign contributions.

12) Please add up to 5 keywords to the manuscript and order the manuscript sections like this, using these names:

Title page - Abstract - Keywords - Introduction - Results - Discussion - Methods - Data availability section - Acknowledgements - Disclosure and Competing Interests Statement - References - Figure legends - Expanded View Figure legends

Keywords have been added and the paper has been reorganized to adhere to the above structure.

13) All Materials and Methods need to be described in the main text using our 'Structured Methods' format, which is required for all research articles. According to this format, the Materials and Methods section should include a Reagents and Tools Table (listing key reagents, primers used, experimental models, software, and relevant equipment and including their sources and relevant identifiers), uploaded as separate file, followed by a Methods and Protocols section in which we encourage the authors to describe their methods using a step-by-step protocol format with bullet points, to facilitate the adoption of the methodologies across labs. More information on how to adhere to this format as well as downloadable templates (.doc) for the Reagents and Tools Table can be found in our author guidelines (section 'Structured Methods'):

A Reagents and Tools table has been included.

14) Please enter all the funding information also into our submission system during resubmission and make sure this is complete and similar to the one mentioned in the acknowledgements section of the manuscript text file.

Funding information was included in the submission.

15) Please provide a final abstract with not more than 175 words.

Abstract has been updated to be 175 words.

In addition, I would need from you:

- a short, two-sentence summary of the manuscript (not more than 35 words).

Circadian rhythms of macrophages are altered by conditions associated with the tumor microenvironment including acidic pH. This may be responsible for the heterogeneity in circadian clock gene expression within the tumor-associated macrophage population.

- three to four short (!) one sentence bullet points highlighting the key findings of your study.

- Circadian rhythms of macrophages are altered by polarizing stimuli, acidic pH, and lactate.
- There is circadian disorder in the tumor-associated macrophage population.
- Heterogeneity of circadian rhythms and circadian gene expression within a population may underlie circadian disorder.
- The circadian clock in tumor-associated macrophages plays a tumor-suppressive role in pancreatic cancer.

- a schematic summary figure (synopsis image) in jpeg or tiff format with the exact width of 550 pixels and a height of not more than 400 pixels that can be used as a visual synopsis on our website.

This new Figure has been included as Summary Figure.

Reviewer Feedback

Referee #1:

In general, the Authors redesigned the figures and performed the Reviewers` requests experimentally.

The Authors incorporated new experimental data into the manuscript to show the first evidence how low pH can affect the circadian rhythms of macrophage populations in tumor microenvironment. The methods and bibliography were extended significantly relevant to this study, which supports the reproducibility of the results.

Referee #2:

The authors have done an excellent job of addressing my comments. I am fully satisfied with the revised version of this manuscript. It is the opinion of this reviewer that the manuscript should be considered for publication, at the discretion of the editor.

We thank Referees #1 and #2 for their excellent suggestions and feedback, which we feel has resulted in a greatly improved manuscript.

Referee #3:

The manuscript has been significantly improved. However, there are still some aspects of the study that require clarification.

We thank Reviewer #3 for their favorable review of the revised manuscript, and appreciate their further questions to strengthen the manuscript. These are addressed below.

1. The manuscript presents a confusing situation regarding the self-induced lower pH in pro-inflammatory macrophages. Specifically, in Figures 1 and 2B, C, pro-inflammatory macrophages, which exhibit an acidic pH of 7.0, show lower amplitude, period, and damping compared to other conditions. However, in unstimulated macrophages, acidic pH appears to increase amplitude. How do the authors explain these contradictory results?

We agree with Reviewer 3 that it is surprising and contradictory how pro-inflammatory and pro-resolution macrophages each react to low pH. Our speculation is that signaling downstream of polarization state directly interacts with components of the molecular circadian clock, leading to different responses to low pH between these states. We had previously addressed this in the Discussion, but we have re-written and re-organized this Discussion paragraph to better emphasize that this applies to differences in response to environmental stimuli between polarization states. The updated paragraph is below.

“Elucidating the role of circadian rhythms in regulation of macrophage biology necessitates a better understanding of the crosstalk between phenotype, altered environmental conditions, and circadian rhythms. The differences in response to acidic pH between pro-inflammatory and pro-resolution macrophages (Figures 1 and 2) may be due to signaling pathways downstream of polarization that directly influence circadian clock gene expression. One previously published finding that may offer mechanistic insight into how phenotype can influence circadian rhythms is the suppression of *Bmal1* by LPS-inducible miR-155[54]. It has also been observed that ROR α -mediated activation of *Bmal1* transcription is enhanced by PPAR γ co-activation[112]. In macrophages, PPAR γ expression is induced upon stimulation with IL-4 and plays a key role in alternative activation of macrophages, promoting a pro-resolution macrophage phenotype, and supporting resolution of inflammation[113-115]. Additionally, lactate polarizes macrophages toward a pro-resolution phenotype similar to acidic pH[30, 93], exposure to lactate had different effects on circadian rhythms – and in some cases, circadian clock gene expression – than exposure to acidic pH (Figure 4). Sensing of lactate occurs through different pathways than acid-sensing, which may contribute to the different ways in which these two stimuli modulate circadian rhythms of macrophages[111]. Such observations prompt the question of whether there are yet-unidentified factors induced downstream of various polarizing stimuli that can modulate expression of circadian genes at the transcriptional and protein levels. Further work is required to understand the interplay between macrophage phenotype, altered environmental conditions, and circadian rhythms.”

2. The study could be significantly strengthened by correlating clinical parameters of patients (NSCLC or PDAC for example) with high and low CREM levels in TAMs.

We thank Reviewer #3 for this suggestion, and explored it in detail. In response to Reviewer #1 Question 4 of the original reviews, we searched in detail for appropriate human data to include in the manuscript. These data had to meet two criteria: first, the TAMs isolated from human tumors had to be fresh and not cultured, since culture in neutral pH media is known to reduce *Crem/Icer* expression. Second, the study had to have a sufficient n to perform CCD analysis, which we determined to be greater than 10 per sample group. The *only* study we could find that fit these criteria is the one we analyzed in Figure 7 from GSE116946; Garrido-Martin 2020 *J Immunother Cancer*.

This paper did include some clinical parameters, but little data on outcomes. The patients in the study had low grade / early-stage tumors, their ECOG performance status was mostly either 0 or 1 (with 0 being completely healthy and 5 being deceased), and survival data were not provided. We collaborated with Dr. Matthew McCall, a biostatistician and Author on the paper, to analyze *Crem* expression as a continuous variable against the following clinical parameters which were provided: T status, N status, M status, Age, Gender, Performance status, Smoking statu, Pack-year, Asbestos exposure, Hb (g/L), WBC (*E09/L), Neutrophils (*E09/L), Lymphocytes (*E09/L), Platelets (*E09/L), Albumin (g/L), ALK mut, EGFR mut, Mutation description, CD8+/field, CD3+/field, CD4+/field, CD20+/field, CD68+/field. We found a weak positive correlation between *Crem* expression in TAMs and levels of circulating white blood cells and lymphocytes, but Dr. McCall determined that these associations did not reach significance. We suspect that TAMs taken from patients with later-stage tumors might yield more insightful clinical associations, but those data were not available to us; therefore, we request to not include these findings in the paper.

Dear Dr. Altman,

Thank you for the submission of your further revised manuscript to our editorial offices. I now went through this and your final p-b-p-response, and I consider the remaining referee points as adequately addressed.

Before I can proceed with formal acceptance, I have these editorial requests I ask you to address in a final revised manuscript:

- I would suggest this slightly shortened title:

Circadian rhythms of macrophages are altered by the acidic tumor microenvironment

- Regarding data quantification and statistics, please check again that the number "n" for how many independent experiments were performed, their nature (biological versus technical replicates), the bars and error bars (e.g. SEM, SD) and the test used to calculate p-values are indicated in the respective figure legends (also for potential EV figures and all those in the final Appendix). Please also check that all the p-values are explained in the legend, and that these fit to those shown in the figure. Please provide statistical testing where applicable. Please also clearly state if these were biological or technical replicates. Please also indicate (e.g. with n.s.) if testing was performed, but the differences are not significant. In case n=2, please show the data as separate datapoints without error bars and statistics.

If n<5, please show single datapoints for diagrams. Presently, it seems diagrams in Fig. 1A and EV2 have only partial statistics. Please check. Moreover:

- Please note that the legends for figures 3b-c is not provided in the sequential manner (legend for figure 3c is provided before legend of figure 3b). This needs to be rectified.

- Please provide exact p values in the legends of figures 1a-b; 2a-f; 3a-b; 4a-c; 5c-d, f; 6a, d-e; 7c; 8c; 9c; 10d; EV 1e; EV 2a-b; EV 3; EV 4; EV 5.

- Please indicate the statistical test used for data analysis in the legends of figures 7b-c; 8c; EV 5.

- Please note that in figures 1b; 2a-c; 3b; EV 3; there is a mismatch between the annotated p values in the figure legend and the annotated p values in the figure file that should be corrected.

- Please note that for the figure 6c, p-values and statistical tests are indicated in the legends. However, comparison for the same, """" has not been represented in the figures. Please rectify this in the figures or legends as applicable. Also, provide the exact p-value for the same.

- Please add the primer information to the reagents and tools table (and remove the primer table from the methods section). Moreover, please remove the instructions from the reagents and tools table and add callouts for the table to the methods section.

- Please make sure that all figure panels (main, EV and Appendix figures) are called out separately and sequentially. Presently, there is a callout to Supplementary Figure 9 (that does not exist) on page 44. Please check.

- Please acknowledge BioRender with a paragraph (named Graphics) in the Methods section, indicating which panels/objects were created using BioRender.com. Then, please remove all the other mentions of BioRender from the manuscript (legends, acknowledgements).

- Please note that the data callouts in the text for "GSE157878, GSE116946, GSE188549, GSE164697, GSE25585, PRJEB34772, GSE13896, GSE260641" data citations need to include "Data ref:" as a prefix. Please add this.

- Please organize the Appendix file in a way that figures and the respective legends are shown on the same page (if possible), as this is more comprehensible for the readers. It is also not necessary to put here the author names and affiliations again. Just use "Appendix for ..." (with the manuscript title) as title, followed by the table of contents.

Best,

Rev_Com_number: RC-2024-02394

New_manu_number: EMBOR-2024-60120V2

Corr_author: Altman

Title: Circadian rhythms of macrophages are altered by the acidic pH of the tumor microenvironment

All editorial and formatting issues were resolved by the authors.

Dr. Brian Altman
University of Rochester
Biomedical Genetics
601 Elmwood Ave. Box 633
Box 633
ROCHESTER, NY 14620
United States

Dear Dr. Altman,

I am very pleased to accept your manuscript for publication in the next available issue of EMBO reports. Thank you for your contribution to our journal.

Yours sincerely,

Rev_Com_number: RC-2024-02394
New_manu_number: EMBOR-2024-60120V3
Corr_author: Altman
Title: Circadian rhythms of macrophages are altered by the acidic tumor microenvironment